# Operator algebra and algorithmic construction of boundaries and defects in (2+1)D topological Pauli stabilizer codes

Zijian Liang[1†], Bowen Yang[2†], Joseph T. Iosue[3], Yu-An Chen[1*]

[1]International Center for Quantum Materials, School of Physics, Peking University, Beijing 100871, China.
[2]Center of Mathematical Sciences and Applications, Harvard University, Cambridge, Massachusetts 02138, USA.
[3]Joint Center for Quantum Information and Computer Science, University of Maryland, College Park, Maryland 20742, USA.

*Corresponding author(s). E-mail(s): yuanchen@pku.edu.cn;
[†]These authors contributed equally to this work.

## Abstract

In this paper, we present a computational algorithm for constructing all boundaries and defects of topological generalized Pauli stabilizer codes in two spatial dimensions. Utilizing the operator algebra formalism, we establish a one-to-one correspondence between the topological data-such as anyon types, fusion rules, topological spins, and braiding statistics-of (2+1)D bulk stabilizer codes and (1+1)D boundary anomalous subsystem codes. To make the operator algebra computationally accessible, we adapt Laurent polynomials and convert the tasks into matrix operations, e.g., the Hermite normal form for obtaining boundary anyons and the Smith normal form for determining fusion rules. This approach enables computers to automatically generate all possible gapped boundaries and defects for topological Pauli stabilizer codes through boundary anyon condensation and topological order completion. This streamlines the analysis of surface codes and associated logical operations for fault-tolerant quantum computation. Our algorithm applies to $\mathbb{Z}_d$ qudits, including both prime and nonprime $d$, thus enabling the exploration of topological quantum codes beyond toric codes. We have applied the algorithm and explicitly demonstrated the lattice constructions of 2 boundaries and 6 defects in the $\mathbb{Z}_2$ toric code, 3 boundaries and 22 defects in the $\mathbb{Z}_4$ toric code, 1 boundary and 2 defects in the double semion code, 1 boundary and 22 defects in the six-semion code, 6 boundaries and 270 defects in the color code, and 6 defects in the anomalous three-fermion code. In addition, we investigate the boundaries of two specific bivariate bicycle codes within a family of low-density parity-check (LDPC) codes. We demonstrate that their topological orders are equivalent to 8 and 10 copies of $\mathbb{Z}_2$ toric codes, with anyons restricted to move by 12 and 1023 lattice sites in the square lattice, respectively.

## 1 Introduction

Quantum error correction is a fundamental requirement for achieving reliable and scalable quantum computation [1–5]. Among the various quantum error-correcting codes developed, surface codes are one of the most promising candidates for practical implementation in real-world experiments [6–15]. The ability to lay qubits on a plane simplifies engineering challenges and enhances the feasibility of large-scale quantum processors. By leveraging topological properties [16–19], surface codes encode quantum information in a way that is inherently protected from local errors, ensuring the fidelity of quantum computations over extended periods. This reliability makes surface codes effective for future fault-tolerant quantum computation.

To date, most surface codes, including the Kitaev surface code [5], the color code [2, 20–23], the double semion code [17, 24, 25], and the bivariate bicycle codes [26, 27] have been meticulously designed and studied individually. The underlying mathematical framework of these error-correcting codes are Dijkgraaf-Witten topological quantum field theories (TQFTs), which describe topological orders and have lattice constructions of fixed-point wave functions [5, 28–33]. Their gapped boundaries are realized by anyon condensation of

the Lagrangian subgroup [34–43]. The original construction of topological quantum field theories (TQFTs) is intricate and often challenging to implement in practice. Ref. [25] adapts the generalized Pauli (qudit) stabilizer formalism to simplify the construction of Abelian twisted quantum doubles, including all (2+1)D Abelian topological orders that admit gapped boundaries. Qudits with nonprime dimensions extend topological Pauli stabilizer codes beyond Kitaev's toric code, enabling a broader class of quantum codes. Ref. [44] highlights the bulk-boundary correspondence in topological Pauli stabilizer codes, where the boundary Hilbert space is anomalous and restricted by the anyon data of the bulk topological order. The boundaries and defects of the standard toric code and the color code have been constructed explicitly through condensation [23, 37, 45–49]. Despite this progress, no general constructive algorithm exists for determining boundaries and defects in arbitrary topological Pauli stabilizer codes. While the theoretical framework of anyon condensation remains valid, the microscopic details necessary for practical lattice constructions remain elusive.[1]

Recently, an algorithm in Ref. [50] was developed to extract the topological orders of generalized Pauli stabilizer codes on a two-dimensional infinite plane. Extending this method to situations with boundaries and defects is essential, as it would enrich the topological information of given stabilizer codes and enable additional logical operations. For instance, introducing the $e - m$ exchange defect in the $\mathbb{Z}_2$ toric code creates non-Abelian endpoints as the Ising anyons $\sigma$ [12, 47, 51]. Distinct boundary conditions, such as $e$-condensed or $m$-condensed, enhance the versatility of surface code design. The interplay between boundaries, defects, and bulk anyons can expand the logical space and introduce new logical operators [6, 47, 52–56], which can be harnessed for universal quantum computation [57]. Therefore, developing a systematic approach for constructing boundaries and defects in general topological Pauli stabilizer codes is crucial for advancing the construction of novel surface codes. Such a framework would be helpful for quantum error correction and facilitate the implementation of two-dimensional quantum codes with open boundaries in experiments, supporting the development of scalable and fault-tolerant quantum computation. This paper presents an algorithm that efficiently constructs boundaries and defects for any two-dimensional topological Pauli stabilizer code using an operator algebra approach. The algorithm aims to streamline the development of surface codes with the aid of classical computers.

In summary, we present an algorithm for constructing gapped boundaries in topological generalized Pauli stabilizer codes with $\mathbb{Z}_d$ qudits in two spatial dimensions, applicable to both prime and nonprime qudits. Our method begins by solving for all boundary gauge operators that commute with bulk stabilizers. These boundary gauge operators are then used to form boundary string operators, which create boundary anyons at their endpoints. There is a one-to-one correspondence between bulk and boundary anyons, and the topological data, including fusion rules, topological spins, and braiding statistics, can be derived from these boundary string operators. To finish the construction of gapped boundaries, we perform boundary anyon condensation and topological order completion to obtain the boundary Hamiltonian, ensuring the topological order condition is satisfied. Defects can be constructed similarly to boundaries through the folding argument. Our algorithm is demonstrated in Fig. 1. As an application, we have implemented the algorithm to construct boundaries and defects for a variety of quantum codes, including the $\mathbb{Z}_2$ toric code, the $\mathbb{Z}_4$ toric code, the color code, the double semion code, the six-semion code, and the three-fermion code. The algorithm can also be applied to the bivariate bicycle (BB) codes from Ref. [26], offering a topological perspective on this family of low-density parity-check (LDPC) codes. Our results indicate that the periodicity[2] of anyons in the BB code family is notably long. For instance, in two cases analyzed in subsequent sections, the periodicities are 12 and 1023, respectively.

The paper is organized as follows: Sec. 2 begins by reviewing the stabilizer code formalism for topological quantum codes in bulk and extends this framework to open boundaries using the subsystem code formalism. This section explicitly defines boundary anyons and string operators and outlines the procedure for constructing gapped boundaries. Sec. 3 introduces the operator algebra formalism, which provides the mathematical foundation to rigorously prove the lemmas and theorems from the previous section, with deep connections to ring theory. In Sec. 4, we adopt the Laurent polynomial formalism to implement our algorithm practically, enabling computers to perform matrix operations, such as computing the Hermite and Smith normal forms, to derive the topological data of boundary anyons and construct gapped boundaries in Pauli stabilizer codes. Finally, Sec. 5 demonstrates the algorithm's application to various quantum codes.

---

[1]More precisely, the anyon condensation procedure generally does not preserve the topological order (TO) condition in the microscopic lattice. While specific cases, such as the standard toric code, the $\mathbb{Z}_4$ double semion code, and the color code, can maintain the TO condition, general models require an additional step to carefully design the Hamiltonian.

[2]Periodicity refers to the shortest distance an anyon can move in a given direction.

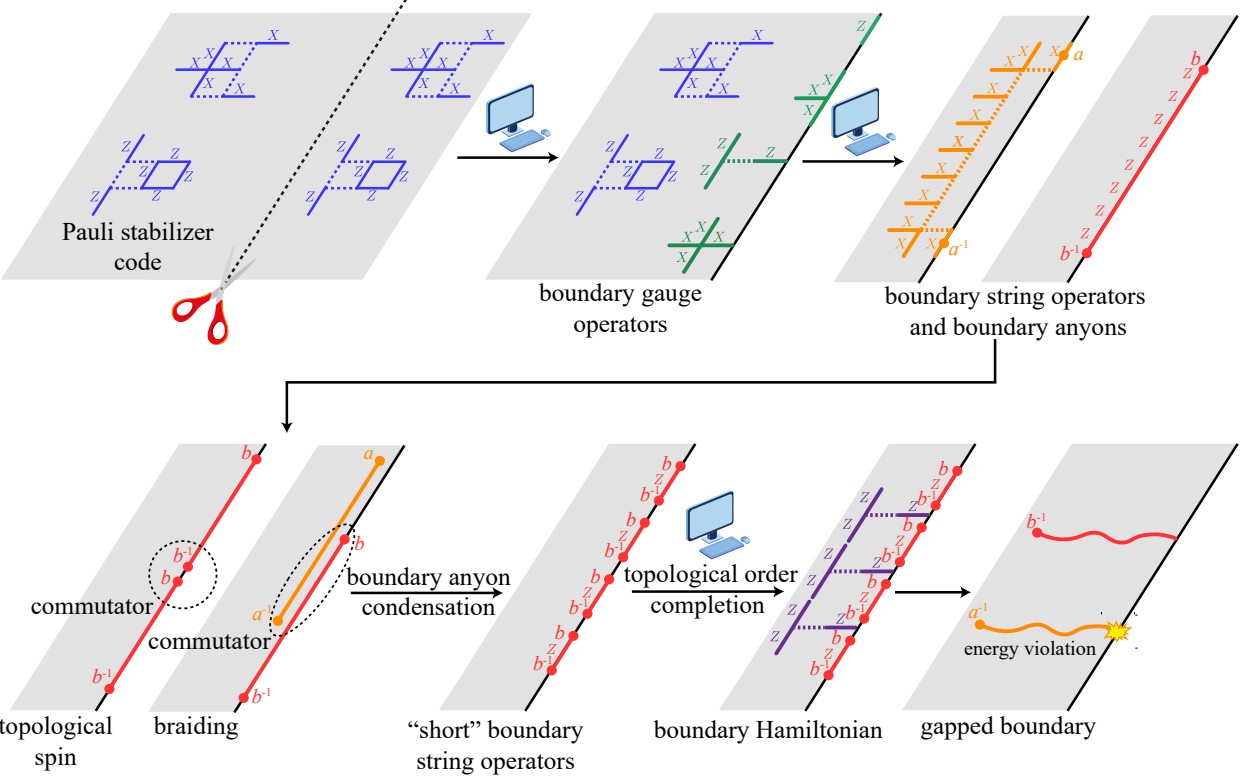

**Fig. 1** Algorithm for constructing gapped boundaries of a topological generalized Pauli stabilizer code. We begin by truncating an infinite system and solving for boundary gauge operators that commute with all bulk stabilizers. These boundary gauge operators are then used to form boundary string operators, which create boundary anyons in the (1+1)D subsystem code at their endpoints. Each boundary anyon has a one-to-one correspondence with a bulk anyon in the (2+1)D stabilizer code. The topological spin and braiding statistics are derived from the commutator of the boundary string operators. Finally, we perform boundary anyon condensation and topological order completion to construct the gapped boundary Hamiltonian. This Hamiltonian ensures that bosons in the Lagrangian subgroup can terminate at the boundary without incurring an energy penalty, whereas other anyons must violate the boundary Hamiltonian.

## 2 Physical intuition and description

This section offers a pedagogical overview of generalized Pauli operators and Abelian anyon theories within the stabilizer code formalism, focusing on the microscopic perspective. Additionally, we extend this framework to systems with open boundary conditions. We will introduce boundary gauge operators, which contrast with the bulk stabilizer operators, and subsequently define boundary anyons and their corresponding string operators. These concepts will be crucial for constructing gapped boundaries and defects.

### 2.1 Review of bulk anyons in stabilizer formalism

We begin by reviewing the bulk stabilizer code formalism and the microscopic definitions of anyons and topological data as discussed in Ref. [50]. We first consider the stabilizer code on an infinite plane with translational symmetry, which will later be truncated to a finite region with an open boundary.

Let us recall the standard definitions of $d \times d$ **generalized Pauli matrices** for a $\mathbb{Z}_d$ qudit:

$$X = \sum_{j \in \mathbb{Z}_d} |j + 1\rangle\langle j|, \quad Z = \sum_{j \in \mathbb{Z}_d} \omega^j |j\rangle\langle j|, \tag{1}$$

where $\omega$ is defined as $\omega := \exp\left(\frac{2\pi i}{d}\right)$. The matrices $X$ and $Z$ satisfy the commutation relation:

$$ZX = \omega XZ. \tag{2}$$

For simplicity, we will refer to these as "Pauli" matrices, using the term as shorthand for "generalized Pauli."

We begin by considering a local Pauli stabilizer Hamiltonian on a two-dimensional lattice that satisfies the **topological order (TO) condition** [58–62]. This condition requires that any local operator $\mathcal{O}$ commuting with all stabilizers can be written as a product of stabilizers, $\mathcal{O} = \prod_{i \in J} S_i$ for some set $J$. The TO condition implies the local indistinguishability of the ground state(s) in a stabilizer code, meaning no local operator can distinguish between them. A stabilizer code satisfying the TO condition is referred to as a **topological**

**Pauli stabilizer code**, indicating the presence of **topological order**. This topological order is described by unitary modular tensor categories (UMTCs) [19, 63–68], which characterize the low-energy excitations. Stabilizer models give rise to Abelian anyon theories, a subset of UMTCs.

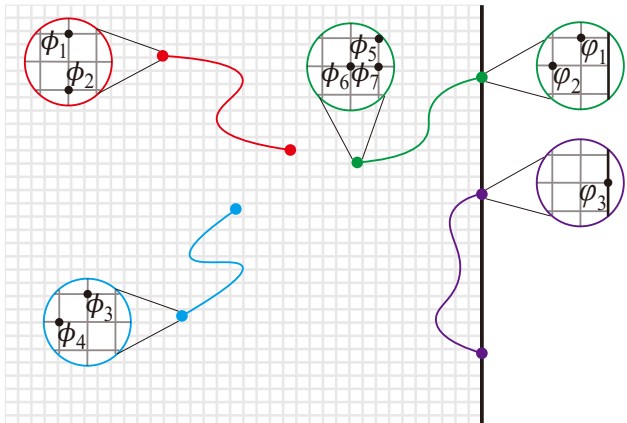

**Fig. 2** The black dots indicate the locations where stabilizers act as $\phi$ (with $\phi \neq 1$) on the state. These syndromes are labeled as $\phi_1$, $\phi_2$, and so forth. When these violated stabilizers are spatially distant from one another, we group the $\phi_i$ into separate patches, treating each patch as a local anyon. In (2+1)D topological Pauli stabilizer codes, any anyon is generated by a string operator, which is a product of Pauli matrices along a string that fails to commute with only a finite number of stabilizers near its endpoints. The concept of bulk stabilizer violations can be extended to boundary gauge operators, as discussed in Sec. 2.2.

An **anyon** is defined as a violation of stabilizers on the lattice [50, 69–71]. Given a ground state $|\Psi_{\mathrm{gs}}\rangle$, which is in the $+1$ eigenstate of all the stabilizers: $S_i|\Psi_{\mathrm{gs}}\rangle = |\Psi_{\mathrm{gs}}\rangle$. However, when a Pauli operator $M$ is applied to the ground state, the resulting perturbed state may no longer reside in the $+1$ eigenspace of the stabilizers. Specifically, we have $S_i(M|\Psi_{\mathrm{gs}}\rangle) = \phi_i(M|\Psi_{\mathrm{gs}}\rangle)$, where $\phi_i \in U(1)$ depends on the commutation relation between $S_i$ and $M$. This perturbed state is characterized by a set of $U(1)$ angles $\{\phi_1, \ldots, \phi_N\}$, which defines a homomorphism from the stabilizer group $\mathcal{S}$ to $U(1)$:

$$\phi : \mathcal{S} \to U(1), \tag{3}$$

where $\phi(S_i) = \phi_i$.

So far, violations ($\phi_i \neq 1$) have been described globally across the entire system. However, locality now plays a crucial role. If the violated stabilizers are spatially distant, for example, due to a long string operator $M$ that only affects stabilizers near its endpoints, we can group the stabilizers with $\phi_i \neq 1$ into local patches, as illustrated in Fig. 2. Each patch represents a local anyon, with $\phi_i$ forming the **syndrome pattern**. Mathematically, a **local anyon** is defined by a homomorphism $\phi$ in Eq. (3), with $\phi(S_i) = 1$ for all but a finite number of $S_i \in \mathcal{S}$, meaning the stabilizers are violated locally [72]. In two-dimensional topological Pauli stabilizer codes, all anyons can be generated by (infinite) string operators [60, 62, 72]. This property does not hold in higher-dimensional models, primarily due to the fracton phases of matter [73–80].

Anyon types, also known as superselection sectors, can be characterized as equivalence classes based on the **equivalence relation** defined between anyons $v$ and $v'$:

$$v := \{\phi_i\} \sim v' := \{\phi_i'\}, \tag{4}$$

if and only if the sets $\{\phi_i\}$ and $\{\phi_i'\}$ differ only by local Pauli operators. In other words, two syndrome patterns $v$ and $v'$ are considered equivalent if the pattern $v'$ can be obtained from $v$ by applying local Pauli operators. Utilizing the concept of anyon types, we can explore the fusion rules. The **fusion rules** of (Abelian) anyons describe the process of bringing two anyons, $a$ and $b$, into close proximity (through their string operators) and identifying their composite as a third anyon, $c$, under the equivalence relation given by Eq. (4). This fusion rule is denoted by

$$a \times b = c. \tag{5}$$

Additionally, the **topological spin** $\theta(a)$ can be calculated for each anyon $a$, which determines its exchange statistics according to the spin-statistics theorem, distinguishing types such as bosons, fermions, and semions. The T-junction process, which involves exchanging the positions of two particles, can be used to determine the topological spin [81–86]. Consider paths $\bar{\gamma}_1$, $\bar{\gamma}_2$, and $\bar{\gamma}_3$ that converge at a common endpoint $p$ and are arranged counter-clockwise around $p$, as depicted in Fig. 3. The topological spin $\theta(a)$ for anyon $a$ is calculated using the formula:

$$W_3^a(W_2^a)^\dagger W_1^a = \theta(a) W_1^a(W_2^a)^\dagger W_3^a, \tag{6}$$

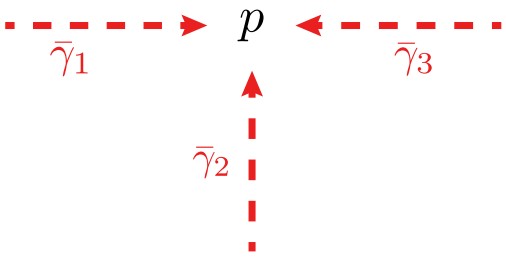

**Fig. 3** The exchange statistics, or topological spin, of an anyon $a$ can be determined using the formula presented in Eq. (6). In this context, $\bar{\gamma}_1$, $\bar{\gamma}_2$, and $\bar{\gamma}_3$ are oriented paths on the lattice that intersect at a common point $p$. The string operators $W_1^a$, $(W_2^a)^\dagger$, and $W_3^a$, corresponding to anyon $a$ and defined along the paths $\bar{\gamma}_1$, $-\bar{\gamma}_2$, and $\bar{\gamma}_3$, respectively, may not commute. This non-commutativity results in the exchange statistics $\theta(a)$.

where $W_i^a$ represents the string operator moving anyon $a$ along the path $\bar{\gamma}_i$. The **braiding statistics** between two anyons, $a$ and $b$, can be derived from their topological spins as follows:

$$B(a,b) = \frac{\theta(a \times b)}{\theta(a)\theta(b)}. \tag{7}$$

See Appendix A of Ref. [50] for a detailed derivation.

## 2.2 Defining boundary gauge operators, boundary anyons, and boundary strings

We have thus far reviewed the microscopic definition of bulk anyons and string operators within the stabilizer code formalism. We now aim to extend this framework to systems with open boundaries. Specifically, we will define boundary anyons and their associated string operators using the subsystem code formalism.

To proceed, we truncate the infinite plane to a finite subspace. The detailed geometry and shape of the boundary do not matter, but for simplicity, we use a square lattice truncated to the left semi-infinite plane $x \leq 0$ for demonstration, as shown in Fig 2. After truncation, we retain only the local stabilizers that are fully supported within the truncated region, referring to them as **bulk stabilizers**. If the original stabilizer (before truncation) contains a Pauli operator on any qudit that is truncated out, this stabilizer is no longer included in the truncated system.

Near the boundary, the TO condition is no longer satisfied. There exists a local operator that commutes with all bulk stabilizers but is not necessarily a product of bulk stabilizers. These operators are referred to as **boundary gauge operators**. More precisely, these local operators that commute with bulk stabilizers, quotiented by bulk stabilizers, form the group $\mathcal{G}$ of boundary gauge operators. Additionally, these operators might not commute with themselves. The terminology is borrowed from the definition of subsystem codes, where the gauge operators do not commute, and their commutants form the stabilizer group. As a result, the Hilbert space of the boundary theory becomes **anomalous**, meaning that only boundary gauge operators can act on it, and the space lacks a tensor product structure [87–94]. Since the bulk stabilizers locally satisfy the TO condition, for any boundary gauge operator, there is an equivalent operator (by multiplying with bulk stabilizers) supported near the boundary and does not extend into the bulk. This follows from the cleaning lemma [61, 95]. Therefore, we can assume that all boundary gauge operators are supported within a finite distance from the boundary.

Anyons in the subsystem code can be defined [95]. We use the intuition similar to Eq. (3) to define our boundary anyons and boundary string operators as follows.

**Definition 1.** A **boundary anyon** is defined as the local "syndrome pattern" of boundary gauge operators that indicates how boundary gauge operators are violated:

$$\varphi : \mathcal{G} \to U(1), \tag{8}$$

such that $\varphi(G_i) = 1$ for all but a finite number of $G_i \in \mathcal{G}$.

We will refer to the "syndrome pattern" of the boundary gauge operators as the **gauge violation**. In Sec. 3, we will prove the following theorem:

**Theorem 1.** For any gauge violation represented by a homomorphism $\varphi : \mathcal{G} \to U(1)$, there exists an (infinite) boundary gauge operator $O_\varphi$ such that $\varphi$ can be expressed as:

$$\varphi(G) = [G, O_\varphi] := G O_\varphi G^{-1} O_\varphi^{-1}, \quad \forall\, G \in \mathcal{G}. \tag{9}$$

*Proof.* See the discussion following Theorem 15. □

This theorem implies that any boundary anyon (gauge violation of boundary gauge operators) can be created by an (infinite) boundary gauge operator, as shown in Fig. 2. This theorem is analogous to the property that anyons in two dimensions can be created by bulk string operators.

Similar to bulk anyons, we can categorize boundary anyons into different equivalence classes (anyon types):

**Definition 2.** *Two boundary anyons are **equivalent** if their syndrome patterns differ by a local boundary gauge operator.*

Note that previously, bulk anyons were considered equivalent if they differed by the syndrome of a local Pauli operator. Here, we are only allowed to apply a boundary gauge operator since we need to ensure that the bulk stabilizers are never violated (only boundary gauge operators can be violated). Next, we define boundary string operators:

**Definition 3.** *A **boundary string operator** along a boundary segment is a product of boundary gauge operators in the neighborhood of this segment that creates boundary anyons at its endpoints.*

A boundary string operator does not violate bulk stabilizers and only fails to commute with boundary gauge operators at its endpoints. From the definitions above, a nontrivial boundary anyon is created by an infinite operator. Without loss of generality, we can assume a boundary anyon is created by a semi-infinite boundary string operator, containing boundary gauge operators only on one side of the anyon location.[3] First, we can show that boundary anyons are mobile along the boundary:

**Lemma 2. (Mobility of boundary anyons along the boundary)** *Given a boundary anyon at a vertex $v_\partial$, which is the endpoint of a semi-infinite boundary string operator, this string operator can be adjusted locally to end at another vertex $v'_\partial$ far from $v_\partial$, such that it only violates boundary gauge operators around $v'_\partial$. In other words, we can apply a finite string operator to move the anyon from $v_\partial$ to another vertex $v'_\partial$.*

This lemma is a direct consequence of the definitions. Given a semi-infinite string, it is evident that the boundary anyon can move in the direction of the string by truncating the string operator. To demonstrate that the anyon can also move in the opposite direction, i.e., that the semi-infinite boundary string is extendable, we first note that the syndrome pattern of the truncated string is bounded within a finite range. Therefore, as we truncate the semi-infinite string progressively, by the pigeonhole principle, there must be two vertices such that the boundary anyons at these vertices are identical (up to a translation). Consequently, we can use the finite string operators between these two vertices to "copy and paste" to form a longer semi-infinite string. Thus, the boundary anyon can move in both directions along the boundary.

A corollary follows directly from the argument above:

**Corollary 2.1. (Weak translational symmetry of boundary anyons)** *For any boundary anyon $\varphi$, there exists a local operator (finite string operator) that transforms $\varphi$ into another anyon $\varphi'$, where $\varphi'$ is $\varphi$ translated by $n$ lattice sites in the y-direction for some integer $n$.*

In other words, a boundary anyon can be shifted by a distance $n$ and still represent the same anyon type. Note that in many examples, such as the color code, $n$ cannot be 1, which means that shifting an anyon by a distance of 1 results in a different anyon. However, by the pigeonhole principle, when the shift is increased, the boundary anyon must eventually map back to itself. Therefore, it exhibits weak translational symmetry relative to the lattice.

Moreover, we will prove the following lemmas and theorems using the operator algebra formalism introduced in Sec. 3. A boundary anyon can be moved into the bulk and become a bulk anyon:

**Lemma 3. (Mobility of boundary anyons into the bulk)** *Given a boundary anyon $\varphi$ at a vertex $v_\partial$ on the boundary, there exists a bulk anyon $\phi$ at a vertex $v$ in the bulk, with a string operator along the path from $v_\partial$ to $v$ that transform the boundary anyon $\varphi$ to the bulk anyon $\phi$.*

Later, we will prove the following important theorem:

**Theorem 4. (Bulk-boundary correspondence)** *There is a one-to-one correspondence between bulk anyon types and boundary anyon types, i.e., there is a bijective mapping between a boundary anyon $\varphi$ at a vertex $v_\partial$ on the boundary and a bulk anyon $\phi$ at a vertex $v$ in the bulk. Moreover, there exist string operators along the path from $v$ to $v_\partial$ that transform one anyon to the other.*

This theorem implies that the (2+1)D bulk stabilizer code and the (1+1)D boundary anomalous[4] subsystem code are dual to each other. The topological spins and mutual braiding of the boundary anyons can be directly determined from the boundary anyon string operators (a detailed proof is provided in Appendix A):

---

[3]This is feasible because we can always divide an infinite string operator into two semi-infinite strings and discuss the two boundary anyons separately.

[4]"Anomalous" refers to a situation where the Hilbert space lacks a tensor product structure. Operators must be "gauge-invariant," meaning they commute with specific gauge constraints, such as the bulk stabilizers in our context.

**Theorem 5. (Topological spins of boundary anyons)**

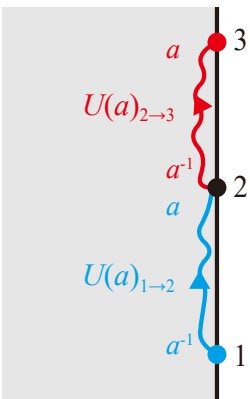

*Given boundary string operators $U(a)_{1\to2}$ and $U(a)_{2\to3}$ that move the boundary anyon $a$ from vertex 1 to 2, and from vertex 2 to 3, respectively (as shown in the figure above), the topological spin of $a$ is given by:*

$$\theta(a) = [U(a)_{1\to2}, U(a)_{2\to3}] := U(a)_{1\to2}U(a)_{2\to3}U(a)^{\dagger}_{1\to2}U(a)^{\dagger}_{2\to3}. \tag{10}$$

**Theorem 6. (Braiding between boundary anyons)**

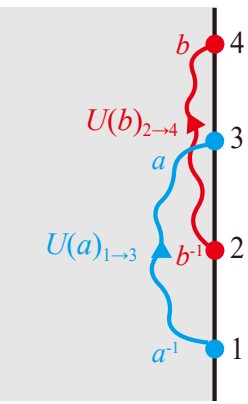

*Given boundary string operators $U(a)_{1\to3}$, which moves the boundary anyon $a$ from vertex 1 to 3, and $U(b)_{2\to4}$, which moves the boundary anyon $b$ from vertex 2 to 4 (as shown in the figure above), the braiding statistics of these two anyons is given by:*

$$B(a,b) = [U(a)_{1\to3}, U(b)_{2\to4}] := U(a)_{1\to3}U(b)_{2\to4}U(a)^{\dagger}_{1\to3}U(b)^{\dagger}_{2\to4}. \tag{11}$$

We emphasize that Eqs.(10) and (11) are consistent with the proposal in Ref. [44], which is derived from the holographic perspective of topological stabilizer codes. Therefore, without knowledge of the bulk stabilizers, we can retrieve all topological data of Abelian anyon theories—such as anyon types, fusion rules, and topological spins—using only boundary gauge operators: *The boundary carries the same topological information as the bulk.*

On the other hand, for practical purposes, such as constructing different boundaries or finding all boundary anyons using computers, information about the bulk stabilizers can save computational effort. To facilitate this, we first divide the boundary gauge operators into two types:

1. **Primary boundary gauge operators** are defined as the operators that are truncated stabilizers, i.e., original stabilizers with Pauli operators directly removed at the qudits disappearing due to truncation.[5]
2. **Secondary boundary gauge operators** are defined as the group of boundary gauge operators modulo the primary boundary gauge operators, meaning that the nontrivial elements in this group correspond to boundary gauge operators that cannot be expressed as truncated stabilizers.[6]

---

[5]More precisely, we consider these operators quotient by bulk stabilizers, so primary boundary gauge operators form a group.
[6]In the standard toric code, only primary boundary gauge operators exist, rendering the secondary boundary gauge operators trivial. Ref. [44] did not address the possibility of secondary boundary gauge operators.

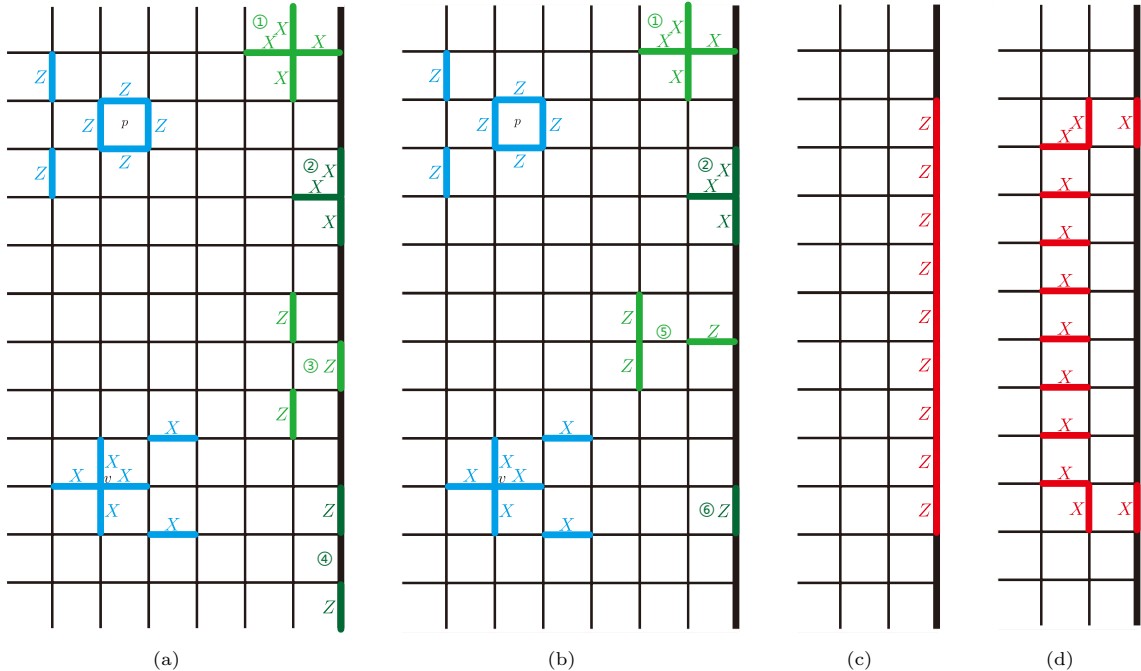

**Fig. 4** Bulk stabilizers and boundary gauge operators of the "fish toric code" in Eq. (13). (a) The blue components represent the bulk stabilizers $A_v^{\text{fish}}$ and $B_p^{\text{fish}}$. The nontrivial *primary boundary gauge operators* are generated by the green terms labeled ①, ②, ③, and ④, as well as their translational counterparts in the vertical directions. Bulk stabilizers are considered trivial primary boundary gauge operators. Note that the primary boundary operators are truncated bulk stabilizers $A_v^{\text{fish}}$ and $B_p^{\text{fish}}$ and therefore commute with all boundary stabilizers. (b) All boundary gauge operators are generated by the terms labeled ①, ②, ⑤, and ⑥, as well as their translational counterparts. The terms ⑤ and ⑥ are nontrivial *secondary boundary gauge operators*, i.e., they commute with all bulk stabilizers but are not truncated $A_v^{\text{fish}}$ and $B_p^{\text{fish}}$.

To demonstrate the primary and secondary boundary gauge operators, we consider the following example **fish toric code**, which is the $\mathbb{Z}_2$ standard toric code conjugated by a finite-depth Clifford circuit:

$$H^{\text{fish}} = -\sum_v A_v^{\text{fish}} - \sum B_p^{\text{fish}}, \tag{12}$$

where the $A_v^{\text{fish}}$ and $B_p^{\text{fish}}$ terms are

$$A_v^{\text{fish}} = \begin{array}{c} X \\ {-}X{-}\!\!\begin{array}{c}|v\\ \end{array}\!\!X{-} \\ X \end{array} \begin{bmatrix} {-}X{-} \\ \\ {-}X{-} \end{bmatrix}, \quad B_p^{\text{fish}} = \begin{array}{c} Z \\ Z \\ Z \;\; p \;\; Z \\ Z \\ Z \end{array}. \tag{13}$$

Its primary and secondary boundary gauge operators are shown in Fig. 4(a) and 4(b).

In Sec. 3, we will prove the following theorems to simplify the process of finding boundary string operators and boundary anyons.

**Theorem 7.** *A boundary anyon is uniquely determined by the syndrome pattern of primary boundary gauge operators.*

*Proof.* See the discussion following Corollary 14.1. □

**Theorem 8.** *Given a boundary string operator, we can multiply local boundary gauge operators around its endpoints so that this modified boundary string operator is a product of primary boundary gauge operators.*

*Proof.* See the discussion following Lemma 16. □

Caveat: We still require that the boundary string operators commute with both primary and secondary boundary gauge operators except around their endpoints.

According to Theorem 7, when comparing two boundary anyons, it is sufficient to know the syndrome pattern of the primary boundary gauge operators. Similarly, Theorem 8 states that when solving for boundary string operators to identify boundary anyons, we only need to consider string operators formed by primary boundary gauge operators. These two theorems reduce the computational complexity.

For the fish toric code defined in Eqs. 12 and 13, the boundary string operators are shown in Fig. 4(c) and 4(d). These boundary strings are products of primary boundary gauge operators.

## 2.3 Lattice construction of boundaries and defects

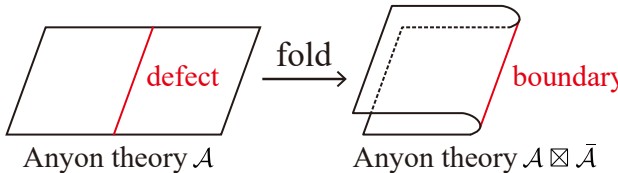

**Fig. 5** The defect in an anyon theory $\mathcal{A}$ can be interpreted as the boundary of the folded anyon theory $\mathcal{A} \boxtimes \bar{\mathcal{A}}$.

As illustrated by the folding argument in Fig. 5, the defect (a local modification of the Hamiltonian) in the two-dimensional code $\mathcal{A}$ can be interpreted as equivalent to the boundary of the "doubled" code $\mathcal{A} \boxtimes \bar{\mathcal{A}}$, where $\bar{\mathcal{A}}$ represents the anyon theory with the same anyons as $\mathcal{A}$, but with all topological spins inverted, i.e., $\theta(\bar{a}) = \theta(a)^{-1}$. For example, defects in the $\mathbb{Z}_2$ toric code are equivalent to the boundaries of $\mathbb{Z}_2 \times \mathbb{Z}_2$ toric codes. This equivalence suggests that studying boundary constructions of topological Pauli stabilizer codes is sufficient for understanding the properties of defects.

It is well-known that each gapped boundary of an Abelian topological order (Abelian anyon theory $\mathcal{A}$) corresponds to a **Lagrangian subgroup** $\mathcal{L} \subset \mathcal{A}$, which is a maximal set of bosons with trivial mutual braiding with each other [34, 35, 38, 41]. For any anyon $a \in \mathcal{A}$ not contained in $\mathcal{L}$, there exists at least one boson $b \in \mathcal{L}$ such that it braids nontrivially with $a$, i.e., $B(a, b) \neq 1$. The physical intuition is that these bosons in $\mathcal{L}$ can condense on the boundary, allowing their string operators to terminate on the boundary without incurring any energy cost.

More precisely, every Abelian anyon theory can be described by a $\mathrm{U}(1)^N$ Chern-Simons theory [96–98], where the boundary hosts a conformal field theory with a certain number of left-moving and right-moving modes. These boundary modes are said to be "gapped" if it is possible to add perturbations to the edge theory that make these modes massive. The key idea is that we can gap the theory by introducing additional terms to the edge. These terms correspond to the string operators of bosons in the Lagrangian subgroup $\mathcal{L}$, truncated at the boundary. Adding these terms to the boundary condenses the bosons (as described in Sec. 2.3.1), and consequently, anyons that braid nontrivially with the bosons in $\mathcal{L}$ are confined—i.e., they become immobile. Thus, the ground state on the boundary contains superpositions of bosons from $\mathcal{L}$, but anyons that braid nontrivially with $\mathcal{L}$ (which, by definition of a Lagrangian subgroup, includes all remaining anyons) are confined and cannot move freely. This confinement effectively gaps the chiral boundary modes. For further details, we refer to Ref. [99].

In the stabilizer formalism (see Theorem 9), the gapless boundary modes arise because the short truncated string operators, which create anyons on the boundary, commute with all bulk stabilizers. This allows anyons to move freely on the boundary. However, by condensing a Lagrangian subgroup and confining the rest of the anyons, there are no longer any mobile anyons, and the boundary modes become gapped. However, the explicit lattice procedure for achieving this condensation is not straightforward. It requires two steps:

1. **Boundary anyon condensation**: Identify mutually commuting short boundary string operators for bosons in the Lagrangian subgroup and include them in the Hamiltonian.
2. **Topological order (TO) completion**: Add local boundary gauge operators into the Hamiltonian to ensure that the topological order condition near the boundary is satisfied.

Both steps are nontrivial and require careful computation. In this work, we will establish a theorem demonstrating how boson condensation near the boundary is achieved on the lattice, and we will provide a computational algorithm to construct the boundary for any given topological Pauli stabilizer code.

### 2.3.1 Boundary anyon condensation

We first describe the procedure for performing boson condensation on the lattice. The physical intuition behind condensing a boson $b$ on the boundary is that the boson proliferates on the boundary, and the ground state becomes a superposition of all possible configurations of the boson. In other words, when

the boson string operator is applied to create or move the boson $b$, the ground state remains unchanged. Consequently, as discussed in Refs. [25, 78, 79, 100], condensing the boson $b$ is equivalent to including the mutual-commuting short string operators of the boson into the Hamiltonian:[7]

**Theorem 9. (Boson condensation on the boundary)** *To condense the boundary bosons $\{b_i\} \in \mathcal{L}$, we introduce mutually commuting short boundary string operators corresponding to each $b_i$ into the Hamiltonian, which initially consists of bulk stabilizers. As a result, the bulk string operator of any anyon $b \in \mathcal{L}$ can terminate on the boundary without causing any energy excitations. In contrast, if the bulk string operator of an anyon $a \notin \mathcal{L}$ terminates on the boundary, it will induce energy excitations, as illustrated in the figure below:*

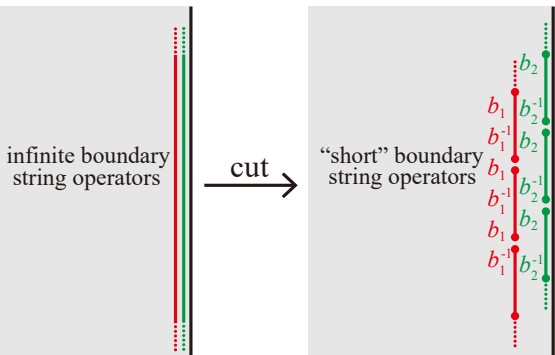

*Proof.* See the proof following Lemma 17. □

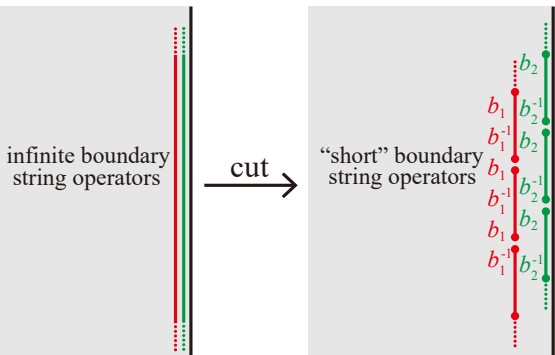

**Fig. 6** To find the mutually commuting short boundary strings of bosons in the Lagrangian subgroup $\mathcal{L}$, we begin by constructing their infinite strings, as described in Corollary 2.1. We then cut these infinite strings into finite segments, ensuring that these segments are long enough so that the commutation relations of the string operators on them match the topological spins given in Theorem 5. Additionally, it is crucial that the cutting point of each string $b_i$ is fully contained within the interval of the finite segments of another string $b_j$, ensuring that the commutation relations between string operators of different bosons satisfy the braiding statistics described in Theorem 6. These finite string operators form the mutually commuting "short" boundary strings of bosons in $\mathcal{L}$.

By Theorem 9, the remaining task for boson condensation is to identify short string operators for the bosons $\{b_i\}$ that commute with each other. However, the condition that the topological spin $\theta(b) = 1$, as given in Eq. (10), only ensures that two strings of $b$ with sufficient length commute. This does not necessarily imply that the shortest string operators will also commute. To address this, we propose a systematic method for constructing mutually commuting "short" string operators. These operators are of finite length, though they may not be the shortest possible string operators. Given a Lagrangian subgroup $\mathcal{L}$, we first identify the infinite boundary string operators corresponding to each boson $b \in \mathcal{L}$ (see the discussion following Theorem 1 and Lemma 2). We then truncate these infinite boundary strings into finite boundary string operators, as illustrated in Fig. 6. The resulting finite boundary string operators are sufficiently long to guarantee mutual commutation. This construction is always feasible because the commutation relations depend only on the finite region around where the operators intersect. This approach ensures that the finite boundary string operators for the bosons $\{b_i\}$ commute, thereby achieving the boson condensation on the boundary.

---

[7]Originally, an additional step is needed to eliminate stabilizer terms that do not commute with the short string operators. However, in our case, the boundary string operators are defined to commute with all bulk stabilizers, which allows us to bypass this step.

### 2.3.2 Topological order completion

This section outlines the final step in constructing a gapped boundary for a topological Pauli stabilizer code.

**Theorem 10. (Topological order completion)** *To ensure that the system satisfies the topological order condition, additional boundary gauge operators that commute with themselves and the existing short boundary string operators of bosons $b \in \mathcal{L}$ must be added into the Hamiltonian. While several valid choices exist for these additional boundary gauge operators, the specific selection does not affect the condensation properties described in Theorem 9.*

*Proof.* See the proof following Lemma 17. □

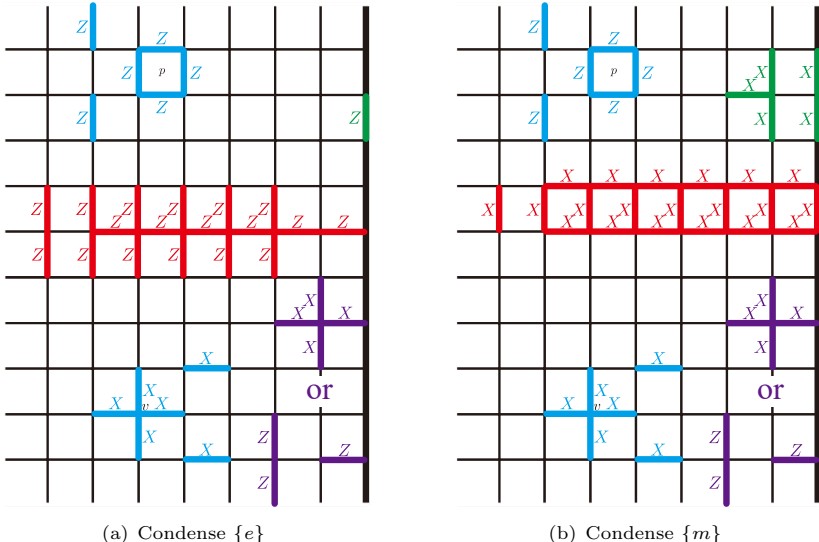

(a) Condense $\{e\}$        (b) Condense $\{m\}$

**Fig. 7** Boundary constructions of the $\mathbb{Z}_2$ fish toric code. The Hamiltonian consists of blue terms representing the bulk stabilizers, green terms corresponding to the short string boundary operators of the $e$ anyon (in (a)) and $m$ anyon (in (b)), and purple terms indicating the operators added for topological order completion. There are two choices for these purple terms, and either one can be selected. All terms have their translational counterparts: bulk stabilizer terms can move in both the $x$ and $y$ directions, while the boundary terms can only move in the $y$ direction. The red operators represent bulk strings that can terminate on the boundary without causing any energy excitation.

In other words, we can add more boundary gauge operators to the Hamiltonian until it "saturates"—meaning no additional boundary gauge operators can be incorporated while commuting with all existing terms. The detailed computational algorithm for incorporating these additional boundary gauge operators into the Hamiltonian will be discussed in Sec. 4. This process, named as topological order completion, is crucial due to the presence of secondary boundary gauge operators, which have been often overlooked in previous literature. For example, the $e$-condensed and $m$-condensed boundaries of the $\mathbb{Z}_2$ fish toric code are illustrated in Fig. 7. It is important to note that the shape of the boundary does not determine its condensation properties. In the $\mathbb{Z}_2$ standard toric code, there is a common assumption that a smooth boundary is $m$-condensed and a rough boundary is $e$-condensed. However, whether a boundary is $m$-condensed or $e$-condensed depends on which short boundary string operators are included in the Hamiltonian. Thus, a smooth boundary can represent either type.

## 3 Operator algebra formalism

In this section, we introduce the operator algebra formalism for generalized Pauli matrices on a (truncated) lattice. This approach is essential for providing a rigorous mathematical foundation for the concepts discussed earlier, allowing us to prove the lemmas and theorems formally. However, for readers more interested in the practical aspects of applying the computational algorithm to construct boundaries and defects in generalized Pauli stabilizer codes, it is possible to proceed directly to Sec. 4 without losing continuity.

The approach in this section is inspired by the symplectic and quasi-symplectic formalisms developed in Refs. [60, 72]. However, we do not presuppose translational symmetry imposed by the Laurent polynomial ring. This formalism provides more flexibility for handling the boundary of a system.

## 3.1 Symplectic Abelian groups

Given any (finite, infinite, or truncated) lattice, label its sites with an index set $I$. Each $i \in I$ refers to a site where a qudit is situated. The clock and shift operators ($X$ and $Z$ defined in Eq. (1)) on each qudit are represented respectively by $\begin{pmatrix} 1 \\ 0 \end{pmatrix}$ and $\begin{pmatrix} 0 \\ 1 \end{pmatrix}$. Products of clocks and shifts operators form the group of generalized Pauli operators. Each generalized Pauli operator has a corresponding sum of $\begin{pmatrix} 1 \\ 0 \end{pmatrix}$ and $\begin{pmatrix} 0 \\ 1 \end{pmatrix}$, which is only unique up to a phase. For example, both $X^2 Z$ and $XZX$ are represented by $\begin{pmatrix} 2 \\ 1 \end{pmatrix}$. As $X^d = Z^d = I$, we have $\begin{pmatrix} d \\ 0 \end{pmatrix} = \begin{pmatrix} 0 \\ d \end{pmatrix} = \begin{pmatrix} 0 \\ 0 \end{pmatrix}$. All calculations shall therefore be done modulo $d$, where $d$ is the dimension of the qudit. We denote the set of all length-2 column vectors modulo $d$ by $\mathbb{Z}_d^2$.

The commutation relation between two operators is recovered by the matrix $\begin{pmatrix} 0 & 1 \\ -1 & 0 \end{pmatrix}$: two Pauli operators $\begin{pmatrix} a \\ b \end{pmatrix}$ and $\begin{pmatrix} c \\ d \end{pmatrix}$ commute up to a phase $\exp\left(2\pi i \frac{m}{d}\right)$ with

$$m = \begin{pmatrix} a & b \end{pmatrix} \begin{pmatrix} 0 & 1 \\ -1 & 0 \end{pmatrix} \begin{pmatrix} c \\ d \end{pmatrix}. \tag{14}$$

This is also called the standard symplectic form

$$\omega : \mathbb{Z}_d^2 \times \mathbb{Z}_d^2 \to \mathbb{Z}_d. \tag{15}$$

On a lattice, generalized Pauli operators come in two flavors: ones with finite support and ones with (possibly) infinite support. They are respectively represented by Abelian groups $P = \bigoplus_I \mathbb{Z}_d^2$ and $\hat{P} = \prod_I \mathbb{Z}_d^2$.[8] An element $a$ in either consist of vectors $a_i \in \mathbb{Z}_d^2$ indexed by $I$, but only the latter allows infinitely many nonzero vectors. For example, the elements $\mathcal{X}$ and $\mathcal{Z}$ with $\mathcal{X}_i = \begin{pmatrix} 1 \\ 0 \end{pmatrix}$ and $\mathcal{Z}_i = \begin{pmatrix} 0 \\ 1 \end{pmatrix}$ for all $i \in I$ only exists in $\hat{P}$ when $I$ is infinite. They represent the tensor product of $X$ and $Z$ operators on all qudits, respectively. Commutation relation $\omega$ naturally extends to

$$\Omega : P \times P \to \mathbb{Z}_d, \tag{16}$$

and

$$\hat{\Omega} : \hat{P} \times P \to \mathbb{Z}_d, \tag{17}$$

by $\sum_{i \in I} \omega_i$ modulo $d$. However, the commutation relation between two infinite operators is not well-defined, and there is therefore no extension of $\omega$ to $\hat{P} \times \hat{P} \to \mathbb{Z}_d$. For example, the commutation relation between $\mathcal{X}$ and $\mathcal{Z}$ as defined above is ill-defined in general.

For any subgroup $A \subset P$, define

$$A^\Omega = \{c \in P : \Omega(c, a) = 0 \text{ for all } a \in A\}. \tag{18}$$

A Pauli stabilizer code is determined by an isotropic subgroup $S \subset P$, meaning $\Omega(s_1, s_2) = 0$ for any $s_1, s_2 \in S$. Equivalently, this can be expressed as $S \subset S^\Omega$. A code satisfies the topological order condition if it has no local logical operator. This condition is met if $S$ is also coisotropic, meaning $S^\Omega \subset S$. Therefore, a topological Pauli stabilizer code satisfies $S^\Omega = S$. We call a subgroup $A$ of $P$ **closed** if $A^{\Omega\Omega} = A$. There are two important examples of closed subgroups. Firstly, for $I$ finite, any subgroup $A$ is closed. Secondly, a stabilizer code satisfying the topological order condition is closed as implied by the following lemma, bearing in mind that $S^\Omega = S$.

**Lemma 11.** *Given a subgroup $A$ of $P$ , $A^{\Omega\Omega\Omega} = A^\Omega$. In other words, $A^\Omega$ is closed.*

*Proof.* From the definition, $A^\Omega \subset A^{\Omega\Omega\Omega}$. Conversely, let $c \in A^{\Omega\Omega\Omega}$. Any $a \in A$ is also in $A^{\Omega\Omega}$. Thus $\Omega(a, c) = 0$. This implies that $c \in A^\Omega$. $\square$

We make another definition whose use will be clear later. Given $M \subset P$ and $N \subset \hat{P}$, define

$$M^\perp = \{\hat{c} \in \hat{P} : \hat{\Omega}(\hat{c}, m) = 0 \text{ for all } m \in M\}$$

---

[8] $P = \hat{P}$ when $I$ is finite.

and
$$N^\perp = \{c \in P : \hat{\Omega}(n, c) = 0 \text{ for all } n \in N\}.$$
Notice that $(\cdot)^\perp$ alternates between $P$ and $\hat{P}$.

Sometimes, we focus our attention on a subsystem within a system. For example, when studying boundaries, we focus on one half of the system. Given a group of Pauli operators $P = \bigoplus_I \mathbb{Z}_d^2$, a truncation is a group homomorphism $\pi : P \to P$ satisfying:

1. (Projection) $\pi \circ \pi = \pi$ and
2. (Orthogonality) $\Omega(\ker \pi, \pi P) = 0$.

For example, given a subset $J \subset I$ of qudits, projection $\pi_J$ onto $\pi P := \bigoplus_J \mathbb{Z}_d^2$ is a truncation. Orthogonality simply states that Pauli operators on qudits in $J$ and qudits in $I \setminus J$ commute with each other.

The following lemma is important for showing bulk-boundary correspondence.

**Lemma 12.** *Given a subgroup $A \subset \hat{P}$, $A^{\perp\perp} = \hat{A}$, where $\hat{A} \subset \mathring{P}$ contains both finite and infinite products generated by elements in $A$.*[9]

*Proof.* We prove this lemma in two steps: $\hat{A} \subset A^{\perp\perp}$ and $\hat{A} \supset A^{\perp\perp}$.

We first show that $\hat{A} \subset A^{\perp\perp}$. Let $\hat{a} \in \hat{A}$. We want to show that $\hat{\Omega}(\hat{a}, c) = 0$ for any $c \in A^\perp \subset P$. Consider the decomposition:

$$\hat{a} = \prod_{\text{supp}(a_\alpha) \cap \text{supp}(c) \neq \emptyset} a_\alpha \cdot \prod_{\text{supp}(a_\beta) \cap \text{supp}(c) = \emptyset} a_\beta, \tag{19}$$

where $a_\alpha, a_\beta \in A$, and the support (supp) refers to the sites where a Pauli operator is non-identity. Note that the product $b := \prod_{\text{supp}(a_\alpha) \cap \text{supp}(c) \neq \emptyset} a_\alpha$ is in $A$ since $c \in P$ has finite support. The other term $\prod_{\text{supp}(a_\beta) \cap \text{supp}(c) = \emptyset} a_\beta$ may have infinite support; however, it does not overlap with $c$ and commutes with $c$. Thus, we find that $\hat{\Omega}(\hat{a}, c) = \Omega(b, c) = 0$, where the final equality follows from the fact that $b \in A$ and $c \in A^\perp$. Therefore, we conclude that $\hat{A} \subset A^{\perp\perp}$.

Next, we prove that $A^{\perp\perp} \subset \hat{A}$. For any $a \in A^{\perp\perp}$, we need to show $a \in \hat{A}$. Specifically, we want to demonstrate that any finite part of $a$ comes from an element of $A$, implying that $a$ can be expressed as, at most, an infinite product of elements from $A$. To do so, it suffices to show that for any finite region $\Gamma \subset I$, there exists $a_\Gamma \in A$ such that $a - a_\Gamma$ is supported outside $\Gamma$. Note that $a_\Gamma$ is not necessarily supported entirely within $\Gamma$; typically, $a_\Gamma$ is chosen to extend slightly beyond $\Gamma$.

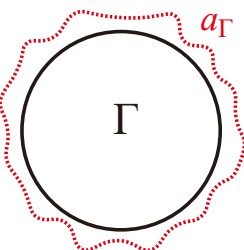

Indeed, consider a sequence of nested finite regions $\{\Gamma_i\}$ with $\Gamma_i \subset \Gamma_{i+1}$ and $\bigcup_i \Gamma_i = I$. In each region $\Gamma_i$, the element $a$ is approximated by some $a_{\Gamma_i} \in A$, with any error confined to the outside of $\Gamma_i$. By the definition of the limit, the sequence $\{a_{\Gamma_i}\}$ converges to $a$. At each step, $a_{\Gamma_i}$ is a product of stabilizers in $A$; therefore, the limit, $a$, is an (infinite) product of stabilizers in $A$, which implies that $a$ is an element of $\hat{A}$.

Thus, we now show that such a $a_\Gamma$ always exists for any $\Gamma$. For a finite region $\Gamma$, denote Pauli operators supported in $\Gamma$ by $P_\Gamma$. Let $A_\Gamma = \pi_\Gamma A$ be image of $A$ under truncation to $\Gamma$. Clearly, $A_\Gamma$ is a closed subgroup of $P_\Gamma$. Since $a \in A^{\perp\perp}$, by definition, $a$ commutes with every element in $A^\perp$. When we project $a$ onto the finite region $\Gamma$, denoted by $\pi_\Gamma a$, this projection will still commute with the elements of $A^\perp$ that are fully supported in $\Gamma$. These elements form the subspace $A_\Gamma^\perp$, where $(\cdot)^\perp$ is taken inside the finite group $P_\Gamma$. By the commutation property, we have $\pi_\Gamma a \in A_\Gamma^{\perp\perp}$, where $A_\Gamma^{\perp\perp}$ is the double orthogonal of $A_\Gamma$ within $P_\Gamma$. In the context of the finite-dimensional space $P_\Gamma$, we know that $A_\Gamma^{\perp\perp}$ coincides with $A_\Gamma$ itself. Therefore, we have shown that $\pi_\Gamma a \in A_\Gamma$. By definition, there exists $a_\Gamma \in A$ such that $\pi_\Gamma a_\Gamma = \pi_\Gamma a$ and $a - a_\Gamma$ is supported outside $\Gamma$. $\square$

**Remark 1.** *The approximation process outlined in the proof gives rise to a useful natural topology on $\hat{P}$. See Ref. [101] for a detailed account and other applications.*

---

[9]For an infinite product, the Pauli matrix at each site must appear finitely many times for the product to be well-defined.

## 3.2 Stabilizer codes and boundary gauge operators

For a stabilizer code $S \subset P$, truncation $\pi$ gives rise to several new objects. $S_B := S \cap \pi P$ denotes the bulk stabilizers. $S_T := \pi S$ denotes the truncated stabilizers (including the bulk stabilizers). $G := S_B^\Omega \cap \pi P$ denotes all operators on the truncated space that commute with the bulk stabilizers. Clearly, $S_B \subset S_T \subset G$. We observe the following:

1. $S_T / S_B$ are referred to as the **primary boundary gauge operators**.
2. $G / S_T$ are referred to as the **secondary boundary gauge operators**.
3. $\mathcal{G} = G / S_B$ represents the **boundary gauge operators**, determined up to bulk stabilizers.

The group $\mathcal{G}$ is often called the **gauge group**.

**Remark 2.** *Note $G = S_B^\Omega \cap \pi P$ is more compactly known as $S_B^\Omega$ treating $S_B$ as a subgroup of $\pi P$ instead of $P$. We also use this notation when there is no ambiguity.*

**Lemma 13.** *Given $a \in P, b \in \pi P$, $\Omega(a, b) = \Omega(\pi a, b)$.*

*Proof.* Using bilinearity, $\Omega(a, b) = \Omega(\pi a, b) + \Omega(a - \pi a, b)$. The latter term vanishes because of orthogonality and projecion. $\square$

The next theorem says that, for a topologically ordered stabilizer code, the set of all operators that commute with all the truncated stabilizers is the same as the set of all bulk stabilizers.

**Theorem 14.** *If $S \subset P$ is isotropic and coisotropic (i.e., satisfies the topological order condition), then $S_B = S_T^\Omega \cap \pi P$, or more compactly (see Remark 2), $S_B = S_T^\Omega$.*

*Proof.* The proof contains two directions:

1. ($\subset$): Given $b \in S_B = S \cap \pi P$, $\Omega(l, b) = 0$ for all $l \in S$ (using isotropy). By the lemma above, $\Omega(\pi l, b) = \Omega(l, b) = 0$ for all $l \in S$, which is equivalent to $\Omega(l, b) = 0$ for all $l \in S_T = \pi S$. Therefore, $b \in S_T^\Omega \cap \pi P$
2. ($\supset$): Given $b \in S_T^\Omega \cap \pi P$, we again have $\Omega(l, b) = \Omega(\pi l, b)$ for all $l \in S$ from the lemma above. As $b \in S_T^\Omega$, $\Omega(\pi l, b) = 0$ for all $l \in S$. In other words, $b \in S^\Omega \subset S$ (using coisotropy). Since $b$ is supported in $\pi P$, we have $b \in S_B$. $\square$

**Corollary 14.1.** *A local Pauli operator on the truncated system that does not violate bulk stabilizers or primary boundary terms must not violate secondary boundary terms either.*

*Proof.* An operator that does not violate primary boundary terms is by definition in $S_T^\Omega \cap \pi P$. By the theorem above, it is in $S_B$, which by definition commutes with everything in $G = S_B^\Omega \cap \pi P$. $\square$

Together with the weak translational symmetry of boundary anyons demonstrated in Corollary 2.1, the above corollary leads to the proof of Theorem 7, which asserts that the syndrome pattern of primary boundary gauge operators is sufficient to uniquely identify an anyon. Consider two finite boundary strings that share the same primary boundary syndrome at their endpoints. Without loss of generality, we can assume they have the same length by finding the least common multiple of their weak translational symmetry. Each string exhibits a syndrome pattern at one end and the opposite pattern at the other. The difference between these two strings does not violate any primary boundary gauge operators, and, as a result, does not violate any secondary boundary gauge operators according to Corollary 14.1. Therefore, only the syndrome pattern of primary boundary gauge operators is enough to describe a boundary anyon.

With the theorems, corollaries, and mathematical framework established above, we can rigorously derive the statements presented in Sec. 2.

*Proof of Theorem 1.* The proof proceeds in two parts:

1. **Creating multiple boundary anyons via a local boundary gauge operator**: We begin by demonstrating that a superposition of boundary anyons $\varphi$ at distinct locations, where the greatest common divisor (gcd) of the anyon multiplicities at each location and $d$ is 1, can be generated by a local boundary gauge operator $O_\partial$.
2. **Constructing an infinite boundary string operator that creates a single boundary anyon**: Given the boundary gauge operator $O_\partial$, we show that it can be applied repeatedly at different locations to construct an infinite operator

$$O_\varphi = \prod_i O_\partial(l_i),$$

where $O_\partial(l_i)$ represents the operator $O_\partial$ translated by $l_i$ in the $y$-direction, and this operator $O_\varphi$ generates a single boundary anyon $\varphi$.

We begin by proving the first part. We claim that for any homomorphism $\varphi : \mathcal{G} \to U(1)$, there exists a local boundary gauge operator whose syndrome pattern equals a superposition of $\varphi$ and its translated copies along

the boundary. In other words, though $\varphi$ may not be creatable by a local boundary gauge operator, several copies of it located at different positions can be. This step follows directly from the following theorem:

**Theorem 15. (Proposition 10 of Ref. [72])** *Let $M$ be a quasi-symplectic $R$-module equipped with a $\mathbb{Z}_d$-bilinear pairing $\Omega : M \times M \to R$. Then $M^*/M$ is a torsion module.*

Let us now explain this theorem in detail. We begin by setting $R = \mathbb{Z}_d[y, y^{-1}]$, the Laurent polynomial ring reviewed in Sec. 4.1, where $y$ represents the translation in the $y$-direction. The key condition for a quasi-symplectic form is that $\Omega(m', m) = 0$ for all $m' \in M$ implies $m = 0$. This ensures that $\Omega$ is a non-degenerate bilinear form. In other words, The map $M \to M^*$ given by $m \mapsto \Omega(-, m)$ is injective.[10] Theorem 15 says that the quotient $M^*/M$ is a torsion module, meaning that for any $m^* \in M^*$, there exists an element $r \in R$ (where $r$ is not a zero divisor[11]) such that $rm^* = \Omega(-, m)$ for some $m \in M$. We now apply this theorem to our specific scenario. By Lemma 12, the module $\mathcal{G}$ fits the definition of a quasi-symplectic module. Specifically, if a local operator commutes with all boundary gauge operators, then it must be a bulk stabilizer, indicating that $\mathcal{G}$ has a non-degenerate bilinear form $\Omega$. Therefore, by the theorem, $\mathcal{G}^*/\mathcal{G}$ is a torsion module. This implies that multiple copies of a boundary syndrome pattern at different locations, indicated by $r \in R$, sum to a trivial syndrome pattern generated by a local boundary operator in $\mathcal{G}$.

Next, we prove the second part. Since there is a local boundary gauge operator $O_\partial$ creating a superposition of multiple $\varphi$ syndromes along a one-dimensional boundary, we can isolate a single $\varphi$ syndrome by repeatedly applying translations of the local operator. For example, consider a special case where the initial boundary anyons are located at positions $y_1, y_2, y_3, \ldots$ such that $y_1 < y_2 \le y_3 \le y_4 \le \cdots$. By applying a translated operator, we create new boundary anyons at positions $y_2, y_2 + (y_2 - y_1), y_3 + (y_2 - y_1), \cdots$. When subtracting the boundary anyons, the anyon at $y_1$ remains, but the anyon at $y_2$ is canceled. The next remaining anyon, $y_2'$, defined as the first anyon to the right of $y_1$, is now either at $y_3$ or at $y_2 + (y_2 - y_1)$. This location is either farther from $y_1$, or the multiplicity of boundary anyons at $y_2$ decreases. By repeating this process, the second anyon adjacent to $y_1$ can be pushed to infinity, effectively trivializing the contribution of the remaining boundary anyons.

For the general case where $y_1$ might be equal to $y_2$, we need to prove that if $r$ is not a zero divisor, then there exists an inverse $I$ such that $rI = 1 \in R$. In Appendix B, we have shown that an element in the formal Laurent series has an inverse if and only if the greatest common divisor of its coefficients is 1. Therefore, since $r$ is not a zero divisor, it satisfies this condition, ensuring the existence of an inverse. This completes the proof of Theorem 1. $\qquad\square$

*Proof of Lemma 3.* For a boundary anyon, there exists an infinite boundary string operator $s$ that commutes with both bulk stabilizers and boundary gauge operators. This is because Theorem 1 shows that a boundary anyon can be created by a semi-infinite boundary gauge operator (see Definition 3), and this string can be extended to infinity without violating any boundary gauge operator by Corollary 2.1.

Since $s$ commutes with both bulk stabilizers and boundary gauge operators, it commutes with all elements of $G$, meaning $s \in G^\perp$. Given that $G = S_B^\Omega \cap \pi P$ or equivalently $G = \tilde{S}_B^\perp$, where $\tilde{S}_B$ is $S_B$ viewed as a subset of $\pi\hat{P}$, we have $G^\perp = \tilde{S}_B^{\perp\perp}$. By Lemma 12, $\tilde{S}_B^{\perp\perp} = \hat{S}_B$, so $s \in \hat{S}_B$. In other words, $s$ is an (infinite) product of bulk stabilizers:

$$s = \prod_{i \in J} S_i, \tag{20}$$

where $J$ is an infinite set of bulk stabilizers.

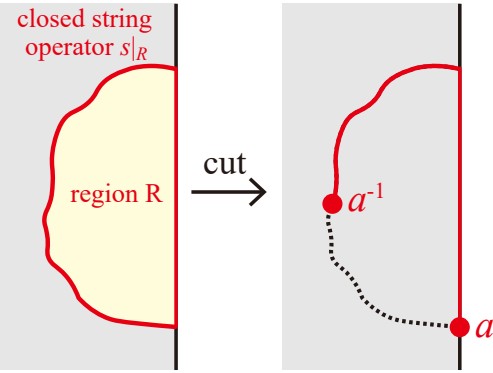

**Fig. 8** We first modify the infinite string operator $s$ into a closed string operator $s|_R$ by selecting a subset of bulk stabilizers within the region $R$, as defined in Eq. (21). Then, we cut this closed string operator to form an open string operator, resulting in a bulk anyon and a boundary anyon at its endpoints.

---

[10] The module $M^*$ is defined as the set of all $R$-linear homomorphisms from $M$ to $R$, i.e., $M^* = \mathrm{Hom}(M, R)$.
[11] This requires that the coefficients of the polynomial $r$ have a greatest common divisor of $1 \in \mathbb{Z}_d$.

Next, we choose a finite subset $J_R \subset J$, containing all stabilizers from $J$ that are fully supported within a large local region $R$, as shown in Fig. 8. We define a new operator

$$s|_R := \prod_{i \in J_R} S_i. \tag{21}$$

Deep inside $R$ (far from $\partial R$, the boundary of $R$, with a distance much larger than the range of each bulk stabilizer), the operator behaves like $s$, which vanishes. Far outside $R$, no operator is applied, so it also vanishes there. Thus, $s|_R$ is supported only near $\partial R$. Near the system's boundary, $s$ and $s|_R$ are identical. This $s|_R$ represents the closed string operator version of the infinite string operator $s$. Finally, we cut the closed string into an open string operator, as shown in Fig. 8. One endpoint of this open string corresponds to a bulk anyon, while the other endpoint corresponds to a boundary anyon. This demonstrates that any boundary anyon can be moved into the bulk.

$\square$

*Proof of Theorem 4.* By Lemma 3, we can move a boundary anyon into the bulk; therefore, it is sufficient to show that the inverse process exists.

For a bulk anyon, create a string with one anyon at each endpoint. We assume two endpoints fall onto two sides of the boundary. Truncate this string to create a string lying fully on one side of the boundary. The truncation creates a boundary anyon while the bulk anyon at the other end remains intact. This operator moves this bulk anyon into the boundary. $\square$

To further establish Theorem 8, we need a lemma concerning bulk Pauli stabilizer codes.

**Lemma 16.** *Given a bulk Pauli stabilizer code $S \subset P$ satisfying the topological order condition $S^\Omega = S$, a closed anyon string operator is equal to a finite product of stabilizers.*

*Proof.* Such a closed string operator commutes with all bulk stabilizers. Therefore, it is contained in $S^\Omega = S$. $\square$

*Proof of Theorem 8.* Embed the boundary string operator into the complete bulk system. It commutes with all bulk stabilizers except at the two endpoints. The boundary string becomes a bulk string, creating two bulk anyons at its endpoints. These anyons are mobile [60, 72] and can be moved out of the truncated system and annihilate each other. The resulting closed string operator can be expressed as a product of bulk stabilizers by Lemma 16. The truncation of this bulk closed string operator gives a boundary string operator equivalent to the boundary string we start with. Moreover, this boundary string consists solely of primary boundary gauge operators.

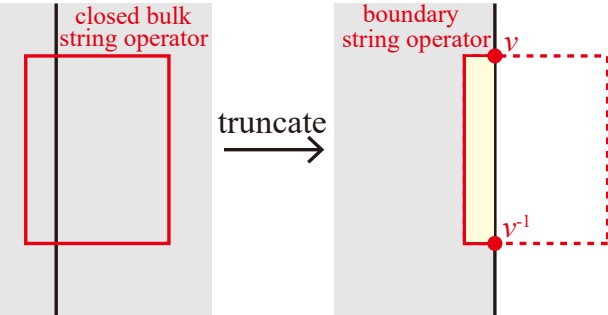

$\square$

Now, we will prove Theorem 9 and Theorem 10. Before proceeding with these proofs, we note the following property, which generalizes the result of Lemma 16:

**Lemma 17.** *Consider the semi-infinite bulk and boundary string operators, as shown in Fig. 9. This combined operator $O$ can be represented as an infinite product of bulk stabilizers.*

*Proof.* Let $\hat{S}$ be the group consisting of infinite products of bulk stabilizers. The orthogonal complement of this group, $\hat{S}^\perp$, consists of all finite products of bulk stabilizers and boundary gauge operators.

Now, consider the infinite string operator $O$, which is composed of a bulk string operator and a boundary string operator, as illustrated in Fig. 9. The construction of this operator $O$ is designed to commute with all bulk stabilizers and boundary gauge operators.

Since $O$ commutes with all elements in $\hat{S}^\perp$ (finite products of bulk stabilizers and boundary gauge operators), it follows by definition that $O \in \hat{S}^{\perp\perp}$. By Lemma 12, we know that $\hat{S}^{\perp\perp} = \hat{S}$. Therefore, $O \in \hat{S}$, which means that $O$ can be expressed as an infinite product of bulk stabilizers. This completes the proof of the lemma. $\square$

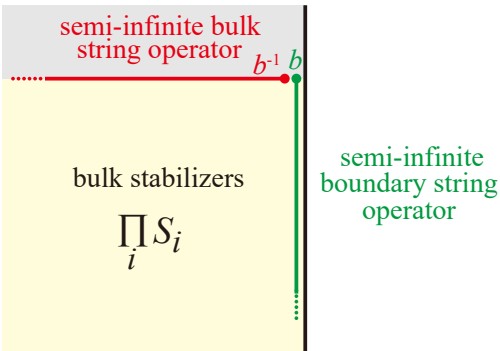

**Fig. 9** According to Lemma 3, a boundary anyon can move into the bulk, allowing the bulk and boundary string operators to merge without violating any bulk stabilizers or boundary gauge operators.

Using Lemma 17, we can establish Theorems 9 and 10.

*Proof of Theorem 9 and 10.*

First, we demonstrate that the bulk string of anyon $b$ can terminate on the boundary without causing any energy excitations. As shown in Fig. 9, we can multiply short boundary string operators corresponding to the boundary anyon $b$ to form a semi-infinite boundary string operator, which can then be attached to a semi-infinite bulk string operator. Since these short boundary string operators are part of the Hamiltonian, they do not contribute to any energy violations associated with the original bulk string operator. By Lemma 17, the combined operator, consisting of the semi-infinite bulk and boundary string operators, can be expressed as an infinite product of bulk stabilizers. Consequently, this combined operator does not violate any bulk stabilizers or boundary gauge operators, including those associated with boson condensation and topological order completion on the boundary. Therefore, the bulk string of anyon $b$ can terminate on the boundary without incurring any energy cost.

Next, we show that if $a \notin \mathcal{L}$, its string ending on the boundary will violate certain boundary terms in the Hamiltonian. According to the bulk-boundary correspondence (Theorem 4), the bulk string of anyon $a$ can be bent into a boundary string, as depicted in Fig. 10. We then consider a boundary string of anyon $b$ that has a long enough overlap with the boundary string of $a$. Since $a \notin \mathcal{L}$, we can choose a $b \in \mathcal{L}$ such that the braiding between $a$ and $b$ is nontrivial, meaning $B(a, b) \neq 1$. By Theorem 6, this nontrivial braiding implies that the commutator of the two boundary strings of $a$ and $b$ is nontrivial, indicating that the string operators do not commute. Given that the boundary string of $b$ can be constructed as a product of short string operators in the Hamiltonian, the presence of the non-commuting string operators means that the bulk string of $a$ must violate at least one term in the Hamiltonian. Thus, we have shown that if $a \notin \mathcal{L}$, its string ending on the boundary results in a violation of certain boundary terms, completing the proof. $\square$

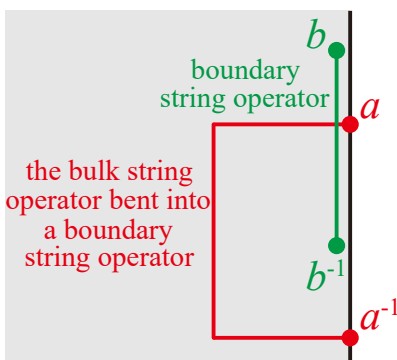

**Fig. 10** A bulk string operator for anyon $a \notin \mathcal{L}$ can terminate on the boundary, creating two boundary anyons, as illustrated by the red string. The green string represents a boundary string operator for anyon $b \in \mathcal{L}$, formed by a product of many short boundary string operators in the boundary Hamiltonian. Both strings must be long relative to the size of the boundary anyons to ensure that their commutation corresponds to the braiding between $a$ and $b$, as described in Theorem 6. By the definition of the Lagrangian subgroup, the braiding $B(a, b) \neq 1$, so the bulk string operator for anyon $a$ terminating on the boundary must violate the boundary Hamiltonian.

# 4 Computational algorithms

This section presents a computer-based algorithm for analyzing the boundary theory of a given topological Pauli stabilizer code. The algorithm addresses three aspects:

1. Determine boundary gauge operators, which generate the anomalous Hilbert space of the boundary theory.
2. Obtain boundary string operators and boundary anyons, classify these anyons using equivalence relations, and compute their fusion rules.
3. Construct gapped boundaries and defects via boundary anyon condensation and topological order completion.

Notably, the algorithm applies to qudit systems with nonprime dimensions, such as $\mathbb{Z}_4$ qudits. This section provides the detailed procedures of the algorithm; readers interested in the applications can proceed directly to Sec. 5 without delving into the technical details presented here.

Sec. 4.1 begins with reviewing the Laurent polynomial formalism along with the necessary notations and conventions used throughout this section. In Sec. 4.2, we present an algorithm that systematically derives the boundary gauge operators for a truncated Pauli stabilizer code. Sec. 4.3 focuses on obtaining boundary string operators as products of these boundary gauge operators. These boundary string operators generate boundary anyons at their endpoints, and we use equivalence relations to classify these anyons. The fusion rules between these anyons can be computed using the Smith normal form. In Sec. 4.4, we delve into the procedure of boundary anyon condensation and the topological order completion. The computation is analogous to the approach for obtaining boundary gauge operators described earlier in Sec. 4.2.

Additional details are provided in Appendix F. Appendix F.1 explains how to derive the condensed strings terminating on the boundary and the bulk strings crossing a defect line. Appendix F.2 describes the process for identifying the endpoint of a defect line when the defect line has a finite length.

## 4.1 Review of the Laurent polynomial formalism

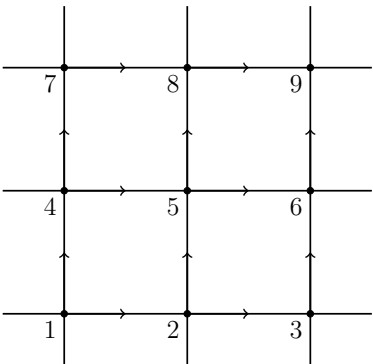

**Fig. 11** We put a qudit on each edge, with generalized Pauli operator $X_e$ and $Z_e$ acting on it.

To efficiently compute translation-invariant Pauli stabilizer models, we use the Laurent polynomial formalism, a well-established method in the study of fracton models, topological orders, bosonization, quantum cellular automata, and error-correcting codes [50, 60, 62, 74, 86, 102–105]. This section briefly reviews the Laurent polynomial formalism and its application to translation-invariant stabilizer codes, following the conventions in Ref. [50]. Readers are encouraged to consult the original references for a more comprehensive explanation.

In this section, we demonstrate the case involving two $\mathbb{Z}_d$ qudits per unit cell, such as the case where a qudit is located at each edge of a square lattice. This framework can be generalized to scenarios with $w$ qudits per unit cell. We begin by showing that any Pauli operator, defined as a finite tensor product of Pauli matrices across different lattice sites, can be expressed (up to an overall constant) as a column vector over the polynomial ring $R = \mathbb{Z}_d[x, y, x^{-1}, y^{-1}]$[12], as described in Ref. [60]. We assign column vectors over $\mathbb{Z}_d$ to the (generalized) Pauli matrices $X_{12}$, $Z_{12}$, $X_{14}$, and $Z_{14}$, depicted in Fig. 11:

$$\mathcal{X}_{12} = \begin{bmatrix} 1 \\ 0 \\ \hline 0 \\ 0 \end{bmatrix} , \ \mathcal{Z}_{12} = \begin{bmatrix} 0 \\ 0 \\ \hline 1 \\ 0 \end{bmatrix} , \ \mathcal{X}_{14} = \begin{bmatrix} 0 \\ 1 \\ \hline 0 \\ 0 \end{bmatrix} , \ \mathcal{Z}_{14} = \begin{bmatrix} 0 \\ 0 \\ \hline 0 \\ 1 \end{bmatrix} .$$

---

[12]This ring contains all Laurent polynomials in $x$, $x^{-1}$, $y$, and $y^{-1}$, with coefficients in $\mathbb{Z}_d$.

From now on, Pauli operators represented as column vectors are denoted by curly letters, where the coefficients in these vectors indicate the corresponding powers. For instance:

$$\mathcal{P} = \begin{bmatrix} i \\ j \\ k \\ l \end{bmatrix} \quad \Rightarrow \quad \mathcal{P}^m = \begin{bmatrix} mi \\ mj \\ mk \\ ml \end{bmatrix}, \quad \forall m \in \mathbb{Z}_d. \tag{22}$$

Translation of operators is represented by polynomials in $x$ and $y$, which denote shifts in the $x$- and $y$-directions, respectively. For example, translating the operator on edge $e_{12}$ to edge $e_{78}$ (with a vector $(0,2)$) or to edge $e_{58}$ (with a vector $(1,1)$) involves multiplying the column vector of the operator by $y^2$ or $xy$, respectively:

$$\mathcal{Z}_{78} = y^2 \mathcal{Z}_{12} = \begin{bmatrix} 0 \\ 0 \\ y^2 \\ 0 \end{bmatrix}, \quad \mathcal{X}_{58} = xy\mathcal{X}_{14} = \begin{bmatrix} 0 \\ xy \\ 0 \\ 0 \end{bmatrix}.$$

In general, any Pauli operator can be written as:

$$P = \eta X_{e_1}^{a_1} X_{e_2}^{a_2} \cdots X_{e_n}^{a_n} Z_{e_1'}^{b_1} Z_{e_2'}^{b_2} \cdots Z_{e_m'}^{b_m}, \tag{23}$$

where $\eta$ is a root of unity of order $2d$. After disregarding the global phase $\eta$, the corresponding column vector for this operator is a linear combination of the individual Pauli matrices, written as:

$$\mathcal{P} = a_1 \mathcal{X}_{e_1} + a_2 \mathcal{X}_{e_2} + \cdots + a_n \mathcal{X}_{e_n} + b_1 \mathcal{Z}_{e_1'} + b_2 \mathcal{Z}_{e_2'} + \cdots + b_m \mathcal{Z}_{e_m'}. \tag{24}$$

Additional examples are provided in Fig. 12. The **antipode map** is a $\mathbb{Z}_d$-linear map from $R$ to $R$ defined by

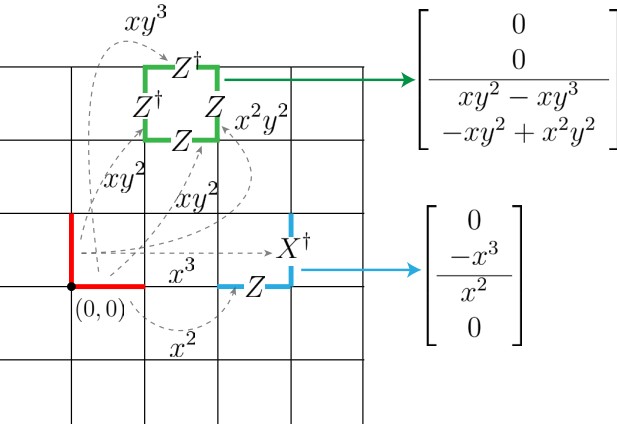

**Fig. 12** Examples of polynomial expressions for Pauli strings. The flux term on a plaquette and the $XZ$ term on edges are shown. The factors such as $x^2y^2$ and $x^2$ represent the locations of the operators relative to the origin.

$$x^a y^b \rightarrow \overline{x^a y^b} := x^{-a} y^{-b}. \tag{25}$$

To determine whether two Pauli operators represented by vectors $v_1$ and $v_2$ commute or not, we define the dot product as

$$v_1 \cdot v_2 = \overline{v}_1^T \Lambda v_2, \tag{26}$$

where $T$ is the transpose operation on a matrix and

$$\Lambda = \left[ \begin{array}{cc|cc} 0 & 0 & 1 & 0 \\ 0 & 0 & 0 & 1 \\ \hline -1 & 0 & 0 & 0 \\ 0 & -1 & 0 & 0 \end{array} \right] \tag{27}$$

is the matrix representation of the standard **symplectic bilinear form**. For simplicity, we denote $\overline{(\cdots)}^T$ as $(\cdots)^\dagger$. The constant term of a polynomial $p(x,y)$ is denoted as $\langle p(x,y)\rangle_0$. The two Pauli operators $v_1$ and $v_2$ commute if and only if $\langle v_1 \cdot v_2\rangle_0 = 0$.

A translation-invariant stabilizer code is $R$-submodule $\sigma$ such that

$$v_1 \cdot v_2 = v_1^\dagger \Lambda v_2 = 0, \quad \forall v_1, v_2 \in \sigma, \tag{28}$$

i.e., a module of commuting Pauli operators. This $\sigma$ is named the **stabilizer module**. The Hamiltonian could have $t$ terms per square to have a unique ground state on a simply connected manifold, denoted as

$$H = -\sum_{\text{cells}} (S_1 + S_2 + \cdots + S_t), \tag{29}$$

where $S_1, S_2, \cdots, S_t$ constitute the **generators** of the stabilizer module $\sigma$, and will henceforth be referred to as **stabilizer generators**.[13] For example, the trivial phase $H_0 = -\sum_e X_e$ is

$$\mathcal{S}_1 = \begin{bmatrix} 1 \\ 0 \\ 0 \\ 0 \end{bmatrix}, \quad \mathcal{S}_2 = \begin{bmatrix} 0 \\ 1 \\ 0 \\ 0 \end{bmatrix}, \tag{30}$$

and the standard $\mathbb{Z}_d$ toric code Hamiltonian

$$H_{\text{TC}} = -\sum_v \; -X^\dagger\!\!\overset{X}{\underset{X^\dagger}{\mid v \mid}}\!\!X- \; - \sum_p \; \overset{Z^\dagger}{Z^\dagger \; p \; Z} \; , \tag{31}$$

corresponds to

$$\mathcal{S}_1 = \begin{bmatrix} 1 - \overline{x} \\ 1 - \overline{y} \\ 0 \\ 0 \end{bmatrix}, \quad \mathcal{S}_2 = \begin{bmatrix} 0 \\ 0 \\ 1 - y \\ -1 + x \end{bmatrix}. \tag{32}$$

Next, we define the **excitation map** for any Pauli operator $\mathcal{P}$ on a general Pauli stabilizer code with stabilizer generators $\sigma = \langle \mathcal{S}_1, \mathcal{S}_2, \cdots, \mathcal{S}_t \rangle$ as

$$\epsilon(\mathcal{P}) := (\sigma^\dagger \Lambda \mathcal{P})^T = [\mathcal{S}_1 \cdot \mathcal{P}, \mathcal{S}_2 \cdot \mathcal{P}, \cdots, \mathcal{S}_t \cdot \mathcal{P}], \tag{33}$$

which indicates how the Pauli operator violates stabilizers $\mathcal{S}_1, \mathcal{S}_2, \cdots, \mathcal{S}_t$.

To obtain the possible anyons in this theory, we solve the **bulk anyon equation**

$$\epsilon\big(\alpha(x,y)\mathcal{X}_1 + \beta(x,y)\mathcal{X}_2 + \gamma(x,y)\mathcal{Z}_1 + \delta(x,y)\mathcal{Z}_2\big) = (1 - x^n)v, \tag{34}$$

where $n$ is an integer and $v$ is a length-$t$ row vector, referred to as an anyon. The physical interpretation of the bulk anyon equation is that when we apply Pauli matrices $X_1$, $X_2$, $Z_1$, and $Z_2$ at locations $\alpha(x,y)$, $\beta(x,y)$, $\gamma(x,y)$, and $\delta(x,y)$, it violates the stabilizers around the origin $(0,0)$ and the point $(n,0)$ with patterns $v$ and $-v$, respectively. This operator creates an anyon $v$ at $(0,0)$ and its antiparticle at $(n,0)$. Note that if $v$ is an anyon, $x^a y^b v$ is also an anyon for all $a, b \in \mathbb{Z}$.

To make this polynomial formalism manipulated by computers practically, we store the coefficients in the polynomial as a vector over $\mathbb{Z}_d$. For instance, a polynomial such as $f(x,y) = 1 + 3y - 2xy^{-1} \in R$ can be expressed as a **coefficient vector** over $\mathbb{Z}_d$

$$\begin{matrix} 1 \;\; x \;\; y \;\; \overline{x} \;\; \overline{y} \;\; x^2 \;\; xy \;\; y^2 \;\; x\overline{y} \;\; \overline{x}y \;\; \cdots \\ \widetilde{f} = \begin{bmatrix} 1 & 0 & 3 & 0 & 0 & 0 & 0 & 0 & -2 & 0 & \cdots \end{bmatrix}, \end{matrix} \tag{35}$$

---

[13]The stabilizer generators are not required to be independent from each other.

where each entry represents the coefficient of the corresponding monomial $x^a y^b$ in the polynomial $f(x, y)$. In the rest of this paper, we denote $\overset{\sim}{f}$ as the coefficient vector of the polynomial $f(x,y) \in R$. In practice, we choose the polynomial within $x^{\pm k}$ and $y^{\pm k}$. Given a fixed $k$, we define the **truncation map** as

$$f \in \mathbb{Z}_d[x, y, x^{-1}, y^{-1}] \to \overset{\sim}{f} \in \mathbb{Z}_d^{\otimes(2k+1)^2}. \tag{36}$$

The truncation size is governed by the parameter $k$, a large positive integer relative to the size of the stabilizers. We allow this truncation map to act on any $a \times b$ matrix $M$ over $\mathbb{Z}_d[x, y, x^{-1}, y^{-1}]$ by acting on each entry to expand to a row vector with length $(2k+1)^2$, and by joining these row vectors to form an $a \times b(2k+1)^2$ matrix $\widetilde{M}$ over $\mathbb{Z}_d$.

Also, we define the **translational duplicate map** $\mathrm{TD}_{m_x, m_y}$ that takes the input as a length-$l$ row vector $F = [f_1, f_2, \cdots, f_l]$ and returns a $(2m_x+1)(2m_y+1) \times l$ matrix formed by its translations within $x^{\pm m_x} y^{\pm m_y}$:

$$F \to \mathrm{TD}_{m_x, m_y}(F) := \begin{bmatrix} x^{-m_x} y^{-m_y} F \\ x^{-m_x+1} y^{-m_y} F \\ \vdots \\ x^{m_x-1} y^{-m_y} F \\ x^{m_x} y^{-m_y} F \\ \hline x^{-m_x} y^{-m_y+1} F \\ x^{-m_x+1} y^{-m_y+1} F \\ \vdots \\ x^{m_x-1} y^{-m_y+1} F \\ x^{m_x} y^{-m_y+1} F \\ \hline \vdots \\ \vdots \\ \hline x^{-m_x} y^{m_y} F \\ x^{-m_x+1} y^{m_y} F \\ \vdots \\ x^{m_x-1} y^{m_y} F \\ x^{m_x} y^{m_y} F. \end{bmatrix}, \tag{37}$$

where $x^a y^b F$ is the row vector multiplying $x^a y^b$ to each entry of $F$, i.e., $x^a y^b F = [x^a y^b f_1, x^a y^b f_2, \cdots, x^a y^b f_l]$.

## 4.2 Boundary gauge operators and gauge violation map

We have introduced computational tools such as vectors of truncated polynomials and the translational duplication map. This section will present an algorithm that determines all possible local boundary gauge operators given the given bulk stabilizers. The steps of the algorithm are outlined below, with detailed pseudocode provided in Appendix D.

The algorithm constructs boundary gauge operators, formed as products of Pauli $X$ and $Z$ operators, that commute with the bulk stabilizers. We first analyze how individual Pauli $X$ and $Z$ operators violate the bulk stabilizers and then combine these operators in such a way that commutes with all bulk stabilizers. For simplicity, we work within a large but finite truncated system, considering only the Pauli operators and bulk stabilizers fully supported within this region. To visualize this analysis, we construct the matrix $M_1$, which captures the commutation relations between single Pauli operators and bulk stabilizers within the truncated system:[14]

$$M_1 = \begin{pmatrix} \langle \mathcal{P}_1 \cdot \mathcal{BS}_1 \rangle_0 & \langle \mathcal{P}_1 \cdot \mathcal{BS}_2 \rangle_0 & \langle \mathcal{P}_1 \cdot \mathcal{BS}_3 \rangle_0 & \cdots \\ \langle \mathcal{P}_2 \cdot \mathcal{BS}_1 \rangle_0 & \langle \mathcal{P}_2 \cdot \mathcal{BS}_2 \rangle_0 & \langle \mathcal{P}_2 \cdot \mathcal{BS}_3 \rangle_0 & \cdots \\ \langle \mathcal{P}_3 \cdot \mathcal{BS}_1 \rangle_0 & \langle \mathcal{P}_3 \cdot \mathcal{BS}_2 \rangle_0 & \langle \mathcal{P}_3 \cdot \mathcal{BS}_3 \rangle_0 & \cdots \\ \vdots & \vdots & \vdots & \ddots \end{pmatrix}. \tag{38}$$

---

[14]In practical applications, incorporating every possible Pauli operator is not required. Instead, retaining only a sufficient subset of Pauli operators located near the boundary is efficient. This subset ensures it can construct all possible boundary gauge operators, quotiented by bulk stabilizers. The criterion for determining the sufficiency of the selected Pauli operators involves a dynamical process: we continue to add Pauli operators to our computation until no new boundary gauge operators (up to translation) are generated. Details are discussed in Appendix F.

The rows $\mathcal{P}$ of matrix $M_1$ are labeled by the Pauli operators, which include:

$$\mathcal{P} = \mathcal{X}_1, x\mathcal{X}_1, y\mathcal{X}_1, \cdots, \mathcal{X}_2, x\mathcal{X}_2, y\mathcal{X}_2, \cdots, \mathcal{Z}_1, x\mathcal{Z}_1, y\mathcal{Z}_1, \cdots, \mathcal{Z}_2, x\mathcal{Z}_2, y\mathcal{Z}_2, \cdots, \tag{39}$$

and the columns $\mathcal{BS}$ of matrix $M_1$ are labeled by the bulk stabilizers, which include:

$$\mathcal{BS} = \mathcal{S}_1, x\mathcal{S}_1, y\mathcal{S}_1, \cdots, \mathcal{S}_2, x\mathcal{S}_2, y\mathcal{S}_2, \cdots. \tag{40}$$

The terms $x^{a_i}y^{b_i}\mathcal{X}_i, x^{a_i}y^{b_i}\mathcal{Z}_i$, and $x^{c_j}y^{d_j}\mathcal{S}_j$ denote the translated versions of the single-Pauli operators $\mathcal{X}_i, \mathcal{Z}_i$ and the bulk stabilizer generator $\mathcal{S}_j$, respectively. The indices $a_i, b_i, c_j$, and $d_j$ are restricted by the size of the truncated system to ensure that the Pauli operators and bulk stabilizers are fully supported. Each entry in the matrix $M_1$, denoted $\langle \mathcal{P} \cdot \mathcal{BS} \rangle_0$, represents the constant term of the polynomial resulting from the dot product as described in Eq. (26). This term characterizes the commutation between a given Pauli operator and a bulk stabilizer.

By applying the **Modified Gaussian Elimination (MGE)** algorithm [50], reviewed in Appendix C, we can identify specific combinations of row vectors (Pauli operators) that commute with the bulk stabilizers. The procedure for constructing the boundary gauge operator $\mathcal{G}$ is outlined as follows:

- **Step 1:** Construct matrix $M_1$ to demonstrate how Pauli $X$ and $Z$ operators interact with the bulk stabilizers, as defined in Eq. (38).
- **Step 2:** Apply the Modified Gaussian Elimination (MGE) algorithm to $M_1$ to derive relation matrix $R_1$. Extract the local operator set $\mathcal{O}$ from the rows of $R_1$ that correspond to zero rows in the elimination process.
- **Step 3:** Identify non-trivial boundary gauge operators from the set $\mathcal{O}$, which may also include bulk stabilizers:

  - **Step 3-1:** Construct matrix $\widetilde{M}_2$ containing the bulk stabilizer generators and their translations:

  $$\widetilde{M}_2 := \begin{bmatrix} \overbrace{\mathrm{TD}_{c_1,d_1}(\mathcal{S}_1)} \\ \overline{\mathrm{TD}_{c_2,d_2}(\mathcal{S}_2)} \\ \vdots \end{bmatrix}, \tag{41}$$

  with translations constrained by the truncated system size.
  - **Step 3-2:** Apply the Modified Gaussian Elimination (MGE) on $\widetilde{M}_2$ to derive $\mathrm{MGE}(\widetilde{M}_2)$.
  - **Step 3-3:** Evaluate whether the first row of $\widetilde{\mathcal{O}}$ is spanned by the rows of $\mathrm{MGE}(\widetilde{M}_2)$. If it is, this row represents a trivial boundary gauge operator, and the process moves to the next row. If it is not, it is identified as a non-trivial boundary gauge operator $\mathcal{G}$. Then, update $\mathrm{MGE}(\widetilde{M}_2)$ by appending:

  $$\widetilde{M}_3 := \widetilde{\mathrm{TD}_{m_x=0,m_y}(\mathcal{G})}, \tag{42}$$

  and reapply the MGE to integrate this newly identified boundary gauge operator and its translations into the generating matrix. Repeat this evaluation for each subsequent row to identify all boundary gauge operators.

With the boundary gauge operators now determined, our next objective is to obtain the boundary anyon and the corresponding boundary string operator. The excitation map (33) demonstrates how Pauli operators violate the bulk stabilizer, leading to the formulation of the bulk anyon equation (34). However, creating boundary anyons requires the boundary string operator to commute with the bulk stabilizers. This constraint necessitates using only boundary gauge operators to construct the boundary string operator. Therefore, our task is to arrange these boundary gauge operators to form a boundary string operator that commutes with all bulk stabilizers while only failing to commute with boundary gauge operators at its endpoints.

To address this, we introduce the **gauge violation map**, which records violations of boundary gauge operators by a specific operator. For the generators[15] of boundary gauge operators $\mathcal{G}_1, \mathcal{G}_2, \ldots, \mathcal{G}_r$, the gauge violation map for a Pauli operator $\mathcal{P}$ is defined as:

$$\zeta(\mathcal{P}) := [\langle \mathcal{G}_1 \cdot \mathcal{P} \rangle_{x^0}, \langle \mathcal{G}_2 \cdot \mathcal{P} \rangle_{x^0}, \ldots, \langle \mathcal{G}_r \cdot \mathcal{P} \rangle_{x^0}] \tag{43}$$

where $\langle \mathcal{G}_i \cdot \mathcal{P} \rangle_{x^0}$ denotes the polynomial component of $\mathcal{G}_i \cdot \mathcal{P}$ where the exponent of $x$ is zero (the exponent of $y$ can be any integer), reflecting the fact that these operators preserve translational symmetry only in the

---

[15]The generators and their translations generate the entire gauge group.

$y$-direction. Each entry of $\zeta(\mathcal{P})$ is a component in $\mathbb{Z}_d[y, y^{-1}]$, forming a row vector with $r$ entries. Utilizing the gauge violation map to document these violations, we will introduce the boundary anyon equation in the subsequent section.

## 4.3 Computing boundary anyons and boundary string operators

In Sec. 4.2, we introduced boundary gauge operators and defined the gauge violation map (43). This section presents the boundary anyon equation and the equivalence relations between anyons. We then show how to solve the boundary anyon equation using boundary gauge operators to determine the possible boundary anyons and their corresponding string operators. Finally, we classify the boundary anyons by computing the Smith normal form, which identifies the **basis anyons**[16] of the boundary theory and their fusion rules. The algorithm pseudocode is provided in Appendix D.

In comparison to the bulk anyon equation (34), the boundary string operators must commute with all stabilizers and only violate the boundary gauge operators at their endpoints, without affecting the boundary gauge operators along the middle of the string. Given this property, any boundary string operator must be constructed from the boundary gauge operators. To achieve this, we use the gauge violation map, which records how an operator violates a boundary gauge operator. Since boundary string operators are constructed from boundary gauge operators, we begin by analyzing the gauge violation map of the generators of boundary gauge operators $\mathcal{G}_i \in \mathcal{G}$, and then combine them to form boundary string operators that create boundary anyons at their endpoints. Specifically, we define the **boundary anyon equation** to determine the boundary anyons:

$$\zeta\big(\alpha_1(y)\mathcal{G}_1 + \alpha_2(y)\mathcal{G}_2 + \alpha_3(y)\mathcal{G}_3 + ... + \alpha_r(y)\mathcal{G}_r\big) = (1 - y^n)[q_1(y), q_2(y), \cdots, q_r(y)] := (1 - y^n)v, \quad (44)$$

where $v$ is a length-$r$ row vector, referred to as a boundary anyon. This equation indicates that when the boundary gauge operators $\mathcal{G}_i$ are applied at locations $\alpha_i(y)$, they violate the boundary gauge operator near the origin $(0,0)$ and the point $(0,n)$ with patterns $v$ and $-v$, respectively.

To determine whether two bulk anyons are of the same type, we check if they differ by applying local operators, as described in Eq. (4). However, for boundary anyons, the local operators must not violate the bulk stabilizer, restricting them to the boundary gauge operators. We define the equivalence relation between anyons $v$ and $v'$ as follows:

$$v' \sim v \quad \text{(i.e., } v' \text{ is equivalent to } v), \tag{45}$$

if and only if there exist finite-degree polynomials $p_1(y), p_2(y), \ldots, p_r(y)$ such that

$$v' = v + p_1(y)\zeta(\mathcal{G}_1) + p_2(y)\zeta(\mathcal{G}_2) + \cdots + p_r(y)\zeta(\mathcal{G}_r), \tag{46}$$

Physically, two boundary anyons are equivalent if they differ only by the application of boundary gauge operators $\mathcal{G}_1, \mathcal{G}_2, \ldots, \mathcal{G}_r$ at positions determined by the polynomials $p_1(y), p_2(y), \ldots, p_r(y)$, meaning they can be transformed into one another through these local operations.

We now detail the algorithm used to solve the boundary anyon equation and ultimately obtain the basis boundary anyons. The steps are outlined as follows:

- **Step 1:** Compute the gauge violation map (43) for generators of boundary gauge operators $\mathcal{G}_i \in \mathcal{G}$ for $i = 1, 2, \cdots r$.
- **Step 2:** To solve the boundary anyon equation (44), construct the following matrix $\widetilde{M}_4$:

$$\widetilde{M}_4 := \begin{bmatrix} \overline{\mathrm{TD}_{m_x=0,m_y}(\zeta(\mathcal{G}_1))} \\ \overline{\mathrm{TD}_{m_x=0,m_y}(\zeta(\mathcal{G}_2))} \\ \vdots \\ \overline{\mathrm{TD}_{m_x=0,m_y}(\zeta(\mathcal{G}_r))} \\ \overline{\mathrm{TD}_{m_x=0,m_y}([(1-y^n), 0, ..., 0])} \\ \overline{\mathrm{TD}_{m_x=0,m_y}([0, (1-y^n), ..., 0])} \\ \vdots \\ \overline{\mathrm{TD}_{m_x=0,m_y}([0, 0, ..., (1-y^n)])} \end{bmatrix}, \tag{47}$$

---

[16]Basis anyons are a minimal set of anyons that generate all anyons.

where $\widetilde{M}_4$ is a $2(2m_y+1)r \times (2k+1)^2 r$ matrix. We examine $n = 1, 2, \ldots, n_0$ for large enough $n_0$ to ensure that all anyons are obtained, similar to the bulk anyons derived in Ref. [50].

- **Step 3:** By applying the modified Gaussian elimination, as outlined in Appendix C, to $\widetilde{M}_4$, we derive relations among the rows. These relations enable us to identify specific combinations of rows that sum to zero, where the coefficients of the top $(2m_y + 1)r$ rows correspond to the boundary string operators, denoted by $\alpha_i(y)$ in Eq. (44). Meanwhile, the coefficients of the bottom $(2m_y + 1)r$ rows correspond to the boundary anyons at the endpoints, labeled by $q_i(y)$ in Eq. (44).
- **Step 4:** At this stage, we have a set of boundary anyons $V = \{v_1, v_2, v_3, \cdots\}$ that may contain redundancies. Two anyons, $v$ and $v'$, are considered equivalent if they are related by local boundary gauge operators shown in Eq. (46). Thus, we aim to retain only the basis boundary anyons while eliminating the redundant ones. To do this, we add local boundary gauge operators at the endpoints of the strings to check whether the endpoints of two strings are equivalent. The following steps should be taken to achieve this:

  – **Step 4-1:** Construct the matrix $\widetilde{M}_5$ as follows:

$$\widetilde{M}_5 := \begin{bmatrix} \overline{\mathrm{TD}_{m_x=0,m_y}(\zeta(\mathcal{G}_1))} \\ \overline{\mathrm{TD}_{m_x=0,m_y}(\zeta(\mathcal{G}_2))} \\ \vdots \\ \overline{\mathrm{TD}_{m_x=0,m_y}(\zeta(\mathcal{G}_r))} \end{bmatrix}, \tag{48}$$

  which corresponds to trivial boundary anyons.

  – **Step 4-2:** We begin with $M_{span} := \mathrm{MGE}(\widetilde{M}_5)$ and an initially empty set $V\mathrm{gen} := \{\}$. The goal is to sequentially examine the boundary anyons in the set $V$, while $M_{span}$ tracks the space spanned by trivial boundary anyons and those that have been processed up to that point. First, we check whether each boundary anyon $\widetilde{v}$ can be expressed as a linear combination of the rows of $M_{span}$. If $\widetilde{v}$ is spanned by the rows of $M_{span}$, it is redundant, and we move on to the next boundary anyon. However, if $\widetilde{v}$ is not spanned by the rows of $M_{span}$, we treat it as a generator of $V$ and append it to the generator set $V_{\mathrm{gen}}$. To maintain the spanning space, we update $M_{span}$ by incorporating $\widetilde{v}$ into the previous $M_{span}$ and performing Modified Gaussian Elimination to clean up the matrix. This process ensures that $M_{span}$ includes the newly identified basis boundary anyon $\widetilde{v}$. This procedure is repeated for each subsequent anyon in $V$ until generators of $V$ have all been identified in $V_{\mathrm{gen}}$.

  – **Step 4-3:** Boundary anyons in $V_{\mathrm{gen}}$ can still be redundant, for example $V_{\mathrm{gen}} = \{e^2, e, m\}$ for the $\mathbb{Z}_4$ toric code. Following Ref. [50], we can construct the relation matrix of these anyons and compute its Smith normal form. This process yields matrices $P$, $Q$, and $A$, which satisfy the relation $PMQ = A$, where $Q$ is unimodular (i.e., $\det Q = \pm 1$) and can be used to identify the rearranged basis boundary anyons, with the orders of the basis boundary anyons corresponding to the diagonal elements of matrix $A$.

## 4.4 Boundary anyon condensation and Topological order completion

After deriving the boundary string operators, the next step is to explore various boundary constructions based on the Lagrangian subgroup. Using the folding argument in Fig. 5, we can subsequently construct defects as the boundary of the folded system. A crucial part of the construction is to ensure that the topological order (TO) condition is satisfied. The construction steps are as follows:

- **Step 1:** Using the boundary anyons and the corresponding short boundary string operators derived from Sec. 4.2, calculate the topological spin $\theta(a)$ from Eq. (10) and the braiding statistics $B(a, b)$ from Eq. (11). According to the bulk-boundary correspondence (Theorem 4), the topological data of the boundary matches that of the bulk. Thus, the Lagrangian subgroup determined for the boundary will dictate the bulk anyon condensation.
- **Step 2:** Based on the Lagrangian subgroup, add products of short boundary string operators (with weak translational symmetry in the $y$-direction) to the Hamiltonian. These short boundary string operators become stabilizers in the new Hamiltonian.
- **Step 3:** After adding all the short boundary string operators for the anyons in the chosen Lagrangian subgroup, apply the procedure from Sec. 4.2 to derive an operator that commutes with the stabilizers but is not itself a product of stabilizers. Incorporate this operator, along with its translations, into the Hamiltonian as new stabilizers. Repeat this process iteratively until no additional operators can be found. This procedure is referred to as topological order completion, ensuring that the TO condition is satisfied.
- **Step 4:** With the boundary explicitly constructed, use the method outlined in Appendix F.1 to derive the bulk string operators that can terminate on the boundary without energy cost.

# 5 Applications to boundary and defect constructions of quantum codes

This section applies our algorithm to various topological Pauli stabilizer codes and presents the corresponding boundary and defect constructions. These include the $\mathbb{Z}_2$ standard toric code (Sec. 5.1), the $\mathbb{Z}_2$ fish toric code (Sec. 5.2), the $\mathbb{Z}_4$ standard toric code (Sec. 5.3), the double semion code (Sec. 5.4), the six-semion code (Sec. 5.5), the color code (Sec. 5.6), and the anomalous three-fermion code (Sec. 5.7). The number of distinct boundaries and defects is shown in Table 1. Since the three-fermion code is anomalous and can only exist on the boundary of (3+1)D topological phases [86, 100, 106], we discuss only its defects. We note that it is not a coincidence that both the $\mathbb{Z}_2$ toric code and the three-fermion code have 6 defects, and both the $\mathbb{Z}_4$ toric code and the six-semion code have 22 defects. By re-arranging the anyons in two copies of the $\mathbb{Z}_2$ toric code, we can obtain two copies of the three-fermion code. Similarly, re-arranging the anyons in two copies of the $\mathbb{Z}_4$ toric code yields two copies of the six-semion code, as we will demonstrate in Secs. 5.5 and 5.7.

Additionally, we provide the boundary gauge operators for two specific bivariate bicycle (BB) codes, $(3, 3)$-BB codes and $(2, −3)$-BB codes, in Secs. 5.8 and 5.9. As proposed in Ref. [26], BB codes offer high-threshold, low-overhead, fault-tolerant quantum memory. These two BB codes are equivalent to 8 and 10 copies of the $\mathbb{Z}_2$ toric code by Clifford circuits, which have 1,270,075,950 and 167,448,083,323,950 distinct gapped boundary constructions, respectively (as calculated in Appendix E). Consequently, we do not present explicit boundary or defect constructions for these codes; instead, we highlight the unique properties of the anyon string operators. Notably, the "shortest string operators" for anyons in BB codes are relatively long compared to the stabilizer generator size. In the two examples below, the string operator lengths are 12 and 1023 times the lattice constant in the square lattice, respectively. Stabilizer codes with long string operators have been studied in Refs. [107, 108].

For general $(a, b)$-BB codes, as defined in detail in Sec. 5.8, with small values of $a$ and $b$, the number of basis anyons, $k$, and the length $l$ of the shortest period of all anyons in the $y$-direction are presented in Table 2.[17] Notably, $k$ corresponds precisely to the dimension of the logical space when the BB code is placed on an $l \times l$ torus.

|  | boundary | defect |
|---|---|---|
| $\mathbb{Z}_2$ toric code | 2 | 6 |
| $\mathbb{Z}_2$ fish toric code | 2 | 6 |
| $\mathbb{Z}_4$ toric code | 3 | 22 |
| double semion | 1 | 2 |
| six-semion code | 1 | 22 |
| color code | 6 | 270 |
| three-fermion code | N/A | 6 |

**Table 1** The number of distinct boundaries and defects in various topological Pauli stabilizer codes, derived by identifying the Lagrangian subgroups of the corresponding anyon theories.

## 5.1 $\mathbb{Z}_2$ standard toric code

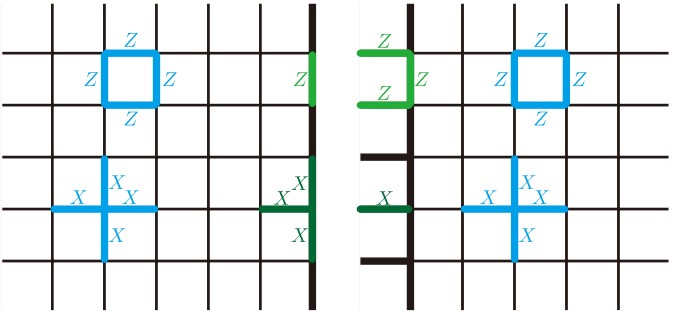

**Fig. 13** The left side illustrates a smooth boundary, while the right side illustrates a rough boundary. The blue components represent the bulk stabilizers of the $\mathbb{Z}_2$ toric code, and the green components represent the boundary gauge operators, which also serve as the short boundary string operators. The corresponding boundary anyons are labeled from top to bottom as $e_1$ and $m_1$ for the smooth boundary, and $e_2$ and $m_2$ for the rough boundary.

---

[17] Due to symmetry, the string length in the $x$-direction for the $(a, b)$-BB code is equivalent to the string length in the $y$-direction for the $(b, a)$-BB code. Therefore, only the string length in the $y$-direction is listed in the table.

| a\b | -3 | -2 | -1 | 0 | 1 | 2 | 3 |
|---|---|---|---|---|---|---|---|
| **-3** | $k=18$ $l=42$ | $k=12$ $l=63$ | $k=6$ $l=7$ | $k=8$ $l=3$ | $k=14$ $l=127$ | $k=20$ $l=341$ | $k=26$ $l=762$ |
| **-2** | $k=12$ $l=21$ | $k=8$ $l=15$ | $k=0$ $l=1$ | $k=8$ $l=15$ | $k=12$ $l=21$ | $k=16$ $l=217$ | $k=20$ $l=889$ |
| **-1** | $k=6$ $l=7$ | $k=0$ $l=1$ | $k=6$ $l=7$ | $k=8$ $l=15$ | $k=10$ $l=31$ | $k=12$ $l=21$ | $k=14$ $l=127$ |
| **0** | $k=8$ $l=3$ | $k=8$ $l=3$ | $k=8$ $l=3$ | $k=8$ $l=3$ | $k=8$ $l=3$ | $k=8$ $l=3$ | $k=8$ $l=3$ |
| **1** | $k=14$ $l=127$ | $k=12$ $l=9$ | $k=10$ $l=31$ | $k=8$ $l=15$ | $k=0$ $l=1$ | $k=6$ $l=7$ | $k=6$ $l=7$ |
| **2** | $k=20$ $l=1023$ | $k=16$ $l=217$ | $k=12$ $l=9$ | $k=8$ $l=15$ | $k=6$ $l=7$ | $k=0$ $l=1$ | $k=10$ $l=31$ |
| **3** | $k=26$ $l=762$ | $k=20$ $l=889$ | $k=14$ $l=127$ | $k=8$ $l=3$ | $k=6$ $l=7$ | $k=10$ $l=31$ | $k=16$ $l=12$ |

**Table 2** For each $(a,b)$-BB code, the entry indicates the number $k$ of basis anyons and the shortest string length $l$ required to obtain all anyons in the $y$-direction. Notably, all lengths can be expressed by the forms 2 and $2^i - 1$. For instance, $217 = 7 \times 31$, $341 = 1023/3$, $762 = 2 \times 3 \times 127$, and $889 = 7 \times 127$.

We consider the $\mathbb{Z}_2$ standard toric code as an introductory example. The boundary gauge operators for both the smooth and rough boundaries, obtained using Algorithm 1 outlined in Sec. 4.2 and Appendix D, are shown in Fig. 13 in green, with the bulk stabilizers depicted in blue. In the case of the $\mathbb{Z}_2$ toric code, the boundary gauge operators also serve as the short boundary string operators, derived through Algorithm 2, as described in Sec. 4.3 and Appendix D. For clarity, we label the corresponding boundary anyons in Fig. 13 from top to bottom as $e_1$ and $m_1$ for the smooth boundary, and $e_2$ and $m_2$ for the rough boundary. The fusion rules are $e_i^2 = m_i^2 = 1$, $\forall i \in \{1,2\}$, indicating that all have order 2. Their topological spins (Eq. (10)) and braiding statistics (Eq. (11)) are given by:

$$\theta(e_i) = \theta(m_i) = 1, \quad B(e_i, m_i) = -1, \quad \forall i \in \{1,2\},$$
$$B(a_1, a_2) = 1, \quad \forall a_1 \in \{e_1, m_1\}, \ a_2 \in \{e_2, m_2\}. \tag{49}$$

This describes two copies of the $\mathbb{Z}_2$ toric code (due to the presence of both smooth and rough boundaries), thereby confirming the bulk-boundary correspondence in Theorem 4.

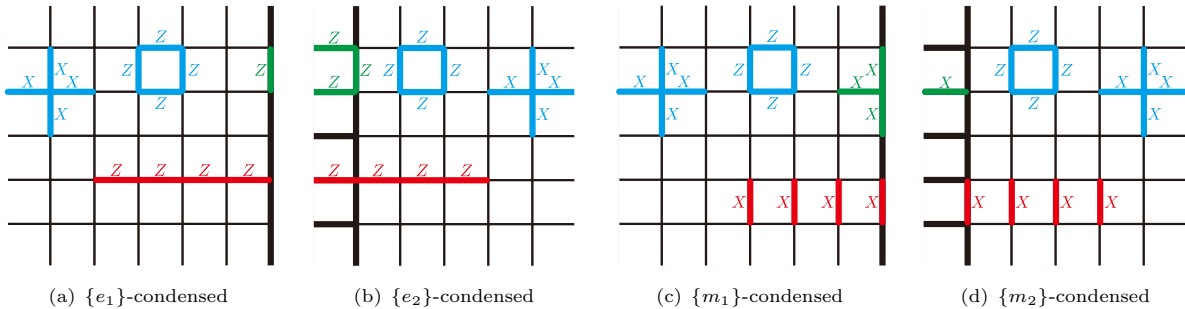

(a) $\{e_1\}$-condensed     (b) $\{e_2\}$-condensed     (c) $\{m_1\}$-condensed     (d) $\{m_2\}$-condensed

**Fig. 14** The boundaries of $\mathbb{Z}_2$ toric code. Blue components represent the bulk stabilizers of the $\mathbb{Z}_2$ toric code, green components represent the boundary Hamiltonian, and red components represent bulk strings that terminate at the boundary without causing energy violations.

We now demonstrate the construction of the gapped boundary. The explicit boundary constructions of the $\mathbb{Z}_2$ toric code are shown in Fig. 14. Green components represent short boundary string operators of bosons in the Lagrangian subgroups (with translational symmetry in the $y$-direction), while red components correspond to bulk anyon string operators. These bulk string operators violate the bulk stabilizers without violating the boundary Hamiltonian, indicating the condensation of bulk anyons at the boundary. For clarity, we denote the short boundary string operators of boundary anyons $e_i$ and $m_i$ as $O_{e_i}$ and $O_{m_i}$, respectively. The boundary construction process is outlined as follows:

1. **Determine Lagrangian subgroups:** First, we identify the Lagrangian subgroups for the boundary anyons: $\{1, e_i\}$ and $\{1, m_i\}$. The $\mathbb{Z}_2$ toric code allows two types of boundary condensation.

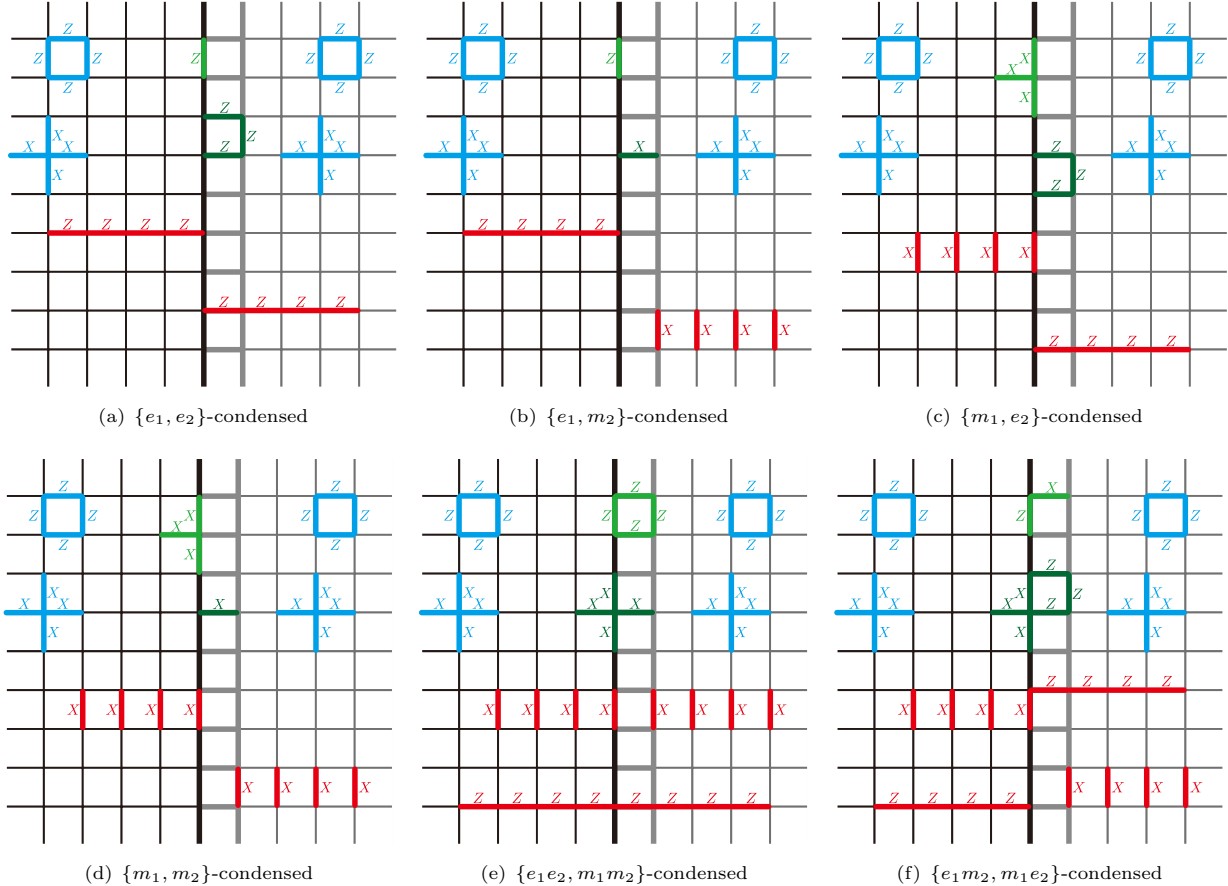

**Fig. 15** The defects of the $\mathbb{Z}_2$ toric code. Blue components indicate bulk stabilizers, green components represent the defect Hamiltonian, and red components show bulk string operators that terminate on or pass through the defect. The red strings commute with the green defect Hamiltonian. Figures (a), (b), (c), and (d) depict non-invertible defects, where the left-hand side and right-hand side are decoupled, with $e_1$ or $m_1$ and $e_2$ or $m_2$ condensed independently. Figures (e) and (f) illustrate invertible defects: (e) corresponds to the trivial defect, where the defect Hamiltonian matches the bulk Hamiltonian, and (f) represents the $e$-$m$ exchange defect, where $e$ and $m$ are permuted as they pass through the defect.

2. **Boundary anyon condensation:** According to the condensation procedure in Theorem 9, there are two possible constructions for the boundary Hamiltonian: either adding $O_{e_i}$ to condense $e_i$, or adding $O_{m_i}$ to condense $m_i$. Since $O_{e_i}$ and $O_{m_i}$ do not commute, adding $O_{e_i}$ with translational symmetry in the $y$-direction to the boundary prevents the inclusion of $O_{m_i}$.

3. **Topological order completion:** After adding the short boundary string operators, we apply Algorithm 1 to search for any additional boundary gauge operators needed to satisfy the TO condition, as described in Theorem 10. In the case of the standard toric code, no additional boundary gauge operators are required.

4. **Bulk strings condensed at the boundary:** Following the procedure in Appendix F.1, we obtain the condensed bulk string operators at the boundary after selecting the boundary Hamiltonian. This corresponds to the red components in Fig. 14.

So far, we have only considered the boundary Hamiltonian on the smooth or rough boundary of the $\mathbb{Z}_2$ toric code. For defect construction, however, we must simultaneously consider both the left and right semi-infinite systems in Fig. 14. Specifically, we select Lagrangian subgroups in the doubled theory $\{1, e_1, m_1, f_1\} \times \{1, e_2, m_2, f_2\}$ to condense at the defect. The explicit defect construction is illustrated in Fig. 15, where blue components represent bulk stabilizers on both sides, green components represent defect Hamiltonian, and red components show bulk string operators terminating on or passing through the defect. In the example of $\{e_1 m_2, m_1 e_2\}$ condensation (Fig. 15(f)), as $e_1$ moves through the defect, it transforms into $m_2$. Similarly, $m_1$ transforms into $e_2$. The defect construction process is outlined below and closely resembles the boundary construction process:

1. **Determine Lagrangian subgroups:** The Lagrangian subgroups of two copies of $\mathbb{Z}_2$ toric codes are: $\{e_1, e_2\}, \{e_1, m_2\}, \{m_1, m_2\}, \{m_1, e_2\}, \{e_1 e_2, m_1 m_2\}, \{e_1 m_2, m_1 e_2\}$.

2. **Defect anyon condensation:** Following the condensation procedure in Theorem 9, there are six distinct types of defect constructions. These defect string operators[18] are formed by appropriately combining boundary string operators at both smooth and rough boundaries, following the structure of the Lagrangian subgroup. A crucial part of this process is ensuring the correct combination and placement of the boundary string operators.

   For example, when condensing the set $\{e_1 m_2, m_1 e_2\}$, we construct the $e_1 m_2$ defect string operator, denoted $O_{e_1 m_2}$, by combining $O_{e_1}$ at the left smooth boundary with $O_{m_2}$ at the right boundary. Similarly, we form the $m_1 e_2$ defect string operator, denoted $O_{m_1 e_2}$, by combining $O_{m_1}$ at the left smooth boundary with $O_{e_2}$ at the right boundary. It is crucial that $O_{m_1 e_2}$ commutes with $O_{e_1 m_2}$.

3. **Topological order completion:** After adding the short defect string operators into the defect Hamiltonian, we apply Algorithm 1 to search for any additional defect gauge operators to satisfy the TO condition. In the case of the standard toric code, no additional defect gauge operators are required.

4. **Bulk strings terminating on or passing through the defect:** Following the procedure in Appendix F.1, we obtain the bulk string operator terminating on or passing through the defect after selecting the defect Hamiltonian. This is represented by the red components in the configuration shown in Fig. 15.

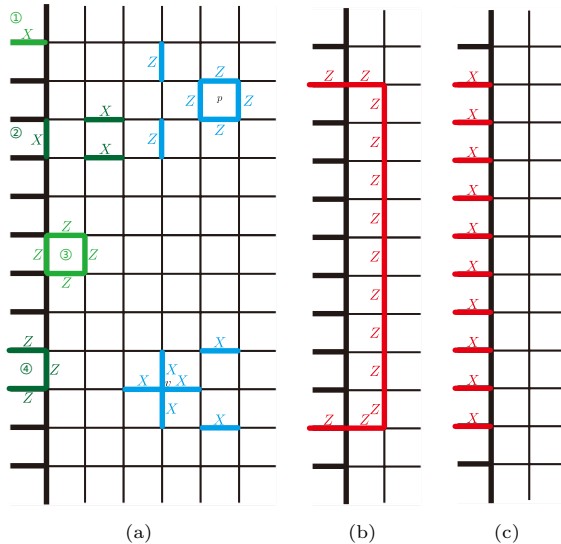

(a)  (b)  (c)

**Fig. 16** (a) Boundary gauge operators for the rough boundary, labeled ①, ②, ③, and ④, along with their translational counterparts. The operators ① and ② represent nontrivial *secondary boundary gauge operators*, meaning they commute with all bulk stabilizers but are not truncated $A_v^{\text{fish}}$ or $B_p^{\text{fish}}$ operators. (b) and (c) depict the boundary string operators along the rough boundary.

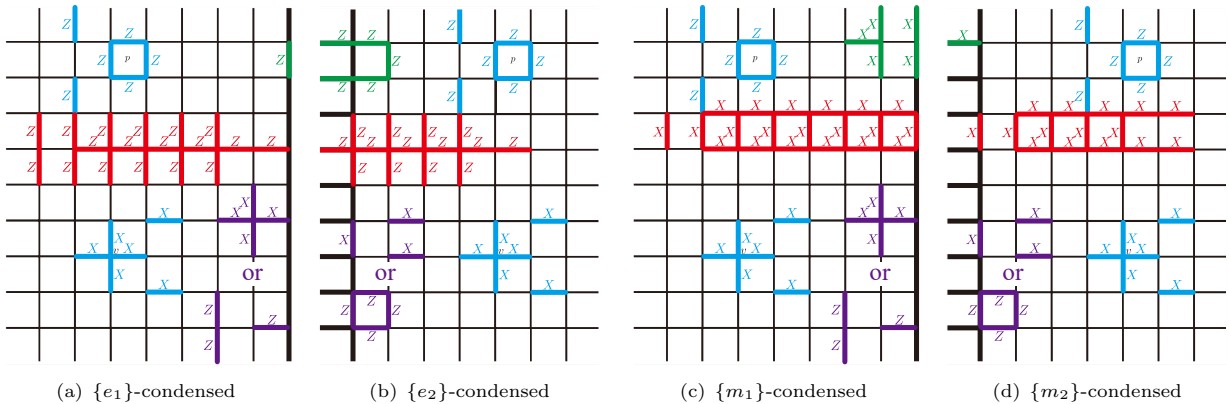

(a) $\{e_1\}$-condensed  (b) $\{e_2\}$-condensed  (c) $\{m_1\}$-condensed  (d) $\{m_2\}$-condensed

**Fig. 17** The boundaries of $\mathbb{Z}_2$ fish toric code. Blue components represent the bulk stabilizers of the $\mathbb{Z}_2$ fish toric code, green and purple components represent the boundary Hamiltonian, and red components represent bulk strings that terminate at the boundary without causing energy violations.

---

[18]We refer to these as defect string operators to maintain consistency with the previously used term boundary string operators. This naming convention will also apply to defect gauge operators in the following discussion.

## 5.2 $\mathbb{Z}_2$ fish toric code

As a second example, in contrast to the $\mathbb{Z}_2$ standard toric code, we demonstrate the necessity of **topological order completion** (Theorem 10) in this case. The bulk stabilizers, boundary gauge operators, and boundary string operators of the $\mathbb{Z}_2$ fish toric code for the smooth boundary are shown in Fig. 4 in Sec. 2, while the rough boundary is depicted in Fig. 16.

Since the $\mathbb{Z}_2$ fish toric code is equivalent to the $\mathbb{Z}_2$ standard toric code conjugated by a finite-depth Clifford circuit, it does not alter the bulk topological data of the standard toric code. Therefore, according to the bulk-boundary correspondence, the topological properties of the boundary anyons remain the same as in Eq. (49). Consequently, there are two types of boundary constructions and six types of defect constructions. Following the procedure outlined in the previous section, the boundary constructions of the fish toric code are depicted in Fig. 17. In addition to the short boundary string operators along the boundary, other boundary gauge operators (highlighted in purple) are included in the boundary Hamiltonian. For the smooth boundary, we can add the boundary gauge operator ① or ⑤ as shown in Fig. 4(b). For the rough boundary, we can add boundary gauge operator ② or ③ as shown in Fig. 16(a). These boundary gauge operators can be selected arbitrarily as long as the topological order condition is satisfied, and the choice does not affect the condensation properties (Theorem 10).

Finally, the explicit defect constructions are illustrated in Fig. G4 in Appendix G. Additional defect gauge operators must be incorporated into the defect Hamiltonian, similar to the boundary case. In each instance of condensation, the defect Hamiltonian satisfies the topological order condition.

## 5.3 $\mathbb{Z}_4$ toric code

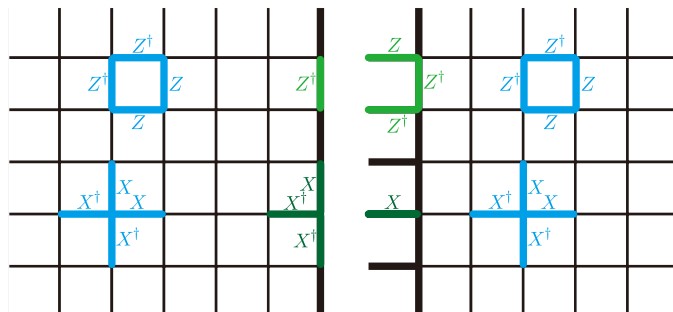

**Fig. 18** The left side illustrates a smooth boundary, while the right side illustrates a rough boundary. The blue components represent the bulk stabilizers of the $\mathbb{Z}_4$ toric code, and the green components represent the short boundary gauge operators, which also serve as the short boundary string operators. The corresponding boundary anyons are labeled from top to bottom as $e_1$ and $m_1$ for the smooth boundary, and $e_2$ and $m_2$ for the rough boundary.

We apply our algorithm to nonprime-dimensional qudits by exploring explicit boundary and defect constructions for the $\mathbb{Z}_4$ toric code, in contrast to the $\mathbb{Z}_2$ case. The boundary gauge operators are shown in Fig. 18, which also serve as the short boundary string operators. For clarity, we label the corresponding boundary anyons from the top boundary string operator to the bottom as $e_1$ and $m_1$ for the smooth boundary, and $e_2$ and $m_2$ for the rough boundary. The fusion rules are $e_i^4 = m_i^4 = 1$, $\forall i \in \{1, 2\}$, indicating that all have order 4. Their topological spins (Eq. (10)) and braiding statistics (Eq. (11)) are given by:

$$\begin{aligned}
\theta(e_j) = \theta(m_j) = 1, \quad B(e_j, m_j) = i, \quad j \in \{1, 2\}, \\
B(a_1, a_2) = 1, \quad \forall a_1 \in \{e_1, m_1\}, \ a_2 \in \{e_2, m_2\}.
\end{aligned} \tag{50}$$

Following the same construction method, Fig. 19 illustrates 3 types of boundary constructions, while 22 types of defect constructions are presented in Figs. G5 and G6 in Appendix G. The Lagrangian subgroups are labeled individually in each of these figures.

## 5.4 Double semion code

We study boundary and defect constructions for the double semion code with $\mathbb{Z}_4$ qudits as another example of nonprime-dimensional qudits. The boundary gauge operators are illustrated in Fig. 20, which also serve as the short boundary string operators. For clarity, we label the corresponding boundary anyons from the top to the bottom as $b_1$ and $s_1$ for the smooth boundary, and $b_2$ and $s_2$ for the rough boundary. The fusion rules are $b_i^2 = s_i^2 = 1$, $\forall i \in \{1, 2\}$, indicating that all have order 2. Their topological spins (Eq. (10)) and

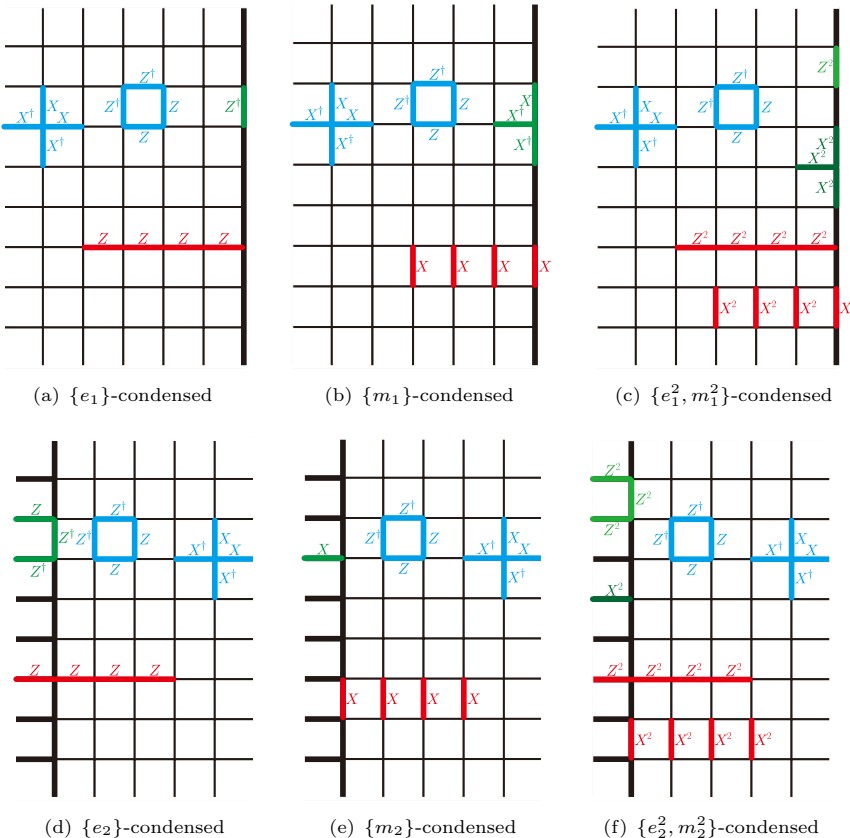

**Fig. 19** The boundaries of $\mathbb{Z}_4$ toric code. Blue components represent the bulk stabilizers of the $\mathbb{Z}_4$ toric code, green components represent the boundary Hamiltonian, and red components represent bulk strings that terminate at the boundary without causing energy violations.

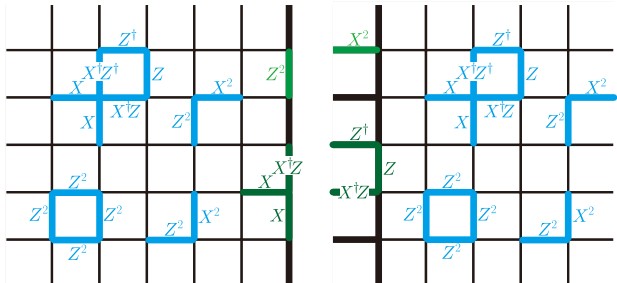

**Fig. 20** The left side illustrates a smooth boundary, while the right side illustrates a rough boundary. The blue components represent the bulk stabilizers of the double semion code with $\mathbb{Z}_4$ qudits, and the green components represent the short boundary gauge operators, which also serve as the short boundary string operators. The corresponding boundary anyons are labeled from top to bottom as $b_1$ and $s_1$ for the smooth boundary, and $b_2$ and $s_2$ for the rough boundary.

braiding statistics (Eq. (11)) are given by:

$$\theta(b_j) = 1, \quad \theta(s_j) = i, \quad B(b_j, s_j) = -1, \quad j \in \{1, 2\},$$
$$B(a_1, a_2) = 1, \quad \forall a_1 \in \{b_1, s_1\}, \ a_2 \in \{b_2, s_2\}.$$

where $s_i$ is a semion and $b_i$ is a boson. Thus, we can only condense $b_i$ at both the smooth and rough boundaries, as shown in Fig. 21 ($s_i$ cannot be condensed since its boundary string operators do not commute with themselves). Furthermore, there are 2 types of defect construction, as illustrated in Fig. 22: condensing $\{b_1, b_2\}$ and condensing $\{b_1 b_2, s_1 s_2\}$.

Interestingly, we can consider orientation-reversing defects in the double semion code, where one side of the semi-infinite plane is flipped upside-down. This setup is equivalent to placing quantum codes on unorientable manifolds, such as a Klein bottle, which can lead to the emergence of additional logical gates [56]. Although the higher-group symmetry structure of these defects has been studied at the field theory level [47, 54], an explicit lattice construction remains to be demonstrated. Fig. 23 illustrates the defect constructions, where the double semion code on the left is flipped upside-down relative to the one on the right.

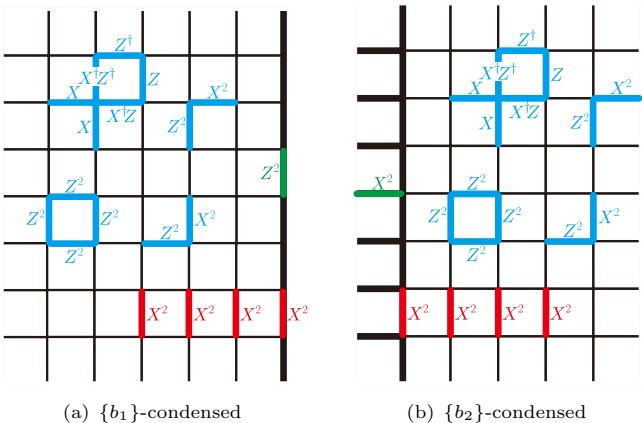

(a) $\{b_1\}$-condensed      (b) $\{b_2\}$-condensed

**Fig. 21** The boundaries of $\mathbb{Z}_4$ double semion. Blue components represent the bulk stabilizers of the $\mathbb{Z}_4$ double semion, green components represent the boundary Hamiltonian, and red components represent bulk strings that terminate at the boundary without causing energy violations.

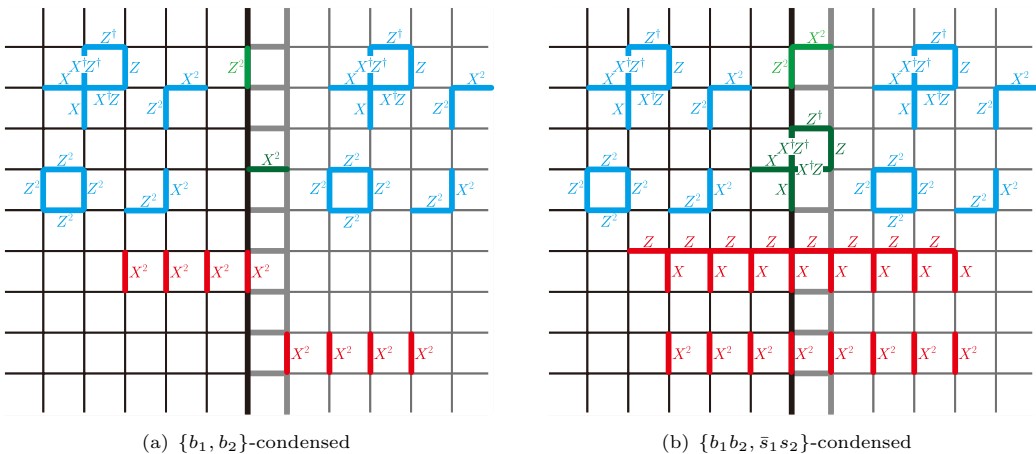

(a) $\{b_1, b_2\}$-condensed      (b) $\{b_1 b_2, \bar{s}_1 s_2\}$-condensed

**Fig. 22** The defects of the $\mathbb{Z}_4$ double semion code. Blue components indicate bulk stabilizers, green components represent the defect Hamiltonian, and red components show bulk string operators that terminate on or pass through the defect. The red strings commute with the green defect Hamiltonian.

## 5.5 Six-semion code

We present another example involving $\mathbb{Z}_4$ qudits: the six-semion code. The stabilizers and boundary gauge operators are depicted in Fig. 24, where each edge contains two $\mathbb{Z}_4$ qudits labeled by subscripts 1 and 2. The boundary gauge operators ② and ④ on the smooth boundary, as well as ⑤ and ⑦ on the rough boundary, also serve as short boundary string operators. We label the corresponding boundary anyons as $\tilde{e}_1$, $\tilde{m}_1$, $\tilde{e}_2$, and $\tilde{m}_2$. The fusion rules are $\tilde{e}_i^4 = \tilde{m}_i^4 = 1$, $\forall i \in \{1, 2\}$, indicating that all have order 4. Their topological spins (Eq. (10)) and braiding statistics (Eq. (11)) are given by:

$$
\begin{aligned}
\theta(\tilde{e}_j) = \theta(\tilde{m}_j) = i, \quad B(\tilde{e}_1, \tilde{m}_1) = i, \quad j \in \{1, 2\}, \\
B(a_1, a_2) = 1, \quad \forall a_1 \in \{\tilde{e}_1, \tilde{m}_1\}, \ a_2 \in \{\tilde{e}_2, \tilde{m}_2\}.
\end{aligned}
\tag{51}
$$

Compared to Eq. (50), the six-semion code is analogous to the $\mathbb{Z}_4$ toric code, but with bosons $e$ and $m$ replaced by semions $\tilde{e}$ and $\tilde{m}$. In this anyon theory, there are 16 anyons, with 4 bosons $1, \tilde{e}^2, \tilde{m}^2, \tilde{e}^2\tilde{m}^2$, 6 semions $\tilde{e}, \tilde{m}, \tilde{e}^3, \tilde{m}^3, \tilde{e}\tilde{m}^3, \tilde{e}^3\tilde{m}$, and 6 anti-semions $\tilde{e}\tilde{m}^2, \tilde{e}^2\tilde{m}, \tilde{e}^3\tilde{m}^2, \tilde{e}^2\tilde{m}^3, \tilde{e}\tilde{m}, \tilde{e}^3\tilde{m}^3$. There is a single boundary construction, corresponding to the condensation of all bosons in the six-semion code, generated by $\tilde{e}^2, \tilde{m}^2$, as shown in Fig. 25. Additionally, 22 types of defect constructions correspond to the condensation

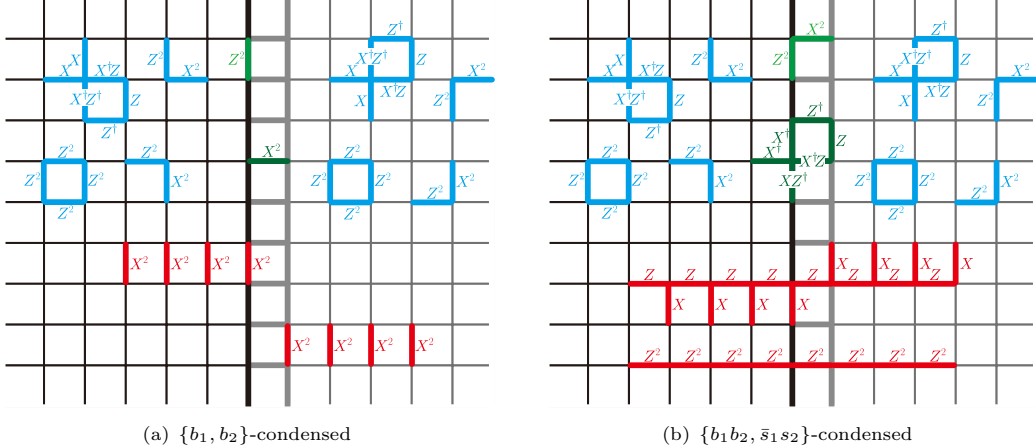

**Fig. 23** Orientation-reversing defects in the $\mathbb{Z}_4$ double semion code. The left side shows the orientation-reversed double semion code, flipped upside-down, compared to the double semion code on the right. Blue components indicate bulk stabilizers, green components represent the defect Hamiltonian, and red components show bulk string operators that terminate on or pass through the defect. The red strings commute with the green defect Hamiltonian.

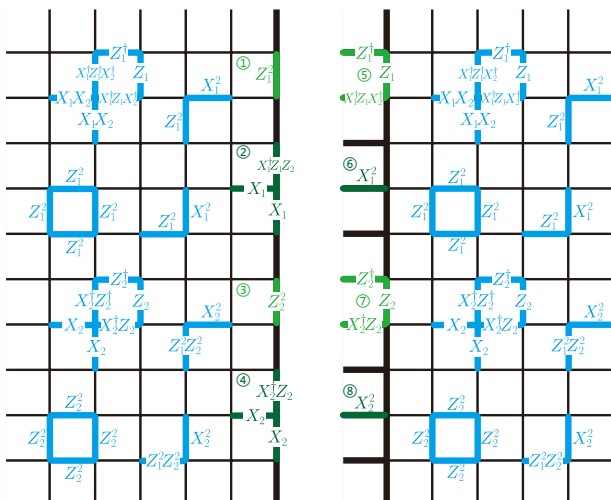

**Fig. 24** The left side illustrates a smooth boundary, while the right side illustrates a rough boundary. The blue components represent the bulk stabilizers of the $\mathbb{Z}_4$ six-semion code, while the green components represent the boundary gauge operators. The boundary gauge operators ② and ④ on the smooth boundary, as well as ⑤ and ⑦ on the rough boundary also serve as the short boundary string operators. The corresponding boundary anyons are labeled as $\tilde{e}_1$, $\tilde{m}_1$, $\tilde{e}_2$ and $\tilde{m}_2$.

of the following Lagrangian subgroups:

$$\mathcal{L}_1 = \{\tilde{e}_1^2, \tilde{m}_1^2, \tilde{e}_2^2, \tilde{m}_2^2\}, \ \mathcal{L}_2 = \{\tilde{e}_1\tilde{m}_1\tilde{e}_2\tilde{m}_2^3, \tilde{m}_1^2\tilde{m}_2^2, \tilde{e}_2^2\tilde{m}_2^2\}, \ \mathcal{L}_3 = \{\tilde{e}_1\tilde{e}_2^2\tilde{m}_2^3, \tilde{m}_1\tilde{e}_2\tilde{m}_2\},$$

$$\mathcal{L}_4 = \{\tilde{e}_1\tilde{e}_2\tilde{m}_2^2, \tilde{m}_1^2\tilde{m}_2^2, \tilde{e}_2^2\}, \ \mathcal{L}_5 = \{\tilde{e}_1\tilde{e}_2^2\tilde{m}_2^3, \tilde{m}_1\tilde{e}_2\tilde{m}_2^2\}, \ \mathcal{L}_6 = \{\tilde{e}_1\tilde{m}_1\tilde{m}_2, \tilde{m}_1^2\tilde{e}_2^2, \tilde{m}_2^2\},$$

$$\mathcal{L}_7 = \{\tilde{e}_1\tilde{e}_2^3\tilde{m}_2^2, \tilde{m}_1\tilde{e}_2^2\tilde{m}_2\}, \ \mathcal{L}_8 = \{\tilde{e}_1^2\tilde{m}_2^2, \tilde{m}_1\tilde{e}_2\tilde{m}_2, \tilde{e}_2^2\tilde{m}_2^2\}, \ \mathcal{L}_9 = \{\tilde{e}_1\tilde{e}_2\tilde{m}_2, \tilde{m}_1\tilde{e}_2^2\tilde{m}_2^3\},$$

$$\mathcal{L}_{10} = \{\tilde{e}_1\tilde{e}_2\tilde{m}_2, \tilde{m}_1^2\tilde{m}_2^2, \tilde{e}_2^2\tilde{m}_2^2\}, \ \mathcal{L}_{11} = \{\tilde{e}_1\tilde{e}_2\tilde{m}_2, \tilde{m}_1\tilde{e}_2^3\tilde{m}_2^2\}, \ \mathcal{L}_{12} = \{\tilde{e}_1\tilde{e}_2^3\tilde{m}_2^3, \tilde{m}_1\tilde{e}_2^2\tilde{m}_2\},$$

$$\mathcal{L}_{13} = \{\tilde{e}_1\tilde{e}_2\tilde{m}_2^2, \tilde{m}_1\tilde{e}_2^3\tilde{m}_2^3\}, \ \mathcal{L}_{14} = \{\tilde{e}_1\tilde{e}_2\tilde{m}_2^2, \tilde{m}_1\tilde{e}_2^2\tilde{m}_2^3\}, \ \mathcal{L}_{15} = \{\tilde{e}_1^2\tilde{m}_2^2, \tilde{m}_1\tilde{e}_2\tilde{m}_2^2, \tilde{e}_2^2\},$$

$$\mathcal{L}_{16} = \{\tilde{e}_1\tilde{e}_2^3\tilde{m}_2^2, \tilde{m}_1\tilde{e}_2\tilde{m}_2\}, \ \mathcal{L}_{17} = \{\tilde{e}_1\tilde{e}_2^2\tilde{m}_2, \tilde{m}_1^2\tilde{e}_2^2, \tilde{m}_2^2\}, \ \mathcal{L}_{18} = \{\tilde{e}_1\tilde{e}_2^3\tilde{m}_2^3, \tilde{m}_1\tilde{e}_2\tilde{m}_2^2\},$$

$$\mathcal{L}_{19} = \{\tilde{e}_1\tilde{m}_1\tilde{e}_2, \tilde{m}_1^2\tilde{m}_2^2, \tilde{e}_2^2\}, \ \mathcal{L}_{20} = \{\tilde{e}_1\tilde{e}_2^2\tilde{m}_2, \tilde{m}_1\tilde{e}_2^3\tilde{m}_2^2\}, \ \mathcal{L}_{21} = \{\tilde{e}_1^2\tilde{e}_2^2, \tilde{m}_1\tilde{e}_2^2\tilde{m}_2, \tilde{m}_2^2\},$$

$$\mathcal{L}_{22} = \{\tilde{e}_1\tilde{e}_2^2\tilde{m}_2, \tilde{m}_1\tilde{e}_2^3\tilde{m}_2^3\}.$$

For simplicity, we only list the generators of each Lagrangian subgroup above. The construction follows the same method in previous sections, so we omit the detailed constructions of all 22 defects. The defect Hamiltonian terms are formed by combining the short boundary string operators of the corresponding bosons in the given Lagrangian subgroup. In this case, we have verified that the topological order completion step is not required for these defect constructions.

Note that these 22 defects correspond to the 22 defects in the $\mathbb{Z}_4$ toric code. This correspondence arises from the fact that anyons in two copies of the six-semion code can be mapped to anyons in two copies of

the $\mathbb{Z}_4$ toric code, and vice versa, as follows:

$$
\begin{cases}
\tilde{e}_1 & \leftrightarrow e_1 m_1, \\
\tilde{m}_1 & \leftrightarrow e_1^2 m_1^3 e_2 m_2^3, \\
\tilde{e}_2 & \leftrightarrow e_2 m_2, \\
\tilde{m}_2 & \leftrightarrow e_1^3 m_1 e_2 m_2^2,
\end{cases}
\quad \text{or} \quad
\begin{cases}
\tilde{e}_1 \tilde{m}_1^3 \tilde{e}_2^3 \tilde{m}_2^2 & \leftrightarrow e_1, \\
\tilde{e}_1^3 \tilde{m}_1^2 \tilde{m}_2^3 & \leftrightarrow m_1, \\
\tilde{m}_1 \tilde{e}_2 \tilde{m}_2^2 & \leftrightarrow e_2, \\
\tilde{e}_1 \tilde{m}_1^2 \tilde{e}_2 \tilde{m}_2 & \leftrightarrow m_2.
\end{cases}
\tag{52}
$$

This mapping is consistent with the topological spins and braiding statistics of the $\mathbb{Z}_4$ toric code and the six-semion code in Eqs. (50) and (51).

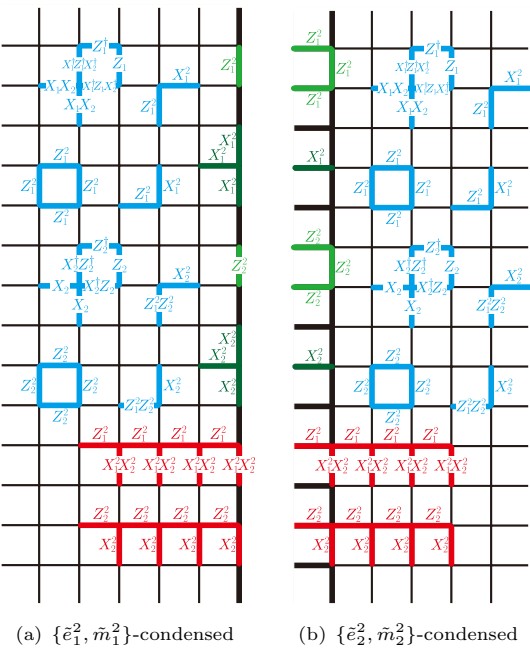

(a) $\{\tilde{e}_1^2, \tilde{m}_1^2\}$-condensed      (b) $\{\tilde{e}_2^2, \tilde{m}_2^2\}$-condensed

**Fig. 25** The boundaries of the $\mathbb{Z}_4$ six-semion code. Blue components represent the bulk stabilizers of the $\mathbb{Z}_4$ six-semion code, green components represent the boundary Hamiltonian, and red components represent bulk strings that terminate at the boundary without causing energy violations.

## 5.6 Color code

In this section, we examine the boundaries and defects of the color code on a honeycomb lattice, where each vertex hosts a single qubit. The honeycomb lattice is naturally embedded into a square lattice, with the vertices of the honeycomb lattice positioned on the edges of the square lattice, following the conventions established in Refs. [50, 109, 110]. The boundary gauge operators of the color code, which also function as short boundary string operators, are illustrated in Fig. 26. The boundary anyons are labeled as $e_1$, $m_1$, $e_2$, $m_2$ for the left boundary, and $e_3$, $m_3$, $e_4$, $m_4$ for the right boundary. The fusion rules are $e_i^2 = m_i^2 = 1$, $\forall i \in \{1, 2, 3, 4\}$, indicating that all have order 2. Their topological spins (Eq. (10)) and braiding statistics (Eq. (11)) are described by the following relations:

$$
\begin{aligned}
\theta(e_i) = \theta(m_i) = 1, \quad & B(e_i, m_i) = -1, \\
B(e_i, e_j) = B(e_i, m_j) = B(m_i, m_j) = 1, \quad & \forall i \neq j,
\end{aligned}
\tag{53}
$$

where $i, j \in 1, 2, 3, 4$. Note that each side corresponds to two copies of the $\mathbb{Z}_2$ toric code, as the color code can be interpreted as the "folded" toric code [21]. The boundaries of the color code exhibit 6 types of boundary anyon condensations, as illustrated in Fig. G8. These boundaries are equivalent to defects in the $\mathbb{Z}_2$ toric code by the folding argument. The color code itself allows for 270 distinct defect constructions, which is proved in Appendix E. A few examples of Lagrangian subgroups for these defect constructions are

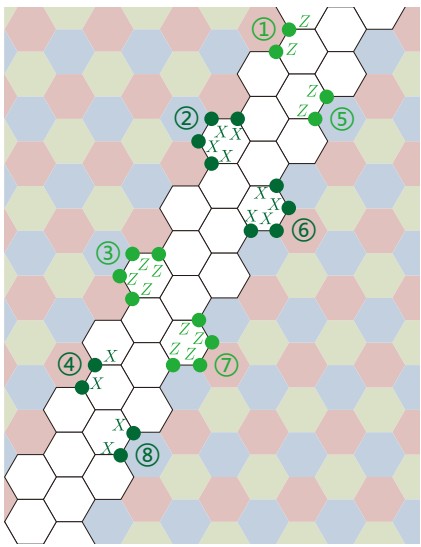

**Fig. 26** The stabilizers of the color code consist of products of Pauli $X$ or $Z$ operators acting on the vertices of each hexagon. The hexagons are colored red, yellow, and blue to distinguish the different types of anyons. A white region in the center represents a defect, with the boundary gauge operators on both sides depicted in green, exhibiting translational symmetry along the defect line. The boundary gauge operators labeled ① through ⑧ also serve as short boundary string operators. The corresponding boundary anyons are labeled $e_1$, $m_1$, $e_2$, $m_2$, $e_3$, $m_3$, $e_4$, and $m_4$, respectively.

listed below:

$$
\begin{aligned}
\mathcal{L}_1 &= \{e_1, e_2, e_3, e_4\}, \\
\mathcal{L}_2 &= \{e_1, e_2, m_1 e_4, e_3 m_4\}, \\
\mathcal{L}_3 &= \{m_1 e_3, e_1 m_1, m_2 m_4, e_2 e_4\}, \\
\mathcal{L}_4 &= \{m_1 e_3 e_4, e_1 e_2 e_3 m_4, m_2 e_3 e_4, m_1 e_3 m_4 e_4\}, \\
&\vdots
\end{aligned}
\tag{54}
$$

The detailed constructions of these Lagrangian subgroups are omitted for brevity. Computational verification confirms that topological order completion is not required for these 270 anyon condensations.

## 5.7 Anomalous three-fermion code

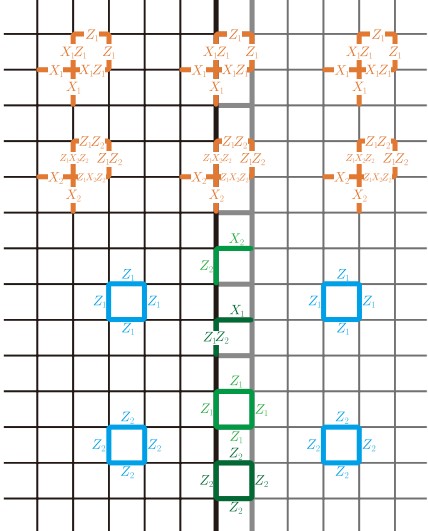

**Fig. 27** Three-fermion code. The orange components represent the gauge constraints, the blue components represent the stabilizers, and the green components represent the defect gauge operators that commute with all gauge constraints and stabilizers.

Our algorithm can also be applied to anomalous Pauli stabilizer codes, where the Hilbert space lacks a tensor product structure. For example, with two $\mathbb{Z}_4$ qudits on each edge, consider the three-fermion code:

$$G_1 = \begin{array}{c} \ulcorner Z_1 \urcorner \\ Z_1 \quad Z_1 \\ \llcorner Z_1 \lrcorner \end{array} = 1, \quad \mathcal{S}_1 = \begin{array}{c} \ulcorner Z_1 \urcorner \\ Z_1 \quad Z_1 \\ \llcorner Z_1 \lrcorner \end{array}, \quad G_2 = \begin{array}{c} \ulcorner Z_1 Z_2 \urcorner \\ Z_1 X_2 Z_2 \quad Z_1 Z_2 \\ X_2 \!-\!\! Z_1 X_2 Z_2 \\ X_2 \end{array} = 1, \quad \mathcal{S}_2 = \begin{array}{c} \ulcorner Z_2 \urcorner \\ Z_2 \quad Z_2 \\ \llcorner Z_2 \lrcorner \end{array}, \quad (55)$$

where $G_1 = 1$ and $G_2 = 1$ are gauge constraints on the Hilbert space of the three-fermion code, indicating that $G_1$ and $G_2$ cannot be violated, and we only consider operators that commute with them. Under these constraints, anyons are defined as violations of $\mathcal{S}_1$ and $\mathcal{S}_2$, which together form the three-fermion topological order [86, 95, 106, 111].

As this anomalous theory lies at the boundary of a (3+1)D invertible topological phase [112], it cannot support a further boundary, i.e., the boundary has no boundary. Therefore, we follow the procedure outlined in Sec. 4.2 to derive the defect gauge operators that commute with $G_1$, $G_2$, $\mathcal{S}_1$, and $\mathcal{S}_2$. Fig. 27 illustrates the gauge constraints $G_1$ and $G_2$, represented in orange, throughout the defect lattice, while the stabilizers $\mathcal{S}_1$ and $\mathcal{S}_2$ are only present away from the defect. The defect gauge operators are depicted in green. For clarity, the bulk anyons in Fig. 28 are labeled as $f_1^a$, $f_1^b$, and $f_1^c$ on the left side, and $f_2^a$, $f_2^b$, and $f_2^c$ on the right side. The fusion rules are $(f_j^a)^2 = (f_j^b)^2 = 1$, $\forall j \in \{1, 2\}$, indicating that all have order 2. Their topological spins (Eq. (10)) and braiding statistics (Eq. (11)) are given by:

$$\theta(f_j^a) = \theta(f_j^b) = -1, \quad B(f_j^a, f_j^b) = -1, \quad j \in \{1, 2\},$$
$$B(a_1, a_2) = 1, \quad \forall a_1 \in \{f_1^a, f_1^b\}, \ a_2 \in \{f_2^a, f_2^b\}, \quad (56)$$

where we have omitted the data for $f_j^c = f_j^a f_j^b$ as it can be generated from $f_j^a$ and $f_j^b$. According to the anyon theory, we can enumerate 6 distinct Lagrangian subgroups, with the corresponding defect constructions provided in Fig. G7 in Appendix G.



**Fig. 28** The bulk string operators are labeled with anyons, from top to bottom, as $f_1^a$, $f_1^b$, and $f_1^c = f_1^a f_1^b$ on the left, and $f_2^a$, $f_2^b$, and $f_2^c = f_2^a f_2^b$ on the right.

Note that these 6 defects are related to the 6 defects in the $\mathbb{Z}_2$ toric code. This is because anyons in two copies of the three-fermion code can be mapped to those in two copies of the $\mathbb{Z}_2$ toric code, and vice versa, as follows:

$$\begin{cases} f_1^a & \leftrightarrow e_1 m_1, \\ f_1^b & \leftrightarrow e_1 e_2 m_2, \\ f_2^a & \leftrightarrow e_2 m_2, \\ f_2^b & \leftrightarrow e_2 e_1 m_1, \end{cases} \quad \text{or} \quad \begin{cases} f_1^b f_2^a & \leftrightarrow e_1, \\ f_1^a f_1^b f_2^a & \leftrightarrow m_1, \\ f_1^a f_2^b & \leftrightarrow e_2, \\ f_1^a f_2^a f_2^b & \leftrightarrow m_2. \end{cases} \quad (57)$$

This mapping is consistent with the topological spins and braiding statistics of the $\mathbb{Z}_2$ toric code and the three-fermion code in Eqs. (49) and (56).

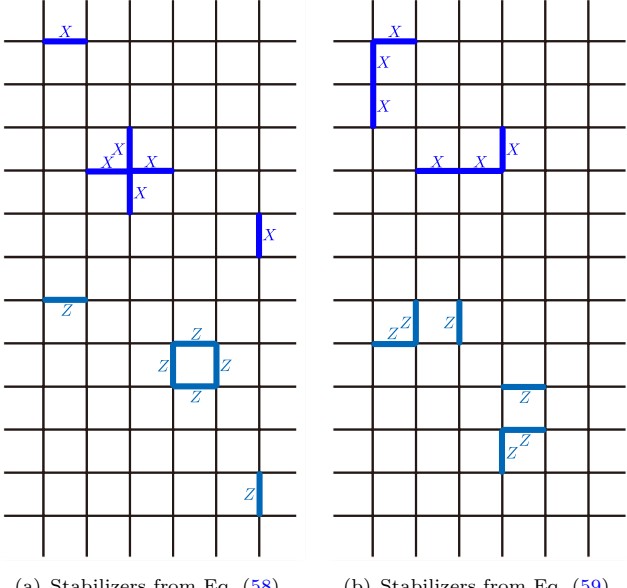

(a) Stabilizers from Eq. (58).     (b) Stabilizers from Eq. (59).

**Fig. 29** The stabilizer $S_1$ is depicted in light blue at the top, while $S_2$ is shown in dark blue at the bottom.

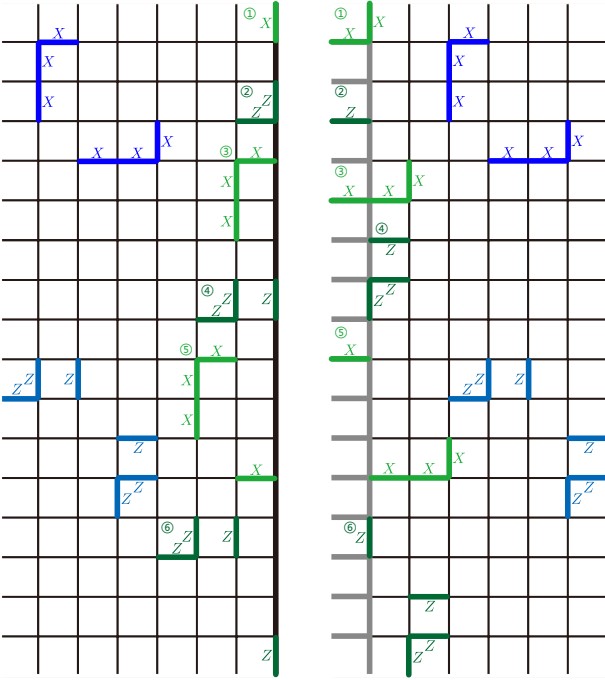

**Fig. 30** The left side illustrates a smooth boundary, while the right side illustrates a rough boundary. The blue components represent the bulk stabilizers of the (3, 3)-BB code in Eq. (59), and the green components represent the boundary gauge operators. The operator ① for the smooth boundary and the operator ② for the rough boundary are secondary boundary gauge operators that cannot be obtained through truncation.

## 5.8 (3,3)-bivariate bicycle codes

This section focuses on the $(3,3)$-bivariate bicycle (BB) code, which is part of the family of LDPC codes proposed in Ref. [26]. This code is represented as the $[[144, 12, 12]]$ code on a finite $12 \times 6$ torus, with its stabilizers expressed as follows:

$$
\mathcal{S}_1 = \begin{bmatrix} f \\ g \\ \hline 0 \\ 0 \end{bmatrix}, \mathcal{S}_2 = \begin{bmatrix} 0 \\ 0 \\ \hline \overline{g} \\ \overline{f} \end{bmatrix}, \quad \begin{cases} f = y^2(x + x^2 + y^b), \\ g = x^2(y + y^2 + x^a), \end{cases}
\tag{58}
$$

where $a = b = 3$, and we are working with $\mathbb{Z}_2$ qubits, making the minus signs irrelevant. Since we have the freedom to redefine the generators of the Pauli operators, we can shift $f$ and $g$ by arbitrary polynomials. For instance, we can use alternative stabilizers given by:

$$\mathcal{S}_1' = \begin{bmatrix} f' \\ g' \\ \hline 0 \\ 0 \end{bmatrix}, \mathcal{S}_2' = \begin{bmatrix} 0 \\ 0 \\ \hline \overline{g'} \\ \overline{f'} \end{bmatrix}, \quad \begin{cases} f' = x + x^2 + y^b, \\ g' = y + y^2 + x^a, \end{cases} \tag{59}$$

with $a = b = 3$. The stabilizers in Eqs. (58) and (59) are illustrated in Fig. 29. Notably, we can shift the horizontal edges two steps down and the vertical edges two steps to the left to transform the stabilizers from Eq. (58) to Eq. (59). Under periodic boundary conditions, these two expressions of stabilizers are equivalent. However, with open boundary conditions, their boundary constructions must be examined separately, as there is no straightforward correspondence achieved by simply shifting the edges. We will analyze both expressions, as Eq. (59) is more compact locally and may be more practical for experimental implementation. We refer to Eq. (59), with general integers $a$ and $b$, as the $(a, b)$-**BB code**.

We apply our algorithm to compute the boundary gauge operators for these stabilizers. For the bulk stabilizers in Eq. (58), the boundary gauge operators consist solely of primary ones, meaning they can all be obtained as truncated stabilizers. In contrast, for the bulk stabilizers in Eq. (59), there are secondary boundary gauge operators, as illustrated in Fig. 30.

Using these boundary gauge operators, the number of boundary anyon generators for string lengths from 1 to 12 is given by:

| string length | 1 | 2 | 3 | 4 | 5 | 6 | 7 | 8 | 9 | 10 | 11 | 12 |
|---|---|---|---|---|---|---|---|---|---|---|---|---|
| # of generators | 0 | 0 | 8 | 0 | 0 | 12 | 0 | 0 | 8 | 0 | 0 | 16 |

We have verified string lengths up to thousands to ensure all anyons are found. Notably, there are 16 generators of boundary string operators, as illustrated in Fig. G9 in Appendix G, each with a string length of 12. When rearranged, the boundary anyons correspond to 8 copies of toric codes. In Ref. [26], the $(3, 3)$-BB code is placed on a $6 \times 12$ torus, resulting in a $[[144, 12, 12]]$ qLDPC code. The logical dimension is 12 because, with a period of 6 in the $x$-direction, only 12 of the 16 boundary anyon generators remain. When the $(3, 3)$-BB code is placed on a $12 \times 12$ torus, it utilize all anyons, resulting in a $[[288, 16, 12]]$ qLDPC code. More generally, on a torus of size $12L \times 12L$, the code parameters can be compared with those of the standard $\mathbb{Z}_2$ toric code as follows:

| Codes on a $12L \times 12L$ torus | $[[n, k, d]]$ |
|---|---|
| $(3, 3)$-BB code | $[[288L^2, 16, 12L]]$ |
| Standard $\mathbb{Z}_2$ toric code | $[[288L^2, 2, 12L]]$ |

(60)

where both codes exhibit similar scaling with $L$, but the logical space of the $(3, 3)$-BB code is eight times larger than that of the toric code.

Moreover, to make the experimental realization of the $(3, 3)$-BB code more feasible, one could choose the open boundary condition, i.e., using a $12L \times 12L$ square instead of a torus. For the most straightforward architecture of the $(3, 3)$-**BB surface code**, we impose $\{e_i\}$-condensed boundaries on the top and bottom edges of the square, while $\{m_i\}$-condensed boundaries are imposed on the left and right edges. Due to the presence of secondary boundary gauge operators, following the anyon condensation, it is necessary to perform the "topological order completion" by adding additional boundary terms to the Hamiltonian.

In this $(3, 3)$-BB surface code, the logical operators include vertical $\{e_i\}$ strings and horizontal $\{m_i\}$ strings, resulting in the number of logical qubits being reduced by half. We expect the $(3, 3)$-BB surface code to have the following parameters:

| Codes on a $12L \times 12L$ square | $[[n, k, d]]$ |
|---|---|
| $(3, 3)$-BB code | $[[288L^2, 8, 12L - O(1)]]$ |
| Standard $\mathbb{Z}_2$ toric code | $[[288L^2, 1, 12L]]$ |

(61)

where the code distance of the $(3, 3)$-BB surface code is affected by the secondary boundary terms added to the boundary and decreases by some constant. It is important to emphasize that the boundary string operators added to the boundary Hamiltonian have a length of 12, making the boundary stabilizers slightly

more non-local than those in the bulk (though they remain local as their length does not scale with $L$). An interesting direction for future research would be to explore whether the boundary terms can be made more local and retain the code distance precisely at $12L$ without subtracting a constant.

## 5.9 (2,-3)-bivariate bicycle codes

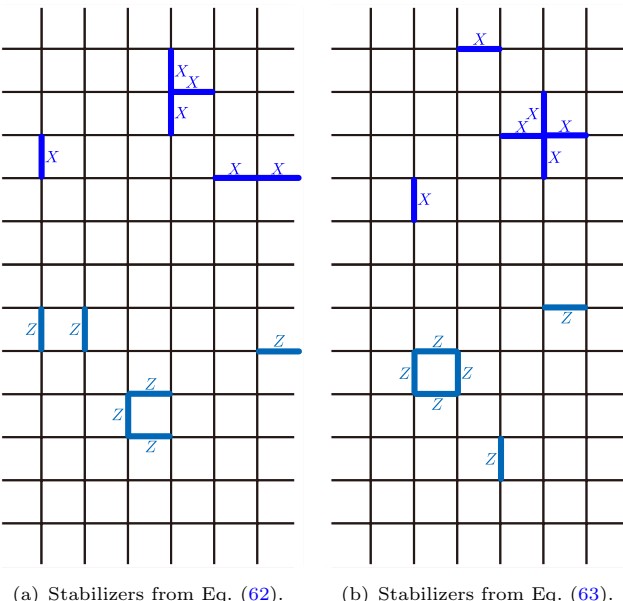

(a) Stabilizers from Eq. (62).          (b) Stabilizers from Eq. (63).

**Fig. 31** The stabilizer $S_1$ is depicted in light blue at the top, while $S_2$ is shown in dark blue at the bottom.

The stabilizers of the $\mathbb{Z}_2$ (2,-3)-bivariate bicycle code are given by the following expressions:

$$
\mathcal{S}_1 = \begin{bmatrix} h \\ k \\ 0 \\ 0 \end{bmatrix}, \ \mathcal{S}_2 = \begin{bmatrix} 0 \\ 0 \\ \overline{h} \\ \overline{k} \end{bmatrix}, \quad \begin{cases} h = x + x^2 + y^2, \\ k = y + y^2 + x^{-3}. \end{cases} \tag{62}
$$

Alternatively, they can also be written as:

$$
\mathcal{S}_1' = \begin{bmatrix} h' \\ k' \\ 0 \\ 0 \end{bmatrix}, \ \mathcal{S}_2' = \begin{bmatrix} 0 \\ 0 \\ \overline{h'} \\ \overline{k'} \end{bmatrix}, \quad \begin{cases} h' = y^2(x + x^2 + y^2), \\ k' = x^2(y + y^2 + x^{-3}), \end{cases} \tag{63}
$$

Both sets of stabilizers are depicted in Fig. 31. On a periodic lattice, such as a finite torus, the stabilizers in Eqs. (62) and (63) are equivalent up to edge shifts. However, when considering open boundary conditions, these two cases are treated separately, similar to the distinction between Eqs. (58) and (59) from the previous section.

From these stabilizers, we compute the boundary gauge operators. We have verified that, for both smooth and rough boundaries, only primary boundary gauge operators are present for the bulk stabilizers in Eqs. (62) or (63).[19]

Unlike the bulk stabilizers, which possess translational symmetry in two dimensions, the boundary gauge operators exhibit translational symmetry in only one dimension, facilitating more efficient computation of the anyons. Consequently, we verified that at a string length of 1023, there are 20 generators of boundary string operators corresponding to the anyons in 10 copies of the toric codes. The bulk anyons also exhibit a periodicity of 1023 due to the bulk-boundary correspondence established in Theorem 4. This long periodicity of anyons is a typical feature of the BB code family.

---

[19]A subtle point arises for the boundary: certain qubits on specific edges are not acted upon by any bulk stabilizers, so these edges are excluded when computing the boundary gauge operators.

# 6 Discussion

This work improves upon the method in Ref. [50] by introducing an operator algebra formalism on a truncated lattice, offering a rigorous framework for proving key lemmas and theorems. We develop algorithms to extract boundary gauge and string operators from truncated Pauli stabilizer codes, enabling the calculation of topological spin, braiding statistics, and Lagrangian subgroups for constructing boundaries and defects. The algorithm applies to both prime and non-prime dimensional qudit stabilizer codes and has been tested on various examples, demonstrating its effectiveness and versatility. Our method extends to lattices with different codes on either side, enabling the construction of defects that bridge and modify bulk anyons. Using bulk-boundary correspondence, we infer bulk properties from the (1+1)D boundary, accelerating the computation of topological data in the (2+1)D bulk, such as identifying anyons and their string operators. For instance, in the (2,-3)-bivariate bicycle (BB) codes, we derive 16 basis anyons with period-1023 string operators at a speed ten times faster than bulk computation using the method in Ref. [50]. Our approach offers an efficient tool for deriving topological data in Pauli stabilizer codes. A promising future direction is optimizing our algorithm to handle stabilizers with larger weights, which is crucial for studying BB codes, where stabilizers typically involve large structures. Recent studies have focused on the properties of BB codes [26, 27, 113–123], making this optimization especially relevant.

Beyond Pauli stabilizer codes, extensions like the XS [124] and XP [125, 126] formalisms modify the stabilizer framework by incorporating roots of the Pauli $Z$ operator. Since XP stabilizer codes also have a symplectic representation, extending the polynomial formalism and our algorithm to these codes is feasible. This represents a crucial step toward exploring non-Pauli or non-Clifford stabilizer codes, which could exhibit non-Abelian anyon statistics—an essential feature for universal topological quantum computation.

Another potential extension of our polynomial formalism involves generalizing the $\mathbb{Z}^2$ translational symmetry to more general, possibly non-Abelian, groups. Currently, we focus on Pauli stabilizer codes on two-dimensional lattices with $\mathbb{Z}^2$ symmetry, represented by the Laurent polynomial generators $x$ and $y$. We aim to extend this to more complex graphs with arbitrary translation groups $G$, where the generators are labeled $g_1, g_2, g_3$, etc. This generalization would still use a "polynomial" ring over these generators, enabling the representation of Pauli stabilizer codes on the Cayley graph of $G$. This approach, widely used in constructing quantum low-density parity-check (qLDPC) codes [127–131], will guide the adaptation of our algorithm, providing new insights into advanced qLDPC codes.

In addition, we are exploring a (3+1)D generalization of our algorithm. A new protocol would be required to detect loop excitations. One approach involves using dimensional reduction by compactifying one dimension of the (3+1)D system, transforming it into a quasi-2D system [132]. This technique would enable efficient detection of both particle and compactified loop excitations. Also, we can study the (2+1)D anomalous boundaries of (3+1)D Pauli stabilizer codes, such as the three-fermion code at the boundary of the Walker-Wang model [133], to investigate chiral boundary theories.

Furthermore, an additional generalization involves subsystem codes [95, 134–136] and Floquet codes [137–141]. Subsystem codes relax the need for all Hamiltonian terms to commute, allowing non-commuting terms to act as gauge operators—an approach we have used to describe boundary theories in this paper. Floquet codes add further complexity by leveraging the temporal sequence of measurements, where each cycle generates an instantaneous stabilizer code, such as the toric code derived from the honeycomb model [137]. Applying our algorithm to a broader range of subsystem and Floquet codes could provide deeper insights into these dynamic systems.

# Acknowledgement

Y.-A.C. would like to thank Nathanan Tantivasadakarn for insightful discussions on the holographic principle for stabilizer codes and for suggesting the name "fish toric code." Y.-A.C. also thanks Tyler Ellison for clarifying subtleties regarding anyons in subsystem codes and appreciates valuable conversations with Po-Shen Hsin, Sri Tata, and Shu-Heng Shao on Lagrangian subgroups. Additionally, Y.-A.C. is grateful to Ke Liu and Hao Song for highlighting the applications of bivariate bicycle codes. B.Y. would like to express heartfelt gratitude to Blazej Ruba for the many insightful hours of discussion that have greatly enriched our perspective. We also express appreciation to Andreas Bauer, Ryohei Kobayashi, Anton Kapustin, and Yijia Xu for their valuable discussions and feedback.

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

# Appendix A Topological spins and braiding statistics from boundary string operators

In this appendix, we will prove Theorems 5 and 6. We begin by showing that the expression for the boundary topological spin in Eq. (10),

$$\theta(a) = [U(a)_{1\to2}, U(a)_{2\to3}] := U(a)_{1\to2}U(a)_{2\to3}U(a)_{1\to2}^{\dagger}U(a)_{2\to3}^{\dagger}, \tag{A1}$$

is a topological invariant, meaning it is independent of the specific choice of $U(a)_{1\to2}$ and $U(a)_{2\to3}$. Next, we will demonstrate that this topological invariant corresponds to the topological spin obtained from the T-junction process for bulk anyons. Finally, we will apply the same concept to derive the braiding statistics between boundary anyons.

## A.1 Topological invariant

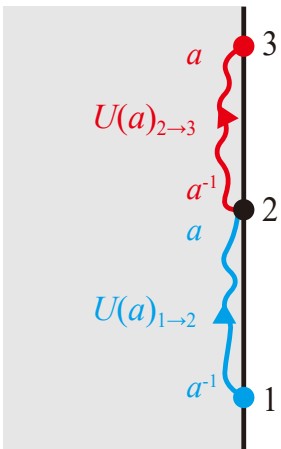

Given the boundary string operators $U(a)_{1\to 2}$ and $U(a)_{2\to 3}$, which move the boundary anyon $a$ from vertex 1 to 2 and from vertex 2 to 3, respectively, as illustrated in the above figure, we first show that Eq. (A1) is independent of the specific choices of $U(a)$. For simplicity, we denote $U(a)_{1\to 2}$ and $U(a)_{2\to 3}$ as $U_1$ and $U_2$.

Since we have the freedom to redefine the boundary anyon by applying local boundary gauge operators, we can "dress" the string operators with some boundary term $\mathcal{G}$ near their endpoints, resulting in new boundary string operators:

$$U_1 \to \widetilde{U_1} = \mathcal{G}_2 U_1 \mathcal{G}_1^\dagger, \quad U_2 \to \widetilde{U_2} = \mathcal{G}_3 U_2 \mathcal{G}_2^\dagger. \tag{A2}$$

As a result, a string operator from point 1 to 3 transforms as follows:

$$U_2 U_1 \to \mathcal{G}_3 U_2 \mathcal{G}_2^\dagger \mathcal{G}_2 U_1 \mathcal{G}_1^\dagger = \mathcal{G}_3 U_2 U_1 \mathcal{G}_1^\dagger = \widetilde{U_2}\widetilde{U_1}. \tag{A3}$$

Thus, we see that the transformation preserves the overall structure of the string operator, with the redefinition occurring only at the endpoints.

Before we proceed to compute the commutator between the new $U_1$ and $U_2$, we need to mention a fact: $[\mathcal{G}_2, U_2 U_1] = 0$. In other words, the boundary term $\mathcal{G}_2$ at point 2 commutes with the longer string operator from point 1 to point 3. This follows from the definition of boundary string operators, which commute with all boundary terms in their middle parts and only violate something near their endpoints.

We now compute the commutator between the new string operators $\widetilde{U_1}$ and $\widetilde{U_2}$:

$$\begin{aligned}
\widetilde{U_1}\widetilde{U_2}\widetilde{U_1}^\dagger\widetilde{U_2}^\dagger &= (\mathcal{G}_2 U_1 \mathcal{G}_1^\dagger)(\mathcal{G}_3 U_2 \mathcal{G}_2^\dagger)(\mathcal{G}_1 U_1^\dagger \mathcal{G}_2^\dagger)(\mathcal{G}_2 U_2^\dagger \mathcal{G}_3^\dagger) \\
&= \mathcal{G}_2 U_1 \mathcal{G}_1^\dagger \mathcal{G}_3 U_2 \mathcal{G}_2^\dagger \mathcal{G}_1 U_1^\dagger U_2^\dagger \mathcal{G}_3^\dagger \\
&= \mathcal{G}_2 U_1 \mathcal{G}_1^\dagger \chi U_2 \mathcal{G}_3 \mathcal{G}_2^\dagger \mathcal{G}_1 U_1^\dagger \chi^{-1} \mathcal{G}_3^\dagger U_2^\dagger,
\end{aligned}$$

where we have used the relation $\mathcal{G}_3 U_2 = \chi U_2 \mathcal{G}_3$, where $\chi \in U(1)$, given that both $\mathcal{G}$ and $U$ are Pauli operators. Since $U_1$, $\mathcal{G}_1$, and $\mathcal{G}_2$ are far from $\mathcal{G}_3$, they commute with $\mathcal{G}_3$. Similarly, $\mathcal{G}_1$ commutes with both $U_2$ and $\mathcal{G}_2$. Therefore, we can simplify the expression as follows:

$$\begin{aligned}
\widetilde{U_1}\widetilde{U_2}\widetilde{U_1}^\dagger\widetilde{U_2}^\dagger &= \mathcal{G}_2 U_1 \mathcal{G}_1^\dagger U_2 \mathcal{G}_2^\dagger \mathcal{G}_1 U_1^\dagger U_2^\dagger \\
&= \mathcal{G}_2 U_1 U_2 \mathcal{G}_2^\dagger U_1^\dagger U_2^\dagger \\
&= U_1 U_2 U_1^\dagger U_2^\dagger.
\end{aligned} \tag{A4}$$

The commutator remains unchanged by the redefinition of the string operators, confirming that the expression is independent of the specific choice of $U(a)$.

## A.2 Topological spin

Next, we will demonstrate that this topological invariant corresponds to the topological spin of the anyon. As shown in Fig. A1, the boundary string operators $U(a)_{1\to 2}$ and $U(a)_{2\to 3}$ move the boundary anyon $a$ from vertex 1 to vertex 2 and from vertex 2 to vertex 3, respectively (as shown in the figure above). Additionally, the bulk string operator $U(a)_{4\to 2}$ moves the bulk anyon $a$ to the boundary.

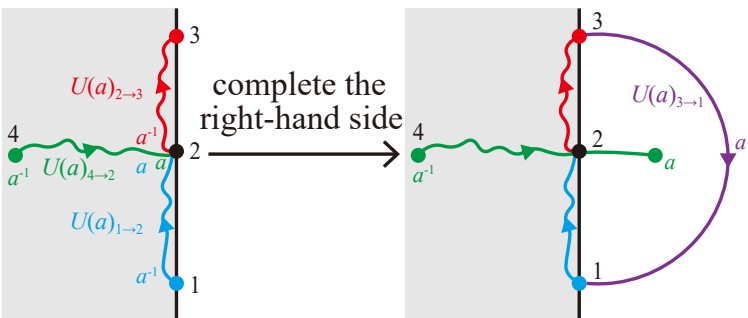

**Fig. A1** The boundary string operators $U(a)_{1\to2}$ and $U(a)_{2\to3}$ move boundary anyon $a$, while $U(a)_{4\to2}$ represents a bulk string operator terminating at the boundary, also creating the boundary anyon $a$. We first embed the truncated system into the original infinite plane by completing the right-hand side, transforming $U(a)_{1\to2}$, $U(a)_{2\to3}$, and $U(a)_{4\to2}$ into bulk string operators in the completed system. Subsequently, we slightly extend $U(a)_{4\to2}$ to the right, represented by the green string, and append a string operator along the semi-circle (purple) to $U(a)_{2\to3}U(a)_{1\to2}$ to form a closed loop.

For simplicity, we will refer to these string operators as $U_1 = U(a)_{1\to2}$, $U_2 = U(a)_{2\to3}$, and $U_3 = U(a)_{4\to2}$. When the right-hand side of the system is restored (recalling that the open system is a truncated version of an infinite system), these string operators are embedded as bulk string operators within the complete system. In this completed configuration, we can apply the T-junction process (6) to compute the topological spin of the anyon $a$:[20]

$$\theta(a) = U_3^\dagger U_2^\dagger U_1^\dagger U_3 U_2 U_1. \tag{A5}$$

Since $U_1$, $U_2$, and $U_3$ are Pauli operators, their commutators result in $U(1)$ phases. Using the relation $[U_i^\dagger, U_j] = [U_i, U_j]^{-1}$, we can express the topological spin as:

$$\theta(a) = [U_3, U_2] \times [U_3, U_1] \times [U_2, U_1]. \tag{A6}$$

This shows that the commutation relations between these string operators determine the topological spin.

Now, we study the two commutators $[U_3, U_2]$ and $[U_3, U_1]$. Using the fact that $U$ represents Pauli operators, we find that their product is equal to $[U_3, U_2U_1]$. We can gain insight into the commutation between $U_3$ and $U_2U_1$ in the completed system. We extend $U_3 = U(a)_{4\to2}$ slightly into the bulk (green string in Fig. A1) and complete $U_2U_1 = U(a)_{2\to3}U(a)_{1\to2}$ by adding a bulk string operator $U(a)_{3\to1}$ (purple semi-circle in Fig. A1). Importantly, the extended semi-circle does not affect the commutation, as it is spatially distant from the extended $U_3$. We define the extension of $U_3$ as $U_3 O_{\text{LHS}} O_{\text{RHS}}$, where $O_{\text{LHS}}$ and $O_{\text{RHS}}$ are Pauli operators fully supported in the LHS and RHS, respectively. Since the middle part of the extended green string still commutes with the bulk stabilizers in the LHS, $O_{\text{LHS}}$ must be a boundary gauge operator. Consequently, by definition, $O_{\text{LHS}}$ commutes with the boundary string operator $U_2U_1$. Furthermore, it is evident that $O_{\text{RHS}}$ also commutes with $U_2U_1$, as they do not overlap. Thus, we conclude that the extensions of $U_3$ and $U_2U_1$ do not affect their commutation relation.

This transforms the commutator $[U_3, U_2U_1]$ into the full (counter-clockwise) braiding of the anyon $a^{-1}$ in the bulk:

$$[U_3, U_2U_1] = U_3(U_2U_1)U_3^\dagger(U_2U_1)^\dagger = \theta(a)^2. \tag{A7}$$

Thus, we obtain

$$[U_2, U_1] = \theta(a)^{-1}, \tag{A8}$$

or more precisely,

$$\theta(a) = [U(a)_{(1\to2)}, U(a)_{(2\to3)}]. \tag{A9}$$

This expression captures the topological spin of anyon $a$ from the boundary string operators.

## A.3 Braiding statistics

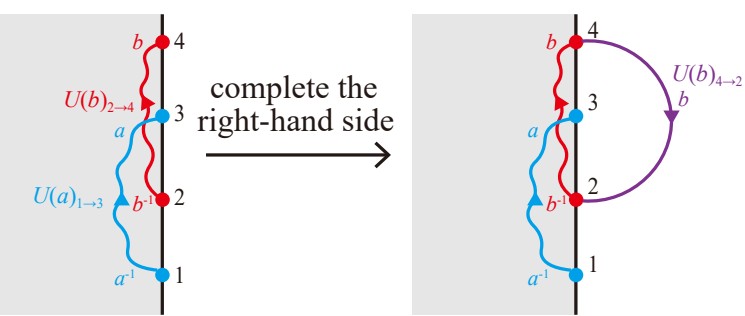

---

[20]The orientation of $U_2$ is reversed compared to the setup in Eq. (6), so $W_2^\dagger$ is replaced with $U_2$.

To analyze the braiding process between boundary anyons, we slightly modify the setup by defining the boundary string operators $U(a)_{1\to3}$ and $U(b)_{2\to4}$, which move boundary anyon $a$ from vertex 1 to 3 and anyon $b$ from vertex 2 to 4 along the boundary, as illustrated in the figure above. After restoring the right-hand side, we append the boundary operator $U(b)_{2\to4}$ with a bulk string operator $U(b)_{4\to2}$, which moves the bulk anyon $b$ from vertex 4 back to 2 along the semi-circle.

Following the same intuition used in deriving the topological spin, we can express the full braiding between anyons $a$ and $b$ as:

$$B(a,b) = U(a)_{1\to3}(U(b)_{2\to4}U(b)_{4\to2})U(a)_{1\to3}^{\dagger}(U(b)_{2\to4}U(b)_{4\to2})^{\dagger}. \tag{A10}$$

Since $U(b)_{4\to2}$ and $U(a)_{1\to3}$ are spatially separated, they commute with each other. Therefore, the braiding statistics of boundary anyons can be computed as:

$$B(a,b) = U(a)_{1\to3}U(b)_{2\to4}U(a)_{1\to3}^{\dagger}U(b)_{2\to4}^{\dagger}. \tag{A11}$$

# Appendix B    Units in the formal Laurent series

In this section, we derive the necessary and sufficient condition for an element in the formal Laurent series $\mathbb{Z}_d((x))$ to be a unit.

First, the **formal Laurent series** $\mathbb{Z}_d((x))$ is defined by the polynomials of $x$ and $x^{-1}$, and the degree of $x$ is allowed to be positive infinite. Any element $f(x) \in \mathbb{Z}_d((x))$ can be written as

$$f(x) = \sum_{i=-k}^{\infty} a_i x^i, \tag{B12}$$

with $k \in \mathbb{Z}$ and $a_i \in \mathbb{Z}_d$. The formal Laurent series $\mathbb{Z}_d((x))$ forms a ring since the multiplication of the polynomials can be defined in the standard way. A **unit** is an element in the ring with a multiplicative inverse.

**Lemma 18.** *An element in the formal Laurent series $f(x) = \sum_{i=-k}^{\infty} a_i x^i \in \mathbb{Z}_d((x))$ with $a_i \in \mathbb{Z}_d$ is a unit if and only if the ideal generated by $\{a_i\}$ is $\mathbb{Z}_d$. Equivalently, there exist coefficients $\{m_i\}$ with each $m_i \in \mathbb{Z}_d$ and an integer $n$ such that*

$$\sum_{i=-k}^{n} m_i a_i \equiv 1 \pmod{d}. \tag{B13}$$

*Proof.* We first show that Eq. (B13) is necessary for $f(x)$ to be a unit in $\mathbb{Z}_d((x))$. If the greatest common divisor (gcd) of the coefficients $\{a_{-k}, a_{-k+1}, \ldots\}$ is greater than 1, then every multiple of $f(x)$ will have coefficients divisible by this gcd, which implies that $f(x)$ cannot have an inverse in $\mathbb{Z}_d((x))$. Therefore, Eq. (B13) is necessary for $f(x)$ to be invertible.

To demonstrate that the condition in Eq. (B13) is sufficient, we begin by considering the primary decomposition of the ring $\mathbb{Z}_d$, which can be expressed as a product of rings corresponding to the prime power factors of $d$:

$$\mathbb{Z}_d \cong \mathbb{Z}_{p_1^{k_1}} \times \mathbb{Z}_{p_2^{k_2}} \times \cdots \times \mathbb{Z}_{p_r^{k_r}}, \tag{B14}$$

where $p_i$ are the distinct prime factors of $d$ and $k_i$ are the corresponding exponents in the factorization of $d$. By the Chinese remainder theorem, $f(x)$ has an inverse in $\mathbb{Z}_d((x))$ if and only if it has an inverse in each $\mathbb{Z}_{p_i^{k_i}}((x))$. More concretely, if $f(x)$ has an inverse $f(x)_{k_i}^{-1}$ modulo $p_i^{k_i}$ for each $i$, then by the Chinese remainder theorem, there exists a unique solution $I(x)$ modulo $d$ that satisfies:

$$\begin{aligned}
I(x) &\equiv f(x)_{k_1}^{-1} \pmod{p_1^{k_1}}, \\
I(x) &\equiv f(x)_{k_2}^{-1} \pmod{p_2^{k_2}}, \\
&\vdots \\
I(x) &\equiv f(x)_{k_r}^{-1} \pmod{p_r^{k_r}}.
\end{aligned} \tag{B15}$$

Multiplying both sides by $f(x)$ and applying the Chinese remainder theorem again, we obtain:

$$f(x)I(x) \equiv 1 \pmod{d}. \tag{B16}$$

Thus, it suffices to show that $f(x)$ has an inverse in each $\mathbb{Z}_{p_i^{k_i}}((x))$.

Without loss of generality, consider $d = p^r$ where $p$ is a prime. If the gcd of the coefficients $\{a_{-k}, a_{-k+1}, \ldots\}$ is 1, then not all $a_i$ are divisible by $p$. This implies that when we reduce modulo $p$, the image $\bar{f}(x) \in \mathbb{Z}_p((x))$ is nonzero. Since $\mathbb{Z}_p((x))$ is a field of Laurent series over the finite field $\mathbb{Z}_p$, $\bar{f}(x)$ is invertible in $\mathbb{Z}_p((x))$. Let $\bar{g}(x) \in \mathbb{Z}_p((x))$ be the inverse of $\bar{f}(x)$. We then lift $\bar{g}(x)$ to some $g(x) \in \mathbb{Z}_{p^r}((x))$ such that:

$$f(x)g(x) = 1 + h(x), \tag{B17}$$

where $h(x) \in (p)$, meaning that all coefficients of $h(x)$ are divisible by $p$.

Since $h(x)$ is nilpotent (i.e., some power $h(x)^m = 0$ in $\mathbb{Z}_{p^r}((x))$ because $p^r = 0$ in $\mathbb{Z}_{p^r}$), the element $1 + h(x)$ has an inverse in $\mathbb{Z}_{p^r}((x))$, denoted $(1 + h(x))^{-1}$ (see Proposition 1.9 in Ref. [142]). Thus, we define the inverse of $f(x)$ as:

$$I(x) = g(x)(1 + h(x))^{-1}. \tag{B18}$$

This inverse satisfies:

$$f(x)I(x) \equiv 1 \pmod{d}. \tag{B19}$$

Hence, $I(x)$ is indeed the inverse of $f(x)$ in $\mathbb{Z}_d((x))$.

Therefore, the condition in Eq. (B13) is both necessary and sufficient for $f(x)$ to be a unit in $\mathbb{Z}_d((x))$. $\quad\square$

In the proof above, we only prove the existence of the inverse but do not construct it explicitly. For practical purposes, we would like to know how to obtain the inverse precisely, which can be used to construct the (semi-infinite) boundary string operators for boundary anyons. Now, we are going to provide an alternative constructive proof for the inverse of $f(x)$ in $\mathbb{Z}_d((x))$ with $d = p^k$.

**Lemma 19.** *Let $f(x) \in \mathbb{Z}_d((x))$, where $d = p^k$ for some prime $p$ and integer $k \geq 1$. If at least one coefficient of $f(x)$ is not divisible by $p$, then $f(x)$ is invertible in $\mathbb{Z}_d((x))$.*

*Proof.* We will prove this lemma by induction on $k$.

For $k = 1$, $\mathbb{Z}_d((x)) = \mathbb{Z}_p((x))$, which is a field. If at least one coefficient of $f(x)$ is not divisible by $p$, then $f(x) \neq 0$ in $\mathbb{Z}_p((x))$. Since $\mathbb{Z}_p((x))$ is a field, any non-zero element has an inverse, which can be computed by recursively solving for its coefficients in the Laurent series. Therefore, $f(x)$ is invertible in $\mathbb{Z}_p((x))$, and the lemma holds for $k = 1$.

Assume that the lemma holds for all $k \leq k_0$, where $k_0 \geq 1$. We need to show that it also holds for $k = k_0 + 1$. Consider $f(x)$ in $\mathbb{Z}_{p^{k_0+1}}((x))$. We can express $f(x)$ as a formal Laurent series:

$$f(x) = \sum_{n=-N_0}^{\infty} a_n x^n. \tag{B20}$$

Now, focus on the terms where the coefficients are not divisible by $p$. Define a series $g(x)$ in $\mathbb{Z}_p((x))$ using only those coefficients $a_n$ of $f(x)$ that are not divisible by $p$. Since $g(x)$ has at least one coefficient not divisible by $p$, there exists an element $h(x)$ in $\mathbb{Z}_p((x))$ such that $g(x)h(x) = 1$ in $\mathbb{Z}_p((x))$. This implies that $f(x)$ and $h(x)$ satisfy $f(x)h(x) = 1$ modulo $p$.

Multiplying through by $p^{k_0}$, we get:

$$p^{k_0} f(x)h(x) = p^{k_0} \pmod{p^{k_0+1}}, \tag{B21}$$

where $h(x)$ has been lifted to $\mathbb{Z}_d((x))$. This shows that $f(x)$ can generate $p^{k_0}$ modulo $p^{k_0+1}$. By the induction hypothesis, there exists an inverse $f(x)_{k_0}^{-1}$ in $\mathbb{Z}_{p^{k_0}}((x))$ such that

$$f(x)f(x)_{k_0}^{-1} = 1 \pmod{p^{k_0}}. \tag{B22}$$

We can express this as

$$f(x)f(x)_{k_0}^{-1} = 1 + p^{k_0}\alpha(x), \tag{B23}$$

where $\alpha(x) \in \mathbb{Z}((x))$. Notice that the term $p^{k_0}\alpha(x)$ represents a correction needed for lifting to $\mathbb{Z}_{p^{k_0+1}}((x))$, and since $f(x)$ can generate $p^{k_0}$ modulo $p^{k_0+1}$ as shown earlier, we can adjust $f(x)_{k_0}^{-1}$ to construct the inverse of $f(x)$ in $\mathbb{Z}_{p^{k_0+1}}((x))$. Specifically, consider the modified inverse:

$$f(x)_{k_0+1}^{-1} = f(x)_{k_0}^{-1} - \alpha(x)p^{k_0}h(x). \tag{B24}$$

Then, we have:

$$f(x)f(x)_{k_0+1}^{-1} = 1 \pmod{p^{k_0+1}}. \tag{B25}$$

This demonstrates that $f(x)$ has an inverse in $\mathbb{Z}_{p^{k_0+1}}((x))$. By the principle of induction, the lemma holds for all $k \geq 1$. $\quad\square$

# Appendix C   Modified Gaussian elimination (MGE)

This appendix reviews the modified Gaussian elimination method introduced in Ref. [50]. Standard Gaussian elimination is not applicable over the ring $\mathbb{Z}_d$ because the multiplicative inverse of an element may not exist; for example, the element 2 in $\mathbb{Z}_4$ does not have an inverse. Furthermore, $\mathbb{Z}_d$ can contain zero divisors, meaning there exist nonzero elements $a$ and $x \in \mathbb{Z}_d$ such that $ax = 0$. For instance, in $\mathbb{Z}_4$, $2 \times 2 = 0$, indicating that 2 is a zero divisor.

Therefore, we introduce the modified Gaussian elimination algorithm over $\mathbb{Z}_d$, which is based on the Hermite normal form:

1. Given a $n \times m$ matrix $A$ over $\mathbb{Z}_d$, we treat the entries in the first column $a_{i,1}$ for all $i \in \{1, 2, \cdots, n\}$ as integers $\{0, 1, 2, \ldots, d-1\}$ in $\mathbb{Z}$. To restore the $\mathbb{Z}_d$ periodicity, we append a new row $[d, 0, 0, \cdots, 0]$ to the bottom of matrix $A$, transforming it into a $(n+1) \times m$ matrix, denoted as $A'$. Next, we find the greatest common divisor of its first column $\{a_{1,1}, a_{2,1}, \ldots, a_{n,1}, d\}$, denoted as $\gcd(a_{i,1})_d$.[21] From the extended Euclidean algorithm, there exists a linear combination:

$$r_1 a_{1,1} + r_2 a_{2,1} + \cdots + r_n a_{n,1} + r_0 d = \gcd(a_{i,1})_d. \tag{C26}$$

Moreover, this linear combination can be obtained by repeatedly subtracting one entry from another entry, starting from

$$[a_{1,1}, a_{2,1}, a_{3,1}, \ldots, a_{n,1}, d], \tag{C27}$$

and obtaining the final form

$$[\gcd(a_{i,1})_d, 0, 0, \ldots, 0, 0]. \tag{C28}$$

Subtracting one entry from another and reordering corresponds to row operations in the matrix $A'$. Therefore, we apply the corresponding row operations in the matrix $A'$ according to the extended Euclidean algorithm, which transforms the first column into

$$[\gcd(a_{i,1})_d, 0, 0, \ldots, 0, 0]^T. \tag{C29}$$

2. If $\gcd(a_{i,1})_d$ is not a zero divisor, meaning there is no $r^*$ such that $0 < r^* \leq d-1$ for which

$$\gcd(a_{i,1})_d \times r^* \equiv 0 \pmod{d}, \tag{C30}$$

we can multiply an invertible number in this row to make it equal to $+1$.

3. The first column and the first row have been processed. We then repeat the above procedures on the submatrix, which excludes the first column and the first row.

In the original matrix $A$, linear **relations** exist among the row vectors; specifically, certain row vectors can be combined linearly to yield the zero row vector (mod $d$). These relations are crucial as they provide insights into the connections between the row vectors. For instance, suppose one row $r_1$ represents the syndrome pattern of an anyon $v$, another row $r_2$ represents the syndrome pattern of a different anyon $v'$, and a third row $r_3$ represents the syndrome of a local Pauli operator $\mathcal{P}$. If these rows satisfy the relation $r_1 - r_2 + r_3 = 0$, it implies that the anyons $v$ and $v'$ are related through the local operator $\mathcal{P}$, or more specifically,

$$v' = v + \epsilon(\mathcal{P}),$$

indicating that $v$ and $v'$ are of the same anyon type. Therefore, identifying all such relations among the row vectors is essential.

We present a concrete example of implementing the modified Gaussian elimination algorithm on a selected matrix $A$ over $\mathbb{Z}_8$. The matrix $A$ is defined as follows:

$$A = \begin{bmatrix} 4 & 2 & 0 \\ 6 & 0 & 3 \\ 0 & 7 & 4 \end{bmatrix} = \begin{bmatrix} - & v_1 & - \\ - & v_2 & - \\ - & v_3 & - \end{bmatrix}, \tag{C31}$$

where $v_1$, $v_2$, and $v_3$ denote the row vectors of $A$. Our objective is to derive the relationships among the row vectors $v_1$, $v_2$, and $v_3$.

---

[21] For convenience, we choose $\gcd(a_{i,1})_d$ to be in the set $\{0, 1, 2, \ldots, d-1\}$.

First, we embed the matrix over $\mathbb{Z}$ such that each entry is chosen from $0, 1, 2, \ldots, 7$. We then insert a row $[8, 0, 0]$ at the bottom:

$$[A'|R] = \begin{bmatrix} 4 & 2 & 0 & 1 & 0 & 0 & 0 \\ 6 & 0 & 3 & 0 & 1 & 0 & 0 \\ 0 & 7 & 4 & 0 & 0 & 1 & 0 \\ 8 & 0 & 0 & 0 & 0 & 0 & 1 \end{bmatrix}, \tag{C32}$$

where the matrix $R$ is used to track row operations during the following process, recording how each current row is derived from the rows in the original matrix $A$. The greatest common divisor (gcd) of the first column is 2, which can be computed from $(-1) \times 6 + 1 \times 8$:

$$\begin{bmatrix} 4 & 2 & 0 & 1 & 0 & 0 & 0 \\ 6 & 0 & 3 & 0 & 1 & 0 & 0 \\ 0 & 7 & 4 & 0 & 0 & 1 & 0 \\ 2 & 0 & -3 & 0 & -1 & 0 & 1 \end{bmatrix}. \tag{C33}$$

Subsequently, we move the last row to the top and use it to eliminate entries in the other rows:

$$\begin{bmatrix} 2 & 0 & -3 & 0 & -1 & 0 & 1 \\ 0 & 2 & 6 & 1 & 2 & 0 & -2 \\ 0 & 0 & 12 & 0 & 4 & 0 & -3 \\ 0 & 7 & 4 & 0 & 0 & 1 & 0 \end{bmatrix} \tag{C34}$$

The first row and column have been completed. From this point onward, the first row will not be involved in subsequent calculations. We will continue the process by initially inserting $[0, 8, 0]$:

$$\begin{bmatrix} 2 & 0 & -3 & 0 & -1 & 0 & 1 & 0 \\ 0 & 2 & 6 & 1 & 2 & 0 & -2 & 0 \\ 0 & 0 & 12 & 0 & 4 & 0 & -3 & 0 \\ 0 & 7 & 4 & 0 & 0 & 1 & 0 & 0 \\ 0 & 8 & 0 & 0 & 0 & 0 & 0 & 1 \end{bmatrix}. \tag{C35}$$

The gcd of the second column (excluding the entry in the first row) is 1, obtained from $8 - 7$:

$$\begin{bmatrix} 2 & 0 & -3 & 0 & -1 & 0 & 1 & 0 \\ 0 & 2 & 6 & 1 & 2 & 0 & -2 & 0 \\ 0 & 0 & 12 & 0 & 4 & 0 & -3 & 0 \\ 0 & 7 & 4 & 0 & 0 & 1 & 0 & 0 \\ 0 & 1 & -4 & 0 & 0 & -1 & 0 & 1 \end{bmatrix}. \tag{C36}$$

Next, we place the last row in the second position and use it to cancel entries in the rows below:

$$\begin{bmatrix} 2 & 0 & -3 & 1 & 0 & 0 & -1 & 0 \\ 0 & 1 & -4 & 0 & 1 & 0 & 1 & -1 \\ 0 & 0 & 14 & 1 & 2 & 2 & -2 & -2 \\ 0 & 0 & 12 & 0 & 4 & 0 & -3 & 0 \\ 0 & 0 & 32 & 0 & 0 & 8 & 0 & -7 \end{bmatrix}. \tag{C37}$$

Finally, we insert $[0, 0, 8]$:

$$\begin{bmatrix} 2 & 0 & -3 & 1 & 0 & 0 & -1 & 0 & 0 \\ 0 & 1 & -4 & 0 & 1 & 0 & 1 & -1 & 0 \\ 0 & 0 & 14 & 1 & 2 & 2 & -2 & -2 & 0 \\ 0 & 0 & 12 & 0 & 4 & 0 & -3 & 0 & 0 \\ 0 & 0 & 32 & 0 & 0 & 8 & 0 & -7 & 0 \\ 0 & 0 & 8 & 0 & 0 & 0 & 0 & 0 & 1 \end{bmatrix}, \tag{C38}$$

and find the gcd of the third column (excluding the entries in the first and second rows) is 2, which can be obtained from $14 - 12$:

$$\begin{bmatrix} 2 & 0 & -3 & 1 & 0 & 0 & -1 & 0 & 0 \\ 0 & 1 & -4 & 0 & 1 & 0 & 1 & -1 & 0 \\ 0 & 0 & 2 & 1 & -2 & 2 & 1 & -2 & 0 \\ 0 & 0 & 12 & 0 & 4 & 0 & -3 & 0 & 0 \\ 0 & 0 & 32 & 0 & 0 & 8 & 0 & -7 & 0 \\ 0 & 0 & 8 & 0 & 0 & 0 & 0 & 0 & 1 \end{bmatrix}. \tag{C39}$$

Finally, we use this 2 to cancel all entries below:

$$[A'|R] = \begin{bmatrix} 2 & 0 & -3 & 1 & 0 & 0 & -1 & 0 & 0 \\ 0 & 1 & -4 & 0 & 1 & 0 & 1 & -1 & 0 \\ 0 & 0 & 2 & 1 & -2 & 2 & 1 & -2 & 0 \\ 0 & 0 & 0 & -6 & 16 & -12 & -9 & 12 & 0 \\ 0 & 0 & 0 & -16 & 32 & -24 & -16 & 25 & 0 \\ 0 & 0 & 0 & -4 & 8 & -8 & -4 & 8 & 1 \end{bmatrix}.$$

We have achieved the row echelon form for the integer matrix $A$. We then select the bottom-left $3 \times 3$ block of matrix $R$ to serve as the relation matrix resulting from the modified Gaussian elimination:

$$\text{relation} := \begin{bmatrix} -6 & 16 & -12 \\ -16 & 32 & -24 \\ -4 & 8 & -8 \end{bmatrix} = \begin{bmatrix} 2 & 0 & 4 \\ 0 & 0 & 0 \\ 4 & 0 & 0 \end{bmatrix} \pmod 8.$$

The last three columns in $R$ will be reduced modulo $\mathbb{Z}_8$, and thus they do not play a role in the obtained relations. The relation matrix traces the relationships among $v_1$, $v_2$, and $v_3$ as derived from Eq. (C31):

$$2v_1 + 4v_3 = [0, 0, 0], \quad 4v_1 = [0, 0, 0] \pmod 8. \tag{C40}$$

# Appendix D  Algorithm pseudocode

This appendix presents the pseudocode for the algorithm described in Sections 4.2 and 4.3. By adjusting the stabilizer polynomials $\mathcal{S}$ and the input range $A$ of Pauli operators, the algorithm can also be used to derive the condensed bulk string at the boundary or the bulk string passing through defects, as outlined in Appendix F.1. Additionally, it can be applied to determine the endpoints of defect lines, as discussed in Appendix F.2.

---

**Algorithm 1** Solving for boundary gauge operators

---

**Input:** Stabilizer polynomials $\mathcal{S}$, truncation range $k$, range of Pauli $A$, range of stabilizer $m_x$ and $m_y$, $\mathbb{Z}_d$ $d$
**Output:** The boundary gauge operators $\mathcal{G}$
1: Construct matrix $M_1$ shown in Eq. (38) based on the range of Pauli $A$, and stabilizer $m_x$ and $m_y$,

$$M_1 \leftarrow \begin{pmatrix} \langle \mathcal{P}_1 \cdot \mathcal{BS}_1 \rangle_0 & \langle \mathcal{P}_1 \cdot \mathcal{BS}_2 \rangle_0 & \langle \mathcal{P}_1 \cdot \mathcal{BS}_3 \rangle_0 & \cdots \\ \langle \mathcal{P}_2 \cdot \mathcal{BS}_1 \rangle_0 & \langle \mathcal{P}_2 \cdot \mathcal{BS}_2 \rangle_0 & \langle \mathcal{P}_2 \cdot \mathcal{BS}_3 \rangle_0 & \cdots \\ \langle \mathcal{P}_3 \cdot \mathcal{BS}_1 \rangle_0 & \langle \mathcal{P}_3 \cdot \mathcal{BS}_2 \rangle_0 & \langle \mathcal{P}_3 \cdot \mathcal{BS}_3 \rangle_0 & \cdots \\ \vdots & \vdots & \vdots & \ddots \end{pmatrix}. \tag{D41}$$

2: Perform modified Gaussian elimination (for nonprime dimensional qudits) or Gaussian elimination (for prime dimensional qudits) on $M_1$ to get a relation matrix $R_1$.
3: Obtain local operator $\mathcal{O}$ from rows of $R_1$ corresponding to zero rows in Modified Gaussian elimination
4: Construct $\widetilde{M}_2$ in Eq. (41) by applying translation duplication map $\text{TD}_{m_x, m_y}$ with range $m_x < k$ and $m_y < k$ to stabilizer matrix $\mathcal{S}$,

$$\widetilde{M}_2 \leftarrow \begin{bmatrix} \widetilde{\text{TD}_{m_x, m_y}(\mathcal{S}_1)} \\ \widetilde{\text{TD}_{m_x, m_y}(\mathcal{S}_2)} \\ \vdots \\ \widetilde{\text{TD}_{m_x, m_y}(\mathcal{S}_t)} \end{bmatrix}. \tag{D42}$$

$\widetilde{M}_2$ is a $[t(2m_x + 1)(2m_y + 1)] \times [(2k+1)^2]$ matrix with $t$ stabilizers.
5: Perform MGE on $\widetilde{M}_2$ to get $\text{MGE}(\widetilde{M}_2)$.
6: Define the boundary gauge operator $\mathcal{G}$.
7: Local operator matrix $\widetilde{\mathcal{O}} \leftarrow$ apply the truncation map to local operator set $\mathcal{O}$
8: **for** $\widetilde{\mathcal{O}}_i$ in local operator matrix $\widetilde{\mathcal{O}}$ **do**
9:     **if** $\widetilde{\mathcal{O}}_i$ is in the row span of $\text{MGE}(\widetilde{M}_2)$ **then**
10:         $\mathcal{G}_i \leftarrow \widetilde{\mathcal{O}}_i$

11:     Construct $\widetilde{M}_3$ in Eq. (42),

$$\widetilde{M}_3 \leftarrow \widetilde{\mathrm{TD}_{m_x=0,m_y}(\mathcal{G})}. \tag{D43}$$

12:     $\mathrm{MGE}(\widetilde{M}_2) \leftarrow \mathrm{concate}(\mathrm{MGE}(\widetilde{M}_2), \widetilde{M}_3)$
13:   **end if**
14: **end for**
15: **return** The boundary gauge operators $\mathcal{G}$

---

**Algorithm 2** Computing boundary anyons and boundary string operators

---

**Input:** The boundary gauge operators $\mathcal{G}$, range of boundary gauge operators $m_y$, search range $N_y$ for $y$-direction.
**Output:** Basis boundary anyons $V$ and boundary string operators $P$
1: **for** $n = 1, 2, ..., N_y$ **do**
2:     Get the error syndromes $\zeta(\mathcal{G})$ between boundary gauge operators. Since $\mathcal{G}$ only has translational symmetry in the y-direction, we should only extract the polynomials containing $x^0$ of each dot product.
3:     Define matrix $\widetilde{M}_4$ in Eq. (47)as

$$\widetilde{M}_4 \leftarrow \begin{bmatrix} \widetilde{\mathrm{TD}_{m_x=0,m_y}(\zeta(\mathcal{G}_1))} \\ \widetilde{\mathrm{TD}_{m_x=0,m_y}(\zeta(\mathcal{G}_2))} \\ \vdots \\ \widetilde{\mathrm{TD}_{m_x=0,m_y}(\zeta(\mathcal{G}_r))} \\ \widetilde{\mathrm{TD}_{m_x=0,m_y}([(1-y^n),0,...,0])} \\ \widetilde{\mathrm{TD}_{m_x=0,m_y}([0,(1-y^n),...,0])} \\ \vdots \\ \widetilde{\mathrm{TD}_{m_x=0,m_y}([0,0,...,(1-y^n)])} \end{bmatrix}. \tag{D44}$$

$\widetilde{M}_4$ is a $[2r(2m+1)^2] \times [r(2k+1)^2]$ matrix with $r$ boundary gauge operators.
4:     Calculate $\mathrm{MGE}(\widetilde{M}_4)$, obtain a boundary anyon matrix

$$\widetilde{V} \leftarrow \begin{bmatrix} \widetilde{v_1} \\ \vdots \\ \widetilde{v_\alpha} \end{bmatrix} \tag{D45}$$

which is a $\alpha \times (r(2k+1)^2)$ matrix and relation matrix $R_1$.[22] Their string operators along the $x$-direction form the string operator matrix

$$\widetilde{P} \leftarrow \begin{bmatrix} \widetilde{P^{v_1}} \\ \vdots \\ \widetilde{P^{v_\alpha}} \end{bmatrix} \tag{D46}$$

which is a $\alpha \times (2r(2k+1)^2)$ matrix obtained from the relation matrix $R_1$.
5:     Construct $\widetilde{M}_5$ in Eq. (48),

$$\widetilde{M}_5 := \begin{bmatrix} \widetilde{\mathrm{TD}_{m_x=0,m_y}(\zeta(\mathcal{G}_1))} \\ \widetilde{\mathrm{TD}_{m_x=0,m_y}(\zeta(\mathcal{G}_2))} \\ \vdots \\ \widetilde{\mathrm{TD}_{m_x=0,m_y}(\zeta(\mathcal{G}_r))} \end{bmatrix}, \tag{D47}$$

6:     Perform MGE on $\widetilde{M}_5$ to get $\mathrm{MGE}(\widetilde{M}_5)$.

---

[22] Assume we get $\alpha$ different boundary anyon solutions here.

 Define the basis boundary anyon matrix $\widetilde{V}'$ and corresponding boundary string operator matrix $\widetilde{P}'$.

8:     **for** $\widetilde{V}_i, \widetilde{P}_i$ in anyon matrix $\widetilde{V}, \widetilde{P}$ **do**

9:         **if** $\widetilde{V}_i$ is not in the row span of $\mathrm{MGE}(\widetilde{M}_5)$ **then**

10:            $\widetilde{V}' \leftarrow \mathrm{concate}(\widetilde{V}', \widetilde{V}_i)$

11:            $\widetilde{P}' \leftarrow \mathrm{concate}(\widetilde{P}', \widetilde{P}_i)$

12:            $\mathrm{MGE}(\widetilde{M}_5) \leftarrow \mathrm{concate}(\mathrm{MGE}(\widetilde{M}_5), \widetilde{V}_i)$

13:         **end if**

14:     **end for**

15:     **if** The $d$ of $\mathbb{Z}_d$ is nonprime **then**

16:         Construct boundary anyon relation matrix $M_v$ by $\widetilde{V}'$ and $\widetilde{P}'$.

17:         Calculate the Smith normal form of $M_v$ as $PAQ = M_v$

18:         $index \leftarrow \arg_i A(i,i) \neq \pm 1$

19:         Refresh the basis boundary anyon matrix by $\widetilde{V}' = \widetilde{V}'Q(:, index)$.

20:         Refresh the corresponding boundary string operator matrix by $\widetilde{P}' = \widetilde{P}'Q(:, index)$.

21:     **end if**

22: **end for**

23: Choose the smallest $n$ as $n^*$ and ensure that the maximum number of boundary anyons can be obtained.

$$
\begin{aligned}
\text{Basis boundary anyons } V &\leftarrow \widetilde{V}'(n_y^*) \\
\text{Boundary string operators } P &\leftarrow \widetilde{P}'(n_y^*)
\end{aligned}
\tag{D48}
$$

24: **return** Basis boundary anyons $V$ and boundary string operators $P$

---

# Appendix E    Counting the Lagrangian subgroups of the $\mathbb{Z}_2^n$ toric code

In this section, we count the number of Lagrangian subgroups in the $n$-copy $\mathbb{Z}_2$ toric codes. For a $G$-gauge theory, it is well known that gapped boundaries (Lagrangian subgroups) are classified by a subgroup $N \subset G$ and a 2-cocycle in $H^2(N, U(1))$ [36, 143, 144]. Ref. [49] provides explicit representations of the Lagrangian subgroups for the $\mathbb{Z}_2^n$ toric code when $n = 1, 2, 4$. Here, we present the formula for counting the number of Lagrangian subgroups for general $n$.

For small values of $n$, we first use a computer to enumerate all Lagrangian subgroups:

1. For $n = 1$, $|\mathcal{L}(\mathbb{Z}_2 \text{ toric code})| = 2$.
2. For $n = 2$, $|\mathcal{L}(\mathbb{Z}_2^2 \text{ toric code})| = 6$.
3. For $n = 3$, $|\mathcal{L}(\mathbb{Z}_2^3 \text{ toric code})| = 30$.
4. For $n = 4$, $|\mathcal{L}(\mathbb{Z}_2^4 \text{ toric code})| = 270$.
5. For $n = 5$, $|\mathcal{L}(\mathbb{Z}_2^5 \text{ toric code})| = 4590$.

In the following, we will prove

**Theorem 20.** *The number of Lagrangian subgroups of the $\mathbb{Z}_2^n$ torc code is given by:*

$$
|\mathcal{L}(\mathbb{Z}_2^n \text{ toric code})| = \prod_{i=0}^{n-1} (2^i + 1).
\tag{E49}
$$

This corresponds to OEIS sequence A028361, enumerating totally isotropic spaces of index $n$ in orthogonal geometry of dimension $2n$. In the $\mathbb{Z}_2$ case, the symplectic bilinear form $\Lambda$, defined in Eq. (27), becomes symmetric and can be interpreted as an orthogonal space. The Lagrangian subgroups correspond to the isotropic spaces. In addition, this sequence represents the number of nodes in the bosonic orbifold groupoid for $\mathbb{Z}_2^n$ symmetry [145].

Using the counting formula from Eq. (E49), the number of Lagrangian subgroups for the two bivariate bicycle codes studied in Sec. 5 are 1,270,075,950 for $n = 8$ and 167,448,083,323,950 for $n = 10$.

*Proof of Theorem 20.* The $\mathbb{Z}_2^n$ toric code corresponds to a $G$ gauge theory with $G = \mathbb{Z}_2^n$, where any subgroup $N$ must be isomorphic to $\mathbb{Z}_2^k$ for $0 \leq k \leq n$. In the second cohomology group $H^2(N, U(1))$, only type-II

cocycles exist, and their generators are composed of pairs of elements from $\mathbb{Z}_2$ within $\mathbb{Z}_2^k$, leading to

$$|H^2(\mathbb{Z}_2^k, U(1))| = 2^{\binom{k}{2}}. \tag{E50}$$

Let $a_k^n$ denote the number of $k$-dimensional subspaces in an $n$-dimensional vector space over $\mathbb{Z}_2$, which corresponds to the number of subgroups $N \subset G$ such that $N \simeq \mathbb{Z}_2^k$. The total number of Lagrangian subgroups is then given by:

$$|\mathcal{L}(\mathbb{Z}_2^n \text{ toric code})| = \sum_{k=0}^{n} a_k^n \times 2^{\binom{k}{2}}. \tag{E51}$$

Next, the basis of a $k$-dimensional subspace can always be reduced to the row echelon form:

$$\left.\begin{bmatrix} \overbrace{0 \cdots 1}^{b_0} \overbrace{* \cdots 0}^{b_1} \overbrace{* \cdots 0}^{b_2} \overbrace{* \cdots 0}^{b_3} \overbrace{* \cdots}^{b_k} \\ 0 \cdots 0\, 0 \cdots 1\, * \cdots 0\, * \cdots 0\, * \cdots \\ 0 \cdots 0\, 0 \cdots 0\, 0 \cdots 1\, * \cdots 0\, * \cdots \\ 0 \cdots 0\, 0 \cdots 0\, 0 \cdots 0\, 0 \cdots 0\, * \cdots \\ \vdots\, \ddots\, \vdots\, \vdots\, \ddots\, \vdots\, \vdots\, \ddots\, \vdots\, \vdots\, \ddots\, \vdots\, \vdots\, \ddots \\ 0 \cdots 0\, 0 \cdots 0\, 0 \cdots 0\, 0 \cdots 1\, * \cdots \end{bmatrix}\right\} k \tag{E52}$$

where $b_0 + b_1 + \cdots + b_k = n - k$, ensuring that the matrix has size $k \times n$. Each $*$ in the matrix (E52) can be either 0 or 1; thus, the number of $k$-dimensional subspaces in an $n$-dimensional vector space over $\mathbb{Z}_2$ is:

$$a_k^n = \sum_{\substack{b_0, b_1, \ldots, b_k | \\ b_0 + b_1 + \ldots + b_k = n-k}} 2^{b_1 + 2b_2 + 3b_3 + \cdots + kb_k}. \tag{E53}$$

To simplify the expression in Eq. (E51), we derive a useful recursive relation for $a_k^{n+1}$:

$$a_k^{n+1} = 2^k a_k^n + a_{k-1}^n. \tag{E54}$$

The first term arises from the case where $b_k \geq 1$, contributing $2^k a_k^n$ by redefining $b_k' = b_k - 1$ and factoring out the overall $2^k$ from the summation. The second term results from the case where $b_k = 0$, which reduces to computing $a_{k-1}^n$.

Before proceeding, we introduce a useful lemma for our subsequent computations:

**Lemma 21.** *The number of $k$-dimensional subspaces of an $n$-dimensional vector space over $\mathbb{Z}_2$, $a_k^n$ defined in Eq. (E53), satisfy the following relation:*

$$\sum_{k=0}^{n-l} \left( \prod_{i=0}^{l-1} 2^k (1 + 2^i) \right) \times a_k^{n-l} \times 2^{\binom{k}{2}} = \sum_{k=0}^{n-l-1} \left( \prod_{i=0}^{l} 2^k (1 + 2^i) \right) \times a_k^{n-l-1} \times 2^{\binom{k}{2}}, \tag{E55}$$

*for any $1 \leq l \leq n - 1$.*

This lemma will be proved later in this appendix.

Now, we simplify Eq. (E51) by substituting $a_k^n$ with $a_k^{n-1}$ and $a_{k-1}^{n-1}$:

$$|\mathcal{L}(\mathbb{Z}_2^n \text{ toric code})| = \sum_{k=0}^{n} (2^k a_k^{n-1} + a_{k-1}^{n-1}) \times 2^{\binom{k}{2}}$$

$$= \sum_{k=0}^{n-1} (2^k a_k^{n-1}) \times 2^{\binom{k}{2}} + \sum_{k=1}^{n} a_{k-1}^{n-1} \times 2^{\binom{k}{2}}, \tag{E56}$$

where we must carefully adjust the range of the summations. Since $\binom{k}{2} = \binom{k-1}{2} + \binom{k-1}{1}$, we introduce the substitution $k' = k - 1$ for the second term and rewrite the expression as:

$$|\mathcal{L}(\mathbb{Z}_2^n \text{ toric code})| = \sum_{k=0}^{n-1} (2^k a_k^{n-1}) \times 2^{\binom{k}{2}} + \sum_{k=1}^{n} a_{k-1}^{n-1} \times 2^{\binom{k-1}{2} + \binom{k-1}{1}}$$

$$= \sum_{k=0}^{n-1} (2^k a_k^{n-1}) \times 2^{\binom{k}{2}} + \sum_{k'=0}^{n-1} 2^{k'} a_{k'}^{n-1} \times 2^{\binom{k'}{2}}$$

$$= \sum_{k=0}^{n-1} (\prod_{i=0}^{0} 2^k (1 + 2^i)) \times a_k^{n-1} \times 2^{\binom{k}{2}}.$$

By repeatedly applying the recurrence relation in Eq. (E55), we reduce the summation involving $a_k^n$ down to $a_k^0$:

$$|\mathcal{L}(\mathbb{Z}_2^n \text{ toric code})| = \sum_{k=0}^{n-2} \left( \prod_{i=0}^{1} 2^k (1 + 2^i) \right) \times a_k^{n-2} \times 2^{\binom{k}{2}}$$

$$= \sum_{k=0}^{n-3} \left( \prod_{i=0}^{2} 2^k (1 + 2^i) \right) \times a_k^{n-3} \times 2^{\binom{k}{2}} \tag{E57}$$

$$\vdots$$

$$= \sum_{k=0}^{0} \left( \prod_{i=0}^{n-1} 2^k (1 + 2^i) \right) \times a_k^{0} \times 2^{\binom{k}{2}}$$

Finaly, by $a_0^0 = 1$ and $\binom{0}{2} = 1$, we derive

$$|\mathcal{L}(\mathbb{Z}_2^n \text{ toric code})| = \prod_{i=0}^{n-1} (2^i + 1). \tag{E58}$$

$\square$

In the final part of this appendix, we prove Lemma 21.

*Proof of Lemma 21.* We begin the proof by simplifying the left-hand side. Using Eq. (E54), we substitute $a_k^{n-l}$ with $a_k^{n-l-1}$ and $a_{k-1}^{n-l-1}$:

$$\sum_{k=0}^{n-l} \left( \prod_{i=0}^{l-1} 2^k (1 + 2^i) \right) \times a_k^{n-l} \times 2^{\binom{k}{2}}$$

$$= \sum_{k=0}^{n-l} \left( \prod_{i=0}^{l-1} 2^k (1 + 2^i) \right) \times (2^k a_k^{n-l-1} + a_{k-1}^{n-l-1}) \times 2^{\binom{k}{2}}$$

$$= \sum_{k=0}^{n-l-1} \left( \prod_{i=0}^{l-1} 2^k (1 + 2^i) \right) \times 2^k a_k^{n-l-1} \times 2^{\binom{k}{2}} + \sum_{k=1}^{n-l} \left( \prod_{i=0}^{l-1} 2^k (1 + 2^i) \right) \times a_{k-1}^{n-l-1} \times 2^{\binom{k}{2}},$$

where we have adjusted the range of the summations. By the identity $\binom{k}{2} = \binom{k-1}{2} + \binom{k-1}{1}$ and the substitution $k' = k - 1$ for the second term, we rewrite the above expression as:

$$\sum_{k=0}^{n-l-1}\left(\prod_{i=0}^{l-1}2^k(1+2^i)\right)\times 2^k a_k^{n-l-1}\times 2^{\binom{k}{2}} + \sum_{k=1}^{n-l}\left(\prod_{i=0}^{l-1}2^k(1+2^i)\right)\times a_{k-1}^{n-l-1}\times 2^{\binom{k-1}{2}+\binom{k-1}{1}}$$

$$= \sum_{k=0}^{n-l-1}\left(\prod_{i=0}^{l-1}2^k(1+2^i)\right)\times 2^k a_k^{n-l-1}\times 2^{\binom{k}{2}} + \sum_{k'=0}^{n-l-1}\left(\prod_{i=0}^{l-1}2^{k'+1}(1+2^i)\right)\times 2^{k'} a_{k'}^{n-l-1}\times 2^{\binom{k'}{2}}$$

$$= \sum_{k=0}^{n-l-1}\left(\prod_{i=0}^{l-1}2^k(1+2^i)\right)\times 2^k a_k^{n-l-1}\times 2^{\binom{k}{2}} + \sum_{k'=0}^{n-l-1}\left(\prod_{i=0}^{l-1}2^{k'}(1+2^i)\right)\times 2^{l+k'} a_{k'}^{n-l-1}\times 2^{\binom{k'}{2}}$$

$$= \sum_{k=0}^{n-l-1}(\prod_{i=0}^{l}2^k(1+2^i))\times a_k^{n-l-1}\times 2^{\binom{k}{2}}.$$

Therefore, we have proved Eq. (E55). $\qquad\square$

# Appendix F  Extended applications of Algorithm 1

As shown in previous sections, we have employed Algorithm 1 to determine boundary gauge operators that commute with the bulk stabilizer. This algorithm has broader applications, allowing us to identify all operators that commute with a given set of input operators within a specified range. In Sec. 4.2, we concentrated on finding boundary gauge operators within a specific boundary range. By selecting different input operators over various ranges, we can derive further results of interest.

The essence of the algorithm lies in computing operators that commute with the input operators. This method can be used to obtain condensed bulk strings that terminate on the boundary or to generate bulk strings passing through a defect, as discussed in Appendix F.1. Moreover, the algorithm can be applied to identify the operator located at the endpoint of a finite defect line, as described in Appendix F.2.

## F.1  Bulk strings terminating on boundaries or passing through defects

We can apply Algorithm 1 to obtain condensed bulk strings, as it ensures that these strings commute with all stabilizers and the additional boundary terms. Instead of selecting the range of Pauli operators along the boundary, as described in Sec. 4.2, we choose a range where the Pauli operators act along a finite-width line extending from the boundary into the bulk. Using Algorithm 1, we can find operators that commute with both the boundary Hamiltonian and the bulk stabilizers, except near the endpoint of the line in the bulk. In practice, this can be done on a finite lattice by defining the range of Pauli operators within a rectangle that spans from the right boundary to the left boundary.

The procedure for bulk strings passing through a defect is similar. We select a range of Pauli operators that crosses the defect from left to right. By applying the same algorithm from Sec. 4.2, we can find the bulk string passing through the defect, ensuring it commutes with all defect Hamiltonians and bulk stabilizers, except at its endpoints. For specific examples, refer to Sec. 5.

## F.2  Determining commuting operators at defect line endpoints

If the defect line is finite, we design the Hamiltonian terms at its endpoints. First, we introduce all the defect terms with translational symmetry in the $y$-direction along the defect line, extending up to the endpoints. Then, we select a region of Pauli operators large enough to cover one of the endpoints and apply the algorithm to identify the commutant of the defect Hamiltonian terms and the bulk stabilizers away from the defect. This process effectively completes the topological order around the defect endpoints.

Using this method, we can identify the operator at the endpoint of the defect line, as illustrated in Fig. F2. Furthermore, by applying the approach outlined in Appendix F.1, we can derive the corresponding bulk string operators that terminate at the defect endpoint, as shown in Fig. F3.

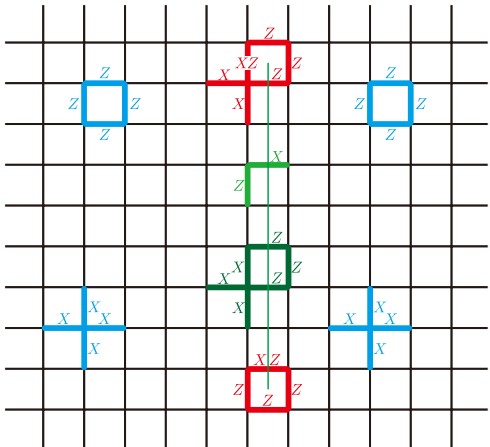

**Fig. F2** The blue components represent the bulk stabilizers of the $\mathbb{Z}_2$ toric code. The green components indicate the defect terms added along the defect line, extending up to its endpoints. The red components highlight the defect Hamiltonian terms specifically introduced near the endpoints.

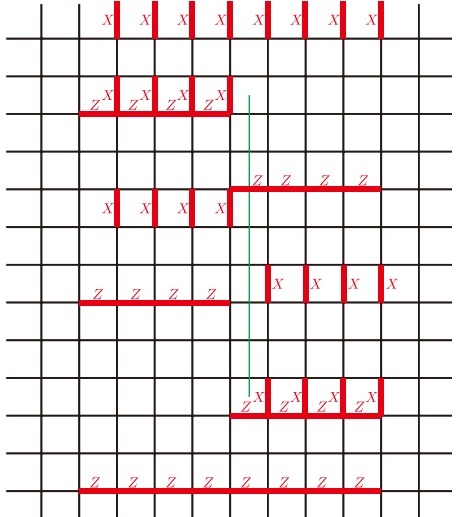

**Fig. F3** Based on the construction shown in Fig. F2, we obtain the string operators that commute with all defect Hamiltonian terms, highlighted in red.

# Appendix G  Explicit boundary and defect constructions for various quantum codes

This section presents the explicit boundary and defect constructions for various examples discussed in Sec. 5. Fig. G4 illustrates 6 defects in the $\mathbb{Z}_2$ fish toric code. Figs. G5 and G6 demonstrate 22 defects in the $\mathbb{Z}_4$ toric code. Fig. G7 shows 6 defects in the three-fermion code. Fig. G8 constructs 6 boundaries of the color code, representing both the left and right boundaries of semi-infinite planes. Fig. G9 lists 16 generators of the boundary string operators for the $(3,3)$-BB code.

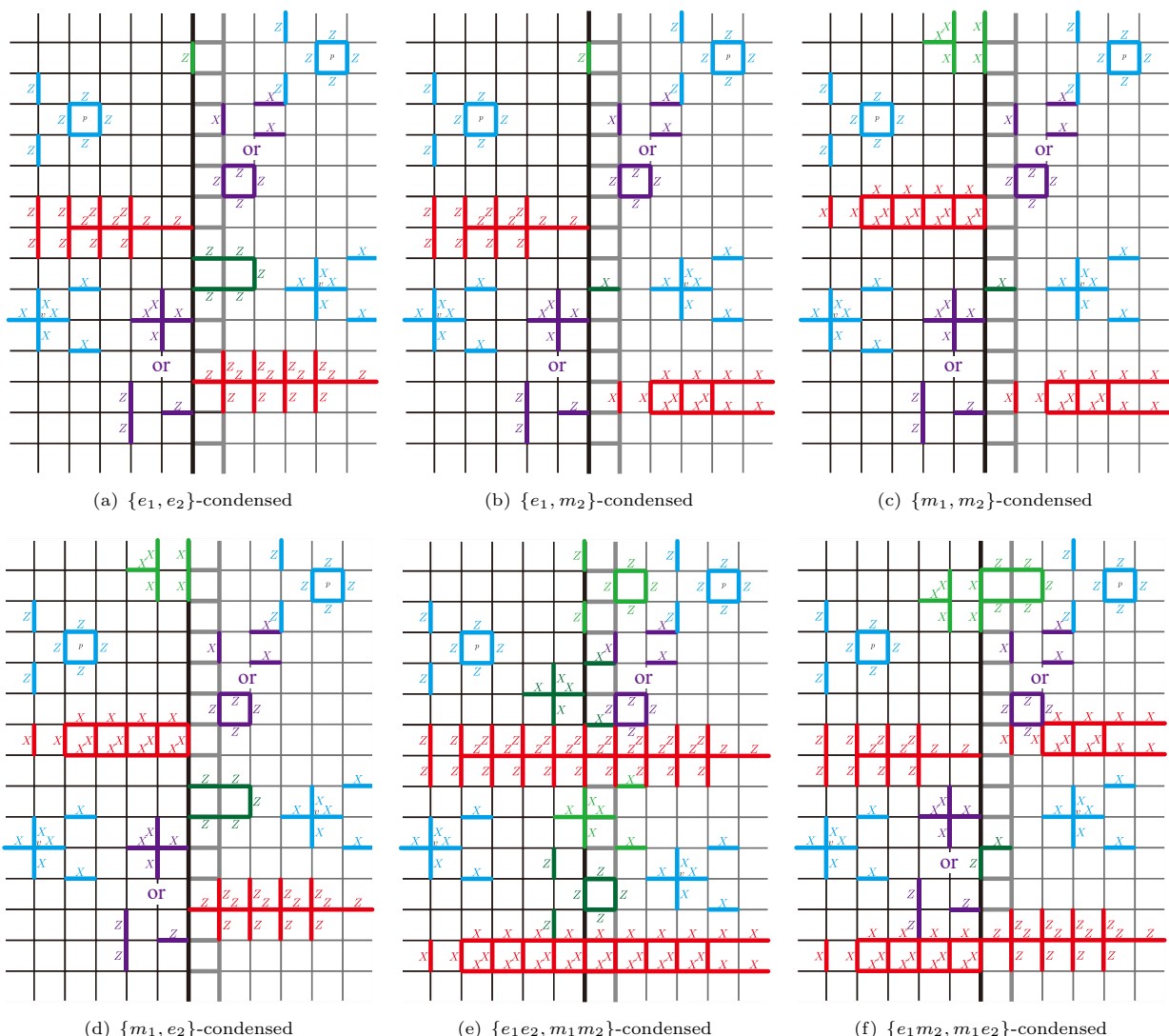

(a) $\{e_1, e_2\}$-condensed

(b) $\{e_1, m_2\}$-condensed

(c) $\{m_1, m_2\}$-condensed

(d) $\{m_1, e_2\}$-condensed

(e) $\{e_1 e_2, m_1 m_2\}$-condensed

(f) $\{e_1 m_2, m_1 e_2\}$-condensed

**Fig. G4** The defects of the $\mathbb{Z}_2$ fish toric code. Blue components indicate bulk stabilizers, green components represent the defect Hamiltonian, and red components show bulk string operators that terminate on or pass through the defect. The red strings commute with the green defect Hamiltonian. Figures (a), (b), (c), and (d) depict non-invertible defects, where the left-hand side and right-hand side are decoupled, with $e_1$ or $m_1$ and $e_2$ or $m_2$ condensed independently. Figures (e) and (f) illustrate invertible defects: (e) corresponds to the trivial defect, where the defect Hamiltonian matches the bulk Hamiltonian, and (f) represents the $e$-$m$ exchange defect, where $e$ and $m$ are permuted as they pass through the defect.

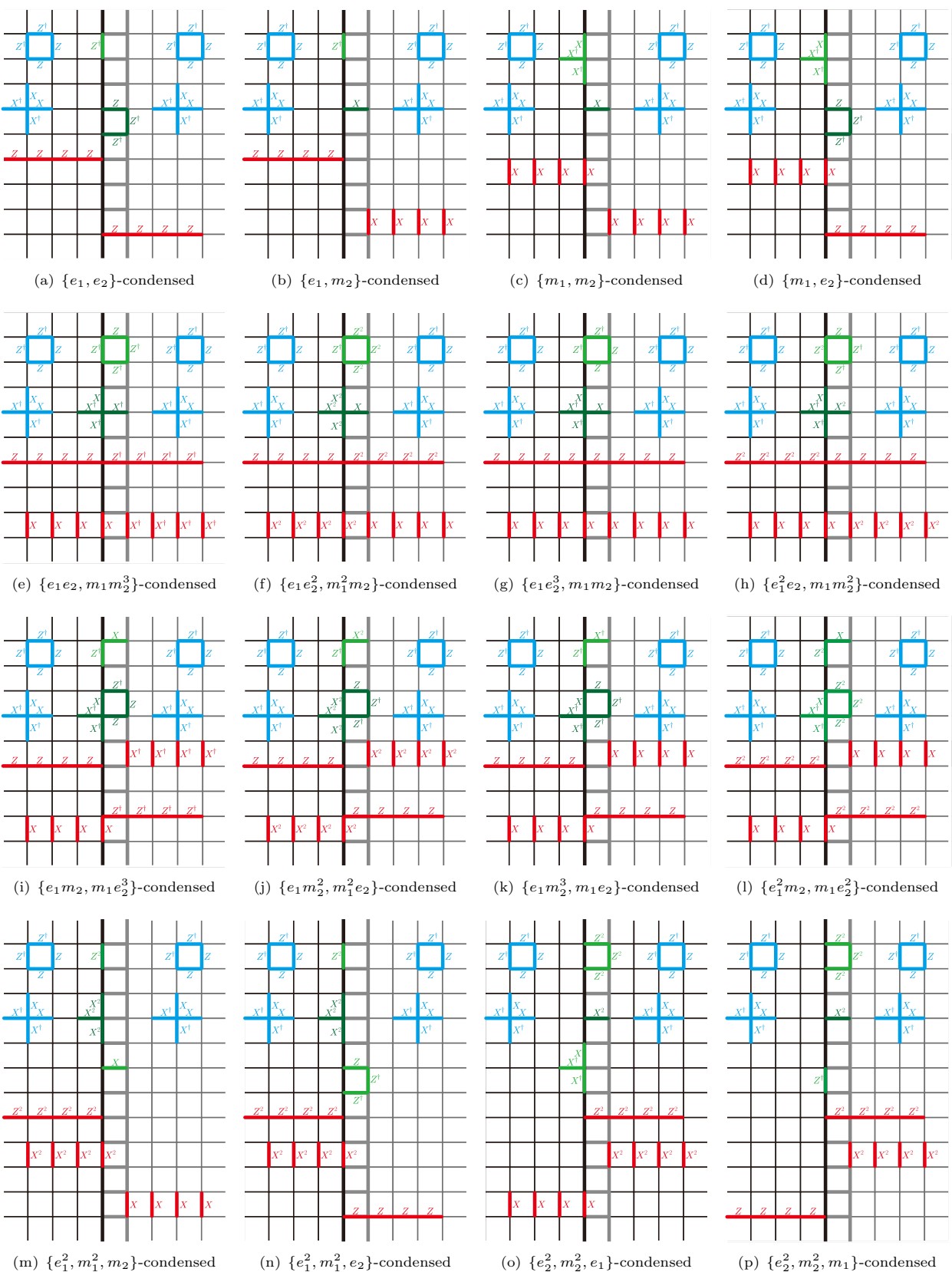

(a) $\{e_1, e_2\}$-condensed

(b) $\{e_1, m_2\}$-condensed

(c) $\{m_1, m_2\}$-condensed

(d) $\{m_1, e_2\}$-condensed

(e) $\{e_1 e_2, m_1 m_2^3\}$-condensed

(f) $\{e_1 e_2^2, m_1^2 m_2\}$-condensed

(g) $\{e_1 e_2^3, m_1 m_2\}$-condensed

(h) $\{e_1^2 e_2, m_1 m_2^2\}$-condensed

(i) $\{e_1 m_2, m_1 e_2^3\}$-condensed

(j) $\{e_1 m_2^2, m_1^2 e_2\}$-condensed

(k) $\{e_1 m_2^3, m_1 e_2\}$-condensed

(l) $\{e_1^2 m_2, m_1 e_2^2\}$-condensed

(m) $\{e_1^2, m_1^2, m_2\}$-condensed

(n) $\{e_1^2, m_1^2, e_2\}$-condensed

(o) $\{e_2^2, m_2^2, e_1\}$-condensed

(p) $\{e_2^2, m_2^2, m_1\}$-condensed

**Fig. G5** The defects of the $\mathbb{Z}_4$ toric code (part 1). Blue components indicate bulk stabilizers, green components represent the defect Hamiltonian, and red components show bulk string operators that terminate on or pass through the defect. The red strings commute with the green defect Hamiltonian.

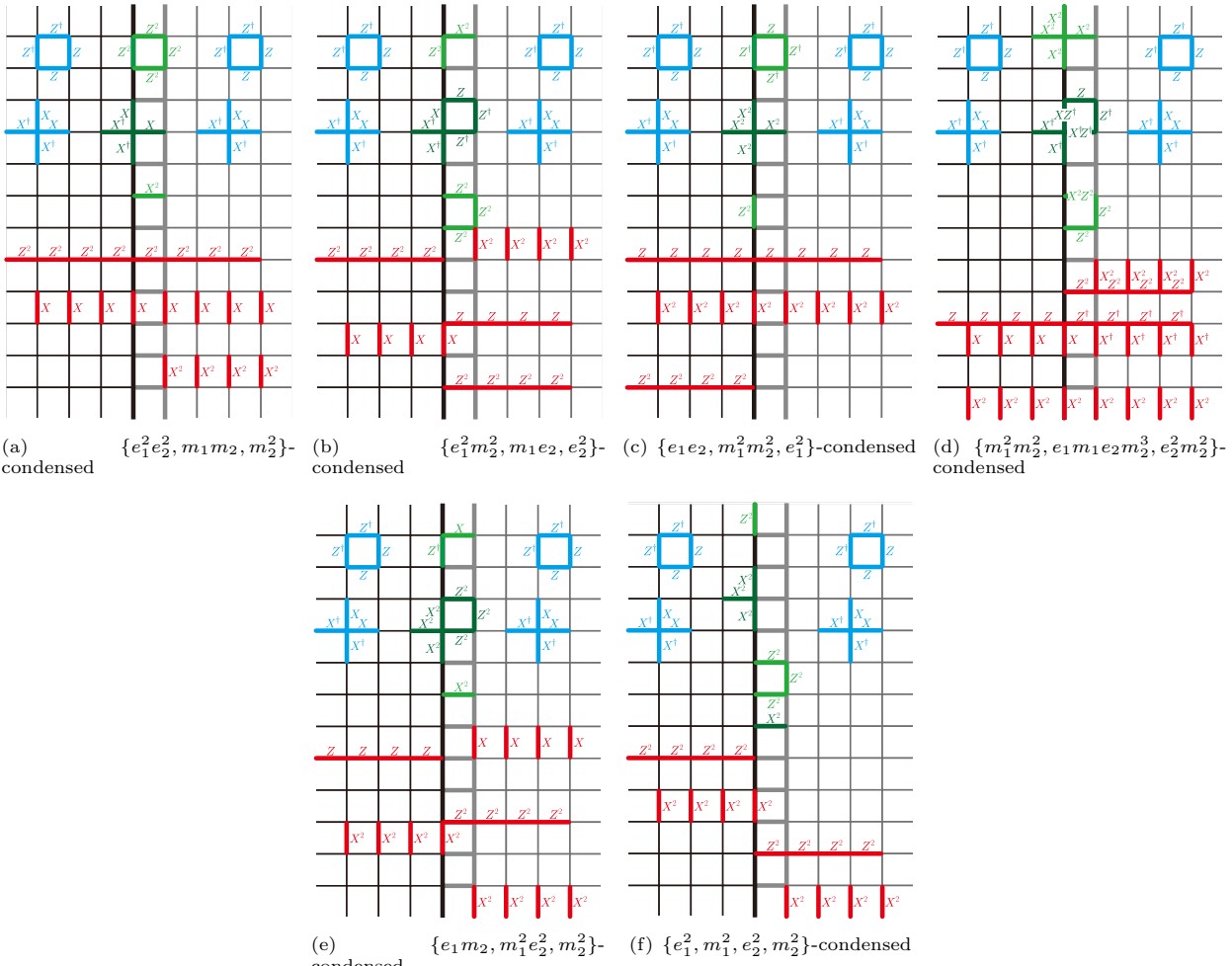

(a) $\{e_1^2 e_2^2, m_1 m_2, m_2^2\}$-condensed

(b) $\{e_1^2 m_2^2, m_1 e_2, e_2^2\}$-condensed

(c) $\{e_1 e_2, m_1^2 m_2^2, e_1^2\}$-condensed

(d) $\{m_1^2 m_2^2, e_1 m_1 e_2 m_2^3, e_2^2 m_2^2\}$-condensed

(e) $\{e_1 m_2, m_1^2 e_2^2, m_2^2\}$-condensed

(f) $\{e_1^2, m_1^2, e_2^2, m_2^2\}$-condensed

**Fig. G6** The defects of the $\mathbb{Z}_4$ toric code (part 2). Blue components indicate bulk stabilizers, green components represent the defect Hamiltonian, and red components show bulk string operators that terminate on or pass through the defect. The red strings commute with the green defect Hamiltonian.

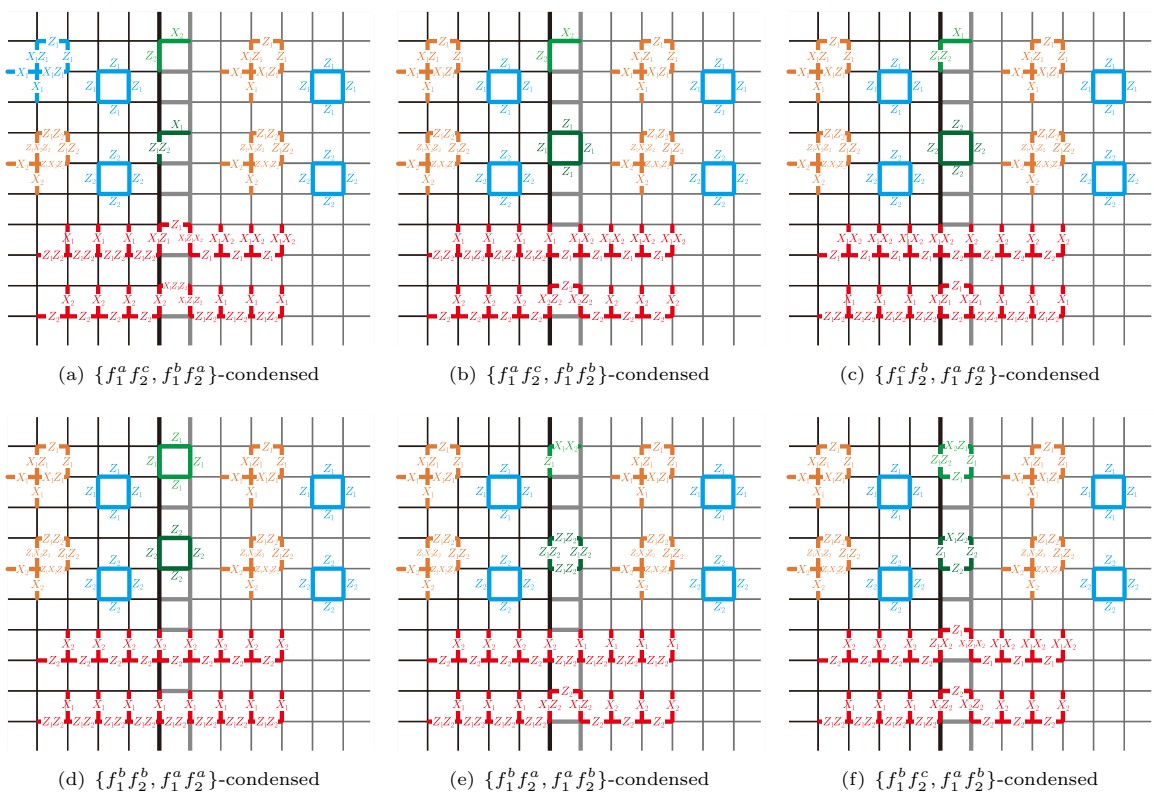

(a) $\{f_1^a f_2^c, f_1^b f_2^a\}$-condensed

(b) $\{f_1^a f_2^c, f_1^b f_2^b\}$-condensed

(c) $\{f_1^c f_2^b, f_1^a f_2^a\}$-condensed

(d) $\{f_1^b f_2^b, f_1^a f_2^a\}$-condensed

(e) $\{f_1^b f_2^a, f_1^a f_2^b\}$-condensed

(f) $\{f_1^b f_2^c, f_1^a f_2^b\}$-condensed

**Fig. G7** The defects in the anomalous three-fermion code are depicted. Blue components represent the bulk stabilizers, green components indicate the defect Hamiltonian, and red components show the bulk string operators that either terminate at or pass through the defect. The red strings commute with the green defect Hamiltonian.

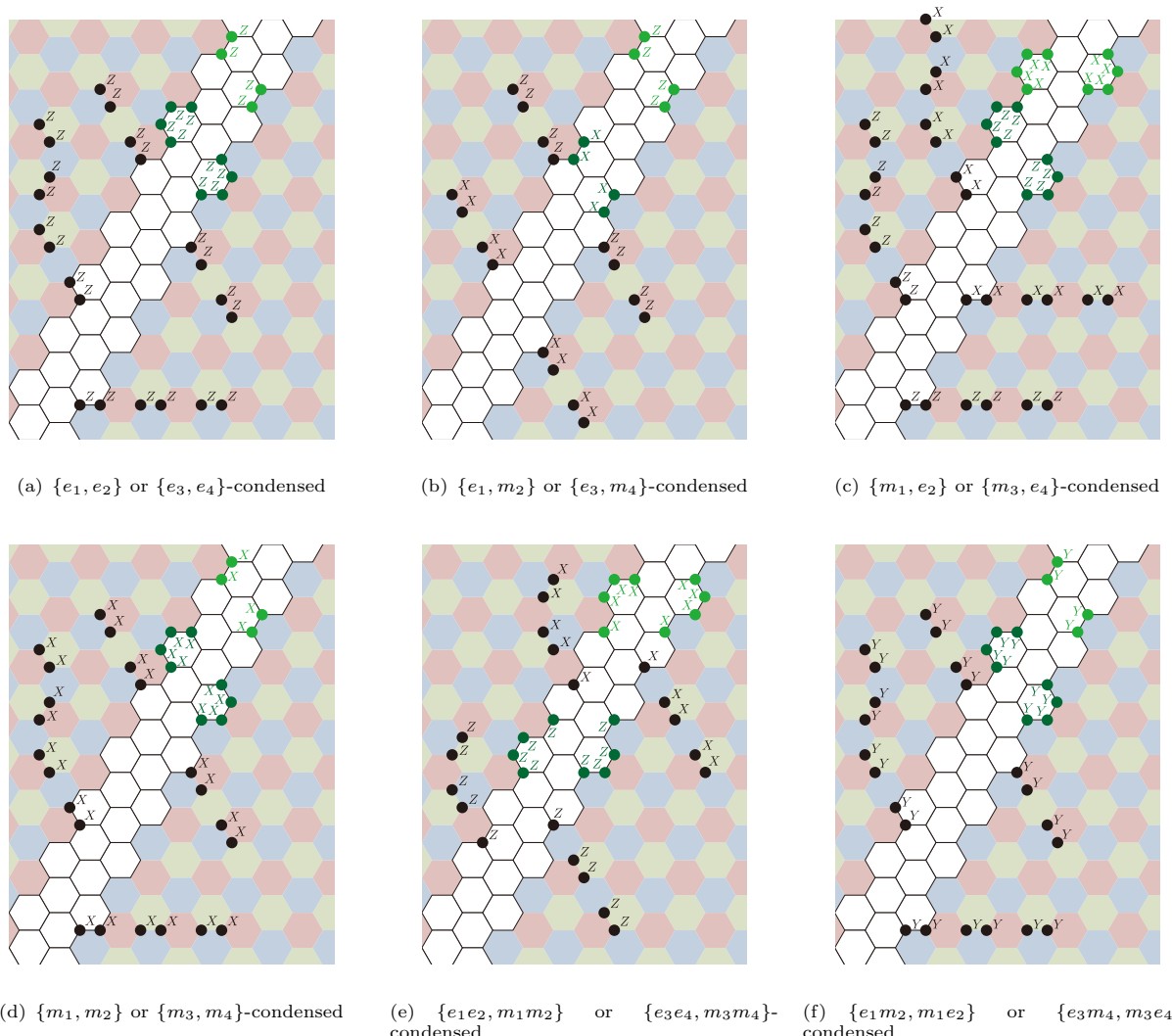

(a) $\{e_1, e_2\}$ or $\{e_3, e_4\}$-condensed

(b) $\{e_1, m_2\}$ or $\{e_3, m_4\}$-condensed

(c) $\{m_1, e_2\}$ or $\{m_3, e_4\}$-condensed

(d) $\{m_1, m_2\}$ or $\{m_3, m_4\}$-condensed

(e) $\{e_1 e_2, m_1 m_2\}$ or $\{e_3 e_4, m_3 m_4\}$-condensed

(f) $\{e_1 m_2, m_1 e_2\}$ or $\{e_3 m_4, m_3 e_4\}$-condensed

**Fig. G8** The 6 boundaries of the color code for both the left and right semi-infinite planes are shown. Blue, red, and green hexagons represent the bulk stabilizers of the color code. The green components indicate the boundary Hamiltonian and the black lines represent bulk strings that terminate at the boundary without causing energy violations.

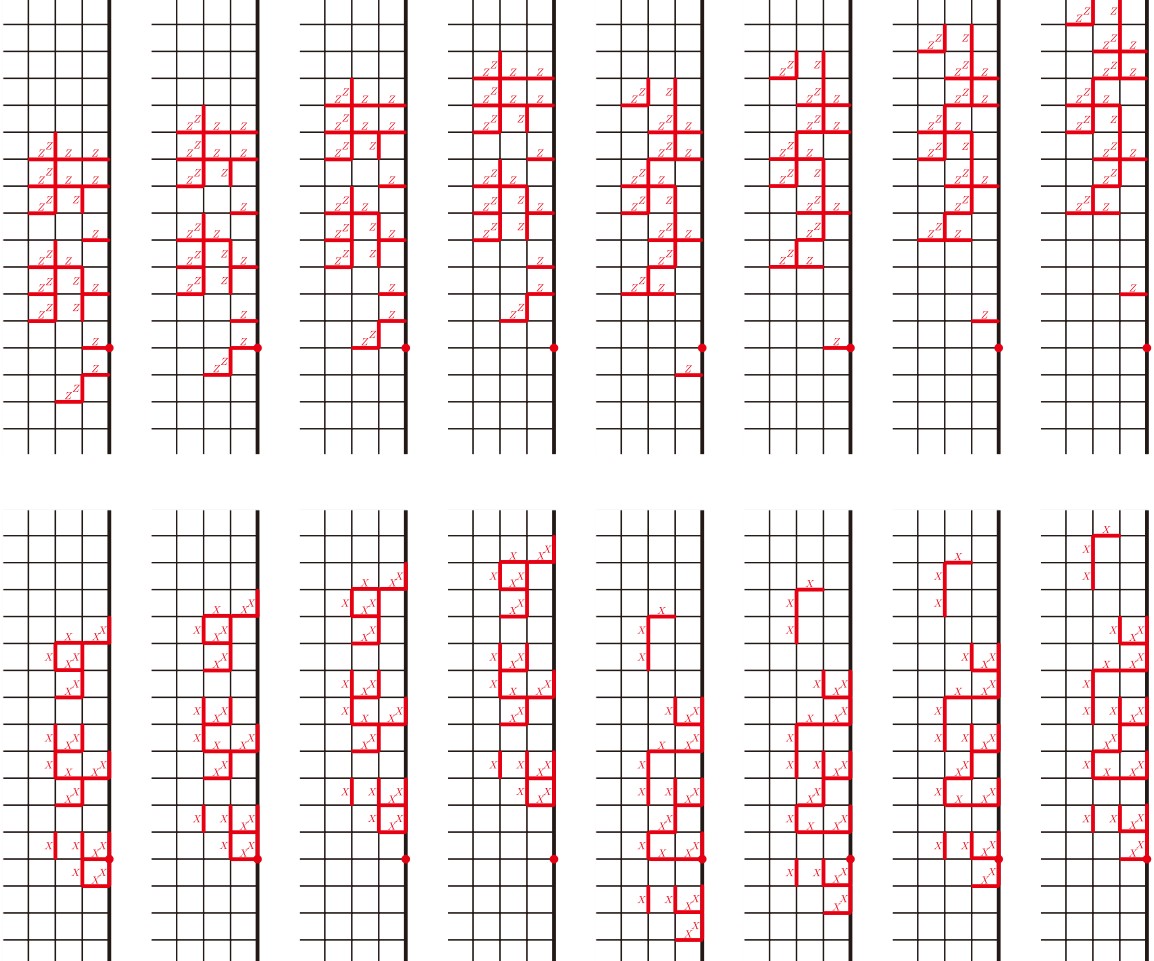

**Fig. G9** The red components depict the 16 generators of the boundary string operators for the $(3,3)$-BB code, as defined in Eq. (59). Each string operator has a length of 12 and can be multiplied by its shifted version to form a longer string operator, moving the boundary anyon by multiples of 12. While some of these string operators are shifted versions of others, the boundary anyons they generate are not equivalent under local boundary gauge operators.