# Peer review of "Operator algebra and algorithmic construction of boundaries and defects in (2+1)D topological Pauli stabilizer codes"

_SciPost Physics_

## Round 4 · Referee Report · Anonymous (Referee 1) · 2025-11-9

The referee discloses that the following generative AI tools have been used in the preparation of this report:
AI was used to improve English grammar.
Strengths
The paper develops a unified operator-algebraic and algorithmic framework for constructing all possible gapped boundaries and defects in two-dimensional topological Pauli stabilizer codes. The authors introduce a correspondence between bulk topological data and one-dimensional boundary subsystem codes, formalized via the symplectic operator algebra of Pauli stabilizers.
The method is general and applies to $\mathbb{Z}_d$ qudits of both prime and nonprime dimension. Using the algorithm, the authors systematically enumerate and construct gapped boundaries and defect configurations for several important codes, including the $\mathbb{Z}_2$ and $\mathbb{Z}_4$ toric codes, the double semion and six-semion models, the color code, and the anomalous three-fermion model. They further apply the formalism to bivariate bicycle codes, revealing new types of boundaries supporting long-period translation symmetries and high logical degeneracy.
Specifically, this paper has the following strength:
-
The paper formulates a rigorous algebraic correspondence between bulk stabilizer groups and boundary gauge operators. Theorems such as the bulk-boundary correspondence (Theorem 4) and topological order completion (Theorem 10) provide a mathematically precise framework for boundary constructions in stabilizer codes.
-
By representing the stabilizer code algebra using Laurent polynomials, the authors turn the boundary condensation problem into an algorithmic problem, which can automatically compute all possible Lagrangian subgroups and corresponding lattice Hamiltonians.
-
The method reproduces known boundary classifications, yielding explicit lattice models for many boundaries and defects. The algorithmic method discussed in this paper also paves the way for computer-assisted calculation of gapped boundaries and domain walls.
Weaknesses
-
The results presented in this paper are restricted to gapped boundaries and defects of Abelian topological orders. In such cases, the boundaries correspond to Lagrangian subgroups of the bulk anyon theory. However, in more general settings, gapped boundaries and domain walls are characterized by Lagrangian algebras (or condensable algebras), which is lack of discussion in this paper.
-
Although the primary goal of the paper is to establish an algorithmic framework for classifying gapped boundaries and domain walls, the discussion of algorithmic complexity is implicit. It would be valuable to analyze how the computational cost scales with system parameters such as the local Hilbert space dimension.
-
From a theoretical perspective, gapped boundaries and domain walls of UMTCs are understood in terms of (bi)module categories. The paper does not yet clarify how the proposed operator-algebraic formalism connects to this categorical interpretation.
Report
Requested changes
I have the following suggestions:
-
A remark on non-Abelian topological orders is recommended. For example, a discussion on whether this method will still hold when the local Hilbert space becomes non-Abelian or categorical will be great. A discussion of the complexity of the algorithm will be beneficial as well.
-
Following prior studies on the lattice construction of gapped boundaries and defects, several works have employed the gauged SPT approach to realize such boundaries or domain walls (see, for example, 1006.5479, 1509.03626, 2208.07367). More recently, 2411.11967 suggests that for non-Abelian quantum double models, gapped domain walls can be constructed by gauging codimension-1 non-invertible SPTs. It would be valuable if the authors could comment on the potential connection between their operator-algebraic framework and this gauged-SPT-based classification of boundaries and domain walls.
-
As I commented before, a general discussion on how this formalism corresponds to the categorical understanding of gapped boundaries and domain walls will also be beneficial.
Recommendation
Publish (easily meets expectations and criteria for this Journal; among top 50%)

Anonymous on 2025-10-14 [id 5921]
I would like to comment on the article "Operator algebra and algorithmic construction of boundaries and defects in (2+1)D topological Pauli stabilizer codes" by Liang et al.
The article studies the systematic construction of boundaries of translationally invariant Pauli stabilizer codes in 2D, in particular boundary anyons and their condensation.
While this work is probably also interesting for other communities (e.g. condensed matter focused), it is safe to say that this work has opened a new pathway in the research direction of finding 2D local, translationally invariant CSS stabilizer codes. In particular, the works https://arxiv.org/abs/2504.09171 (accepted in PRL) and https://arxiv.org/abs/2504.08887 (accepted in PRX Quantum) are strongly influenced by this work and the novel, systematic way of constructing and studying boundaries.

---

## Editorial Decision

unknown